# Leveraging Drift to Improve Sample Complexity of Variance Exploding Diffusion Models

**Ruofeng Yang**
John Hopcroft Center for Computer Science
Shanghai Jiao Tong University
wanshuiyin@sjtu.edu.cn

**Zhijie Wang**
John Hopcroft Center for Computer Science
Shanghai Jiao Tong University
violetevergarden@sjtu.edu.cn

**Bo Jiang**
John Hopcroft Center for Computer Science
Shanghai Jiao Tong University
bjiang@sjtu.edu.cn

**Shuai Li**[*]
John Hopcroft Center for Computer Science
Shanghai Jiao Tong University
shuaili8@sjtu.edu.cn

## Abstract

Variance exploding (VE) based diffusion models, an important class of diffusion models, have shown state-of-the-art (SOTA) performance. However, only a few theoretical works analyze VE-based models, and those works suffer from a worse forward convergence rate $1/\text{poly}(T)$ than the $\exp(-T)$ of variance preserving (VP) based models, where $T$ is the forward diffusion time and the rate measures the distance between forward marginal distribution $q_T$ and pure Gaussian noise. The slow rate is due to the Brownian Motion without a drift term. In this work, we design a new drifted VESDE forward process, which allows a faster $\exp(-T)$ forward convergence rate. With this process, we achieve the first efficient polynomial sample complexity for a series of VE-based models with reverse SDE under the manifold hypothesis. Furthermore, unlike previous works, we allow the diffusion coefficient to be unbounded instead of a constant, which is closer to the SOTA models. Besides the reverse SDE, the other common reverse process is the probability flow ODE (PFODE) process, which is deterministic and enjoys faster sample speed. To deepen the understanding of VE-based models, we consider a more general setting considering reverse SDE and PFODE simultaneously, propose a unified tangent-based analysis framework, and prove the first quantitative convergence guarantee for SOTA VE-based models with reverse PFODE. We also show that the drifted VESDE can balance different error terms and improve generated samples without training through synthetic and real-world experiments.

## 1 Introduction

Recently, diffusion modeling has shown impressive performance in different areas [Ho et al., 2022, Rombach et al., 2022, Esser et al., 2024, Li et al., 2024]. Diffusion models consist of two processes: the forward and reverse process. The forward process gradually converts data $q_0$ to Gaussian noise, which can be described by an intermediate marginal distribution sequence $\{q_t\}_{t\in[0,T]}$. The reverse process sequentially predicts noise and removes it from data to generate samples.

There are two common forward processes: (1) Variance preserving (VP) SDE and (2) variance exploding (VE) SDE. The VPSDE corresponds to an Ornstein-Uhlenbeck process, and the stationary distribution is $\mathcal{N}(0, \mathbf{I})$. The VESDE has an exploding variance in the forward process. In earlier

---

[*]Corresponding author

38th Conference on Neural Information Processing Systems (NeurIPS 2024).

times, VP-based models [Ho et al., 2020, Lu et al., 2022] provide an important boost for developing diffusion models. Recently, VE-based models have shown the ability to generate data distribution supported on low-dimensional manifolds [Song and Ermon, 2019, 2020]. Since image and text datasets typically exhibit a low-dimensional manifold nature [Pope et al., 2021, Tang and Yang, 2024], VE-based models have achieved great performance in image generation, one-step generation, and reinforcement learning [Teng et al., 2023, Song et al., 2023, Ding and Jin, 2023]. Furthermore, Karras et al. [2022] unify VP and VESDE and prove that the ODE solution trajectory of a specific VESDE is linear and directly towards the data manifold, which makes the denoise process easy.

After determining a forward SDE, diffusion models reverse it and generate samples by running the corresponding reverse process. Since the reverse drift term contains the gradient of forward logarithmic density $\nabla \log q_t$ (a.k.a. score function), we estimate it by using the score matching technique [Vincent, 2011]. After that, diffusion models discretize the continuous reverse process and run this discrete process starting from pure Gaussian. There are two widely used reverse processes: reverse SDE [Ho et al., 2020] and probability flow ODE (PFODE) [Song et al., 2020a]. The reverse SDE usually generates higher quality samples [Kim et al., 2022]. The reverse PFODE always has a faster generation speed and is useful in other aspects such as calculating likelihoods [Song et al., 2020b] or obtaining one-step generation models [Song et al., 2023]. Hence, these processes are both critical, and providing the guarantee for VE-based models with these processes is necessary.

Recently, many works analyze the convergence guarantee of the VP-based diffusion models under the reverse SDE setting and prove that the VP-based models can sample from the target data distribution with polynomial complexity [Chen et al., 2023c,a,b, Lee et al., 2023, Benton et al., 2023]. As the first step of this work, we also analyze VE-based models under the reverse SDE setting. Different from the VP-based models, only a few works consider VE-based models and all of them suffer from slow $1/\text{Poly}(T)$ forward convergence rate [Lee et al., 2022, Gao et al., 2023, Gao and Zhu, 2024], which is worse than $\exp(-T)$ one for VPSDE. A slow forward convergence rate makes a large distance between $q_T$ and pure Gaussian noise, which leads to a large reverse beginning error. From the theoretical perspective, this error introduces hardness to balance three error sources, as shown in Section 5. From the empirical perspective, Lin et al. [2024] show that this error introduces a data information leakage problem, which leads to bad performance. To deal with this problem, De Bortoli et al. [2021] introduce a small drift term to obtain a $\exp\left(-\sqrt{T}\right)$ reverse beginning error. However, they introduce an additional $\exp\left(T\right)$ in the discretization error term. Furthermore, their results do not allow unbounded $\beta_t$, which is the key point of the optimal solution trajectory and used by the SOTA models [Karras et al., 2022, Song et al., 2023]. Therefore, the following question remains open:

*Is it possible to design a VESDE with a faster forward convergence rate than $1/poly(T)$ and achieve the polynomial sample complexity when the diffusion coefficient is unbounded?*

In this work, for the first time, we propose a new drifted VESDE forward process, which enjoys a faster forward convergence rate and allows unbounded coefficients. We first show that the drifted VESDE has similar trends but performs better than the original SOTA VESDE on synthetic data (Section 7). After that, we analyze the sample complexity of drifted VESDE under the realistic manifold hypothesis. The manifold hypothesis means the data $q_0$ is supported on a lower dimensional compact set $\mathcal{M}$, and much empirical evidence shows that image and text dataset satisfy this hypothesis [Fefferman et al., 2016, Pope et al., 2021, Tang and Yang, 2024]. Furthermore, as shown in Section 2, the manifold hypothesis is more realistic than previous data assumptions since it allows the blow-up phenomenon of the score function at the end of the reverse process, which matches the empirical observation [Kim et al., 2021]. Under the manifold hypothesis, we prove that the drifted VESDE with a suitable larger $\beta_t$ balances the reverse beginning, discretization, and approximated score errors and achieves the first efficient polynomial sample complexity for VE-based models with reverse SDE.

To better understand VE-based models, we analyze reverse SDE and PFODE simultaneously after obtaining polynomial complexity for reverse SDE. Despite the great performance, a few theoretical works consider reverse PFODE [Chen et al., 2023d,b, Gao and Zhu, 2024], and these works either focus on VPSDE or have strong assumptions. Hence, we propose the tangent-based framework for VE-based models and achieve the first quantitative convergence for the SOTA VE-based models with reverse PFODE. In conclusion, we accomplish the following results under the manifold hypothesis:

1. We propose a new drifted VESDE forward process, which allows $\exp\left(-T\right)$ forward convergence guarantee with suitable $\beta_t$. With this process, we achieve the first polynomial sample complexity for a series of VE-based models under the reverse SDE setting.

2. When considering the general setting, we propose the tangent-based unified framework and analyze reverse SDE and PFODE simultaneously. Under this framework, we prove the first quantitative guarantee for SOTA VE-based models with reverse PFODE.

3. We show that the drifted VESDE balances different error terms and improves generated samples without training via synthetic and real-world experiments.

## 2   Related Work

Before providing current results, we first discuss different assumptions about the data distribution from strong to weak. The strongest assumption is the log-concave distribution. While the log-Sobelev inequality (LSI) assumption is slightly weaker, it does not allow the presence of substantial non-convexity, which is far away from the multi-modal distribution. Recently, some works assume the score function is $L$-Lipschitz to allow the multi-modal distribution. However, this assumption can not explain the blow-up phenomenon of score [Kim et al., 2021]. The last assumption is the manifold hypothesis, which is supported by much empirical evidence and allows the blow-up score.

**Analyses for VP-based models.** For the reverse SDE, Lee et al. [2022] achieve the first polynomial complexity with strong LSI assumption. Chen et al. [2023c] remove the LSI assumption, assume the Lipschitz score and achieve polynomial complexity. Bortoli [2022] is the first work to focus on the sample complexity of diffusion models under the manifold hypothesis, and it is the most relevant work to our unified framework. However, as discussed in Section 6.1, the original tangent-based lemma can not deal with reverse PFODE even in VPSDE. We carefully control the tangent process to avoid additional $\exp(T)$ by using the exploding variance property of VESDE. Recently, Chen et al. [2023a] and Benton et al. [2023] also remove the Lipschitz score assumption, and Benton et al. [2023] achieve optimal dependence on $d$. More recently, Conforti et al. [2023] use bounded Fisher information assumption and replace $d$ with a Fisher information term.

For the PFODE, Chen et al. [2023d] propose the first quantitative result with exponential dependence on the Lipschitz constant. Chen et al. [2023b] achieve polynomial complexity by introducing a corrector component to inject suitable noise. More recently, Li et al. [2023] remove the additional corrector. However, their results rely heavily on the very specific $\beta_t$, which goes to $0$ as $T \to +\infty$. Since VE-based models have an unbounded $\beta_t$, this method is not suitable for our models.

**Analyses for VE-based models.** When considering VESDE, most works focus on constant $\beta_t$ and reverse SDE. De Bortoli et al. [2021] provide the first convergence guarantee with exponential dependence on $T$. Lee et al. [2022] analyze a constant diffusion coefficient VESDE and achieve polynomial sample complexity under the LSI assumption. When considering the reverse PFODE, Chen et al. [2023d] only consider the discretization error and provide a quantitative convergence guarantee. However, their results introduce additional $\exp(T)$ compared to ours (Section 6.1). Recently, Gao et al. [2023] and Gao and Zhu [2024] provide the polynomial results for a series of VESDE with reverse SDE and reverse PFODE under the log-concave assumption, respectively.

## 3   The Drifted Variance Exploding (VE) SDE

Diffusion models usually consist of a forward process and a reverse process. The forward process gradually injects noise to convert the data distribution to pure noise. To generate samples, diffusion models reverse the forward process and run the corresponding reverse process.

This section first recalls two previous forward processes: VPSDE and VESDE. Recently, the VE-based models achieve great performance in application [Karras et al., 2022, Song et al., 2023]. However, unlike the widely analyzed VP-based models [Benton et al., 2023, Chen et al., 2023b], the VE-based models suffer from challenges in obtaining an efficient sample complexity due to the slow forward convergence rate. To address this limitation, we introduce a new drifted VESDE forward process, which has a faster forward convergence rate, balances different error terms and achieves the first efficient polynomial sample complexity (see Section 5). Finally, we introduce how to reverse this new forward process and obtain an implementable algorithm.

### 3.1   The VP and VESDE of Diffusion Models

We first introduce the general form of the forward process and then recall two common forward processes, VPSDE and VESDE, adopted in previous works [Ho et al., 2020, Karras et al., 2022]. Let

$q_0$ be the data distribution. Given $\mathbf{X}_0 \in \mathbb{R}^d$, the forward process is defined by

$$\mathrm{d}\mathbf{X}_t = f(\mathbf{X}_t, t)\,\mathrm{d}t + g(t)\,\mathrm{d}\mathbf{B}_t, \quad \mathbf{X}_0 \sim q_0,$$

where $(\mathbf{B}_t)_{t \geq 0}$ is the standard Brownian motion in $\mathbb{R}^d$, $f(\mathbf{X}_t, t)$ is a drift coefficient and $g(t)$ is a diffusion coefficient. Let $q_t$ be the density function of $\mathbf{X}_t$ at time $t$. With a suitable choice of drift and diffusion terms (e.g. Section 3.1 and 3.2), the forward process gradually converts the data distribution into Gaussian noise. More specifically, the conditional distribution $\mathbf{X}_t | \mathbf{X}_0$ is exactly $\mathcal{N}(m_t \mathbf{X}_0, \sigma_t^2 \mathbf{I})$ given $\mathbf{X}_0$, where $m_t$ is determined by the drift term and $\sigma_t^2$ is determined by the diffusion term.

**The VPSDE forward process.** Let $\{\beta_t\}_{t \geq 0}$ be a non-decreasing sequence with bounded range $[1/\bar{\beta}, \bar{\beta}]$. The VPSDE has the following formula:

$$\mathrm{d}\mathbf{X}_t = -\beta_t \mathbf{X}_t\,\mathrm{d}t + \sqrt{2\beta_t}\,\mathrm{d}\mathbf{B}_t, \text{ where } \mathbf{X}_0 \sim q_0. \tag{1}$$

In this case, $m_t = \exp(-\int_0^t \beta_s \mathrm{d}s)$ and $\sigma_t^2 = 1 - m_t^2$. Note that $m_T \leq \exp(-T/\bar{\beta})$, which indicates a fast forward convergence rate $\mathrm{TV}(q_T | \mathcal{N}(0, \mathbf{I})) \leq \exp(-T/\bar{\beta})$ [Chen et al., 2023c].

**The VESDE forward process.** The VESDE forward process is defined without a drift term:

$$\mathrm{d}\mathbf{X}_t = \sqrt{\mathrm{d}\sigma_t^2/\mathrm{d}t}\,\mathrm{d}\mathbf{B}_t, \text{ where } \mathbf{X}_0 \sim q_0. \tag{2}$$

Two common choices for $\sigma_t^2$ are $t$ and $t^2$, with the latter achieving SOTA performance [Karras et al., 2022, Teng et al., 2023]. However, VESDE only has a slow polynomial-decay forward convergence rate (Theorem 4.2), which introduces hardness to obtain an efficient sample complexity (see Section 5). This motivates us to design an improved VESDE process with a fast forward convergence rate.

### 3.2 The Drifted VESDE Forward Process

Note that the forward convergence rate of the general process is upper bounded by $m_T/\sigma_T^2$ (Theorem 4.2). In practical applications [Ho et al., 2020, Karras et al., 2022, Song et al., 2023], the variance of the forward process $\sigma_t^2$ at time $T$, which is determined by the diffusion term, does not exceed $T^2$. This indicates the contribution of $\sigma_T^2$ to the forward convergence rate is only $1/\mathrm{Poly}(T)$. Hence, the exponential-decay forward convergence rate of VPSDE comes from the drift term, which introduces an exponential-decay $m_t \leq \exp(-T/\bar{\beta})$. Due to the absence of the drift term in VESDE, the data information, such as expectation $\mathbb{E}[q_0]$ and covariance $\mathrm{Cov}[q_0]$, does not decay and $m_t \equiv 1$, which is a key to an only polynomial-decay forward convergence rate $m_T/\sigma_T^2 \leq 1/\mathrm{Poly}(T)$. With the drift term, the VPSDE gradually removes the data information from $q_t$ during the process, which makes $q_t$ quickly converge to pure Gaussian noise. Inspired by this elimination effect of the drift term, we propose a drifted VESDE forward process:

$$\mathrm{d}\mathbf{X}_t = -\frac{1}{\tau}\beta_t \mathbf{X}_t\,\mathrm{d}t + \sqrt{2\beta_t}\,\mathrm{d}\mathbf{B}_t, \quad \mathbf{X}_0 \sim q_0, \tag{3}$$

where $\tau \in [T, T^2]$ is the coefficient used to balance the drift and diffusion term [2], and $\{\beta_t\}_{t \geq 0}$ is a positive non-decreasing sequence. In this case,

$$m_t = \exp\left(-\int_0^t \beta_s/\tau\,\mathrm{d}s\right) \text{ and } \sigma_t^2 = \tau\left(1 - m_t^2\right). \tag{4}$$

We show that the drifted VESDE is not only an effective representation of the existing VESDE but also extends beyond it (see Section 7 and Appendix A.1). We also prove that this process with suitable $\beta_t$ has a $\exp(-T)$ forward convergence rate and enjoys an efficient polynomial sample complexity.

### 3.3 The Reverse Process of the Drifted VESDE

To generate samples from Gaussian noise, a diffusion model reverses the forward process. Let $q_t^\tau$ be the density function of the drifted VESDE forward process at time $t$ and $(\mathbf{Y}_t)_{t \in [0,T]} = (\mathbf{X}_{T-t})_{t \in [0,T]}$. As shown in Cattiaux et al. [2021], the reverse process of drifted VESDE has the following form [3]:

$$\mathrm{d}\mathbf{Y}_t = \beta_{T-t}\left\{\mathbf{Y}_t/\tau + (1+\eta^2)\nabla \log q_{T-t}^\tau(\mathbf{Y}_t)\right\}\mathrm{d}t + \eta\sqrt{2\beta_{T-t}}\,\mathrm{d}\mathbf{B}_t. \tag{5}$$

---

[2] We note that the choice $\tau \in [T, T^2]$ is used to guarantee the exploding variance of the forward process. In fact, our drifted VESDE is a general formula that covers the VP and VESDE by choosing $\tau \in [1, T^2]$. More details are shown in Section 5.

[3] Now $(\mathbf{B}_t)_{t \geq 0}$ is the reversed Brownian motion, and we abuse this notation here for ease of notation.

The parameter $\eta \in [0, 1]$ is used to determine the type of reverse processes. There are two common reverse processes in the application: reverse probability flow ODE (PFODE) ($\eta = 0$) [Song et al., 2020b, 2023] and reverse SDE ($\eta = 1$) [Ho et al., 2020].

To generate distribution $q_0$ through running the above reverse process, diffusion models need the true score function $\nabla \log q_{T-t}^\tau(\mathbf{Y}_t)$ and the accurate reverse beginning distribution $q_T^\tau$. However, $\nabla \log q_{T-t}^\tau(\mathbf{Y}_t)$ and $q_T^\tau$ contain the data information and usually can not be exactly calculated. For the score function, diffusion models approximate it using a score network $\mathbf{s}(T - t, \cdot)$ by minimizing the score matching objective function [Vincent, 2011]. For the initial distribution of the reverse process, since $q_T^\tau$ should be close to a pure Gaussian, we choose $q_\infty^\tau = \mathcal{N}(0, \sigma_T^2 \mathbf{I})$ as an approximation. Then, the continuous reverse process $(\widehat{\mathbf{Y}}_t)_{t \in [0,T]}$, incorporating $\mathbf{s}(T - t, \cdot)$ and $q_\infty^\tau$, is defined as:

$$\mathrm{d}\widehat{\mathbf{Y}}_t = \beta_{T-t}\big\{\widehat{\mathbf{Y}}_t/\tau + (1 + \eta^2)\mathbf{s}(T - t, \widehat{\mathbf{Y}}_t)\big\}\mathrm{d}t + \eta\sqrt{2\beta_{T-t}}\,\mathrm{d}\mathbf{B}_t\,, \text{ where } \widehat{\mathbf{Y}}_0 \sim q_\infty^\tau\,.$$

Since diffusion models can not run a continuous process due to the nonlinear score function, these models usually discretize the above continuous process and freeze the approximated score at the beginning of each interval. Let $\{\gamma_k\}_{k \in [K]}$ be the stepsize and $t_{k+1} = \sum_{j=0}^{k} \gamma_j$. As shown in Kim et al. [2021], $\nabla \log q_{T-t}^\tau(\mathbf{Y}_t)$ goes to $+\infty$ at the end of the reverse process. To mitigate this issue, they use the early stopping technique $t_K = T - \delta$, and we also employ this technique in this work. With the stepsize, we choose the exponential integrator discretization scheme [Zhang and Chen, 2022] to discretize the above process, which runs the following process:

$$\mathrm{d}\widetilde{\mathbf{Y}}_t = \beta_{T-t}\big\{\widetilde{\mathbf{Y}}_t/\tau + (1 + \eta^2)\mathbf{s}(T - t_k, \widetilde{\mathbf{Y}}_{t_k})\big\}\mathrm{d}t + \eta\sqrt{2\beta_{T-t}}\,\mathrm{d}\mathbf{B}_t\,, \text{ where } t \in [t_k, t_{k+1}]\,. \quad (6)$$

As shown in Karras et al. [2022], the choice of $\beta_t$ significantly affects the performance of models, and we need to determine $\beta_t$ before running the reverse process. The state-of-the-art diffusion models adopt $\beta_t = t$, which increases rapidly and has an unbounded range. However, current theoretical works assume $\beta_t$ to be a constant [Chen et al., 2023c] or confined to a bounded interval $[1/\bar{\beta}, \bar{\beta}]$ [Bortoli, 2022] to match the setting of VPSDE. To align more closely with practical applications of VE-based models, we allow an unbounded $\beta_t$ in this work. Furthermore, we make a detailed assumption on $\beta_t$ when considering different reverse processes.

**Assumption 3.1.** Let $\{\beta_t\}_{t \geq 0}$ be a positive, non-decreasing sequence. For any $\tau \in [T, T^2]$, there exists constants $\bar{\beta}$ and $C$, such that for any $t \in [0, T]$: (1) for $\eta = 0$, then $1/\bar{\beta} \leq \beta_t \leq \max\{\bar{\beta}, t\}$ and $\int_0^T \beta_t/\tau \, \mathrm{d}t \leq C$; (2) for $\eta = 1$, then $1/\bar{\beta} \leq \beta_t \leq \max\{\bar{\beta}, t^2\}$.

As shown in Chen et al. [2023b], due to the absence of the stochasticity, the small errors for quickly accumulate and are magnified. Hence, we assume a conservative $\beta_t$ for the reverse PFODE, whose growth rate is at most $t$, to avoid an additional $\exp(T)$ in the convergence guarantee (see Section 6.1). We note that this choice of $\beta_t$ is satisfied in practical applications [Song et al., 2020b, Karras et al., 2022]. For the reverse SDE setting, we assume the growth rate of $\beta_t$ can depend on $\tau$ instead of at most linear. For example, when $\tau = T^2$, we can choose $\beta_t = t^2$, which has the same order as $\tau$. As shown in Theorem 4.2, the drifted VESDE with aggressive $\beta_t = t^2$ has an exponential-decay forward convergence rate, which leads to the first efficient polynomial complexity for VE-based models.

**Notations.** For $x \in \mathbb{R}^d$ and $A \in \mathbb{R}^{d \times d}$, we denote by $\|x\|$ and $\|A\|$ the Euclidean norm for vector and the spectral norm for matrix. We denote by $\bar{\gamma}_K = \mathrm{argmax}_{k \in \{0,...,K-1\}} \gamma_k$ the maximum stepsize for $k \in [0, K-1]$. We denote by $q_0 P_T$ the distribution of $\mathbf{X}_T$, $Q_{t_K}^{q_\infty^\tau}$ the distribution of $\mathbf{Y}_{t_K}$, $R_K^{q_\infty^\tau}$ the distribution of $\widetilde{\mathbf{Y}}_{t_K}$ and $Q_{t_K}^{q_0 P_T}$ the distribution which does reverse process starting from $q_T^\tau$ (Eq. 5). We denote by $\mathrm{W}_1$ and $\mathrm{W}_2$ the Wasserstein distance of order one and two, respectively.

# 4  The Faster Forward Convergence Rate for the Drifted VESDE

This section shows that the drifted VESDE has a fast forward convergence rate. Since $q_T^\tau$ contains the data information, we first introduce the manifold hypothesis before controlling $\mathrm{TV}(q_T^\tau, q_\infty^\tau)$.

**Assumption 4.1.** $q_0$ is supported on a compact set $\mathcal{M}$ and $0 \in \mathcal{M}$.

We denote $R$ the diameter of the manifold by $R = \sup\{\|x - y\| : x, y \in \mathcal{M}\}$ and assume $R > 1$. As shown in Section 1, the manifold hypothesis is supported by much empirical evidence [Bengio

et al., 2013, Fefferman et al., 2016, Pope et al., 2021] and allows the blow-up phenomenon of the score. Recently, Tang and Yang [2024] show that diffusion models can adapt to the intrinsic manifold structure. With Assumption 4.1, we obtain the forward process guarantee for the drifted VESDE.

**Theorem 4.2.** *Assume Assumption 4.1 and 3.1. Let $q_\infty^\tau = \mathcal{N}(0, \sigma_T^2 \mathbf{I})$. With $m_T, \sigma_T$ defined in Equation* (4)*, we have* $\mathrm{TV}(q_T^\tau, q_\infty^\tau) \leq \sqrt{m_T}\bar{D}/\sigma_T$*, where* $\bar{D} = d|c| + \mathbb{E}[q_0] + R$ *and* $c$ *is the eigenvalue of Cov*[$q_0$] *with the largest absolute value.*

Recall that $m_T = \exp(-\int_0^T \beta_t/\tau \, dt)$, the previous VESDE [Song et al., 2020b, Karras et al., 2022, Lee et al., 2022] chooses a conservative $\beta_t$ satisfies $\int_0^T \beta_t/\tau \, dt \leq C$. However, with an aggressive $\beta_t$, the drifted VESDE will have a faster convergence rate. To illustrate the accelerated forward process, we use $\tau = T^2$ as an example and discuss different $\beta_t = t^{\alpha_1}, \alpha_1 \in [1, 2]$. Due to the definition of $\sigma_T$, $\sigma_T \approx T$, and the forward convergence rate mainly depends on $\sqrt{m_T}$. When $\alpha_1 = 1$ is conservative, $m_T$ is a constant, and the convergence rate is $1/T$. When $\alpha_1 = 1 + \ln(2r\ln(T))/\ln(T)$ is slightly aggressive, the convergence rate is $1/T^{r+1}$ for $r > 0$. When $\alpha_1 \geq 1 + \ln(T - \ln(T))/\ln(T)$ is aggressive, the convergence rate is faster than $\exp(-T)$. In our analysis, whether $\beta_t$ can be aggressive depends on the reverse process (see Section 6). When choosing an aggressive $\beta_t$, the drifted VESDE achieves an improved sample complexity compared with pure VESDE (see Section 5).

## 5 The Polynomial Complexity for a Series of VESDE with Reverse SDE

In this section, we first pay attention to the reverse SDE ($\eta = 1$) to show the power of the drifted VESDE. More specifically, we show that our general drifted VESDE form covers the current models (VP and VESDE). After that, we show that drifted VESDE can go beyond the current models and achieve an improved complexity with an aggressive $\beta_t$. Since the objective function minimizes the $L_2$ distance between the ground truth and the approximated score, we assume that the approximated score is $L_2$-accuracy, which is exactly the same with Chen et al. [2023c] and Benton et al. [2023].

**Assumption 5.1.** $\mathbb{E}_{q_{t_k}}\left[\left\|s_{t_k} - \nabla \ln q_{t_k}^\tau\right\|^2\right] \leq \epsilon_{\mathrm{score}}^2$ for $\forall k \in [K]$.

With this assumption, we provide a universe convergence guarantee for $\tau \in [1, T^2]$ and $\beta_t \in [1, t^2]$ and discuss the sample complexity of VP and VE-based models in detail.

**Theorem 5.2.** *Assume Assumption 3.1, 4.1, 5.1. Let $\bar{D}$ defined in Theorem 4.2, $\bar{\gamma}_K = \mathrm{argmax}_{k \in \{0,...,K-1\}} \gamma_k$, $\tau = T^2$ and $\beta_t \in [1, t^2]$. Then, we have that*

$$\mathrm{TV}\left(R_K^{q_\infty^\tau}, q_\delta\right) \leq \frac{\bar{D}\sqrt{m_T}}{\sigma_T} + \frac{R^2\sqrt{d}}{\sigma_\delta^4}\sqrt{\bar{\gamma}_K \beta_T \tau T} + \epsilon_{\mathrm{score}}\sqrt{\beta_T T}.$$

To guarantee the above convergence guarantee smaller than $\widetilde{O}(\epsilon_{\mathrm{TV}})$, each component of the result needs to be smaller than $\epsilon_{\mathrm{TV}}$. As shown in Remark 5.3, it is difficult for pure VESDE to balance the approximated score and the first two error terms to achieve an efficient sample complexity. Hence, we discuss how to balance the reverse beginning and discretization error. More specifically, we require $\bar{D}\sqrt{m_T}/\sigma_T \leq \epsilon_{\mathrm{TV}}$ and $\bar{\gamma}_K \leq \sigma_\delta^8 \epsilon_{\mathrm{TV}}^2/\left(R^4 d\beta_T \tau T\right)$, where the first inequality determines the order of $T$ and the second inequality determines the stepsize $\bar{\gamma}_K$. After that, with sample complexity $K = T/\bar{\gamma}_K$, we have that $\mathrm{TV}(R_K^{q_\infty^\tau}, q_\delta) \leq \widetilde{O}(\epsilon_{\mathrm{TV}})$. The last step is to guarantee $q_\delta$ and $q_0$ is close enough $W_2^2(q_0, q_\delta) \leq \epsilon_{W_2}^2$, which requires $\sigma_\delta^2 \leq \epsilon_{W_2}^2/(d + R\sqrt{d})$.

Following the above process, this general convergence guarantee leads to the polynomial sample complexity for VP and VE-based models. When $\beta_t = 1$ and $\tau = 1$, the drifted VESDE becomes VPSDE and achieve the complexity $\widetilde{O}(1/(\epsilon_{W_2}^8 \epsilon_{\mathrm{TV}}^2))$, which achieve exactly the same order compared with Chen et al. [2023c]. When $\beta_t = 1$ and $\tau = T$, our formula is similar but slightly better (Figure 2 and 3) to pure VESDE ($\sigma_t^2 = t$) and achieves $O(1/\epsilon_{W_2}^8 \epsilon_{\mathrm{TV}}^8)$ result. For $\beta_t = t$ and $\tau = T^2$, the general formula is similar to SOTA pure VESDE ($\sigma_t^2 = t^2$) and achieves the first polynomial results $O(1/\epsilon_{W_2}^8 \epsilon_{\mathrm{TV}}^7)$ for this model under the manifold hypothesis[4].

Although we achieve the first polynomial sample complexity for VE-based models under the manifold hypothesis, it is clear that the results of the VE-based models are significantly worse than the result

---

[4]We note these results still hold for pure VESDE with $\sigma_t^2 = t$ and $t^2$, and we use the general drifted VESDE for simplicity. Here, we only consider the dependence of $\epsilon$. Readers can find detailed results in Appendix C.1.

of VP-based models since the slow $1/\text{Poly}(T)$ forward convergence guarantee. More specifically, the forward convergence rate of VPSDE is $\exp(-T)$, which means $T$ has the order $\log(1/\epsilon_{\text{TV}})$ and can be ignore. When considering the pure VESDE with $\sigma_t^2 = t^2$, the forward convergence rate is $\bar{D}/T$, which indicates $T \geq \bar{D}/\epsilon_{\text{TV}}$ is a polynomial term and can not be ignored. Hence, the results $K = R^4 dT^5/(\sigma_\delta^8 \epsilon_{\text{TV}}^2)$ of pure VESDE ($\sigma_t^2 = t^2$) is heavily influenced by $T$. When considering pure VESDE with $\sigma_t^2 = t$, it suffers from a slower forward convergence guarantee $\bar{D}/\sqrt{T}$, which indicates $T \geq \bar{D}^2/\epsilon_{\text{TV}}^2$ and $K = R^4 dT^3/(\sigma_\delta^8 \epsilon_{\text{TV}}^2)$. This is the first work to explain why pure VESDE ($\sigma_t^2 = t^2$) performs better than pure VESDE ($\sigma_t^2 = t^2$) from a theoretical perspective.

*Remark* 5.3. In this part, we explain the reason why the pure VESDE fails to balance the reverse beginning and the approximated score. We use the pure VESDE with $\sigma_t^2 = t^2$ as an example. Under this setting, the guarantee has the form $1/T + \sqrt{\bar{\gamma}_K T^4}/\delta^4 + \epsilon_{\text{score}}\sqrt{T^2}$, which requires $T \geq 1/\epsilon_{\text{TV}}$. Then, $\epsilon_{\text{score}}\sqrt{T^2}$ is larger than $\epsilon_{\text{score}}/\sqrt{\epsilon_{\text{TV}}^2}$. Hence, it is hard to achieve non-asymptotic results.

**Drifted VESDE with an aggressive $\beta_t$ balances different error terms.** This part shows that our drifted VESDE with a suitable aggressive $\beta_t$ can balance the above three error terms. More specifically, we show that introducing aggressive $\beta_t$ only slightly affects the discretization error and significantly benefits in balancing reverse beginning and approximated errors. As a result, we obtain an efficient polynomial complexity for a series of VE-based models with unbounded $\beta_t$.

**Corollary 5.4.** *Following the setting of Theorem 5.2. When considering $\tau = T^2, \beta_t = t^2$, by choosing $\delta \leq \frac{\epsilon_{W_2}^{2/3}}{(d+R\sqrt{d})^{1/3}}$, $T \geq 2\ln(\bar{D}/\epsilon_{TV})$, $\bar{\gamma}_K \leq \delta^{12}\epsilon_{TV}^2 \ln^5(\bar{D}/\epsilon_{TV})/(R^4 d)$ and assuming $\epsilon_{\text{score}} \leq \widetilde{O}(\epsilon_{\text{TV}})$, $R_K^{q_\infty^\tau}$ is $(\epsilon_{\text{TV}} + \epsilon_{\text{score}})$ close to $q_\delta$, which is $\epsilon_{W_2}$ close to $q_0$, with sample complexity*

$$K \leq \widetilde{O}\left(\frac{dR^4(d+R\sqrt{d})^4}{\epsilon_{W_2}^8 \epsilon_{\text{TV}}^2}\right).$$

*Defined by $R_{K,R_0}^{q_\infty^\tau}$ the output $R_K^{q_\infty^\tau}$ projected onto $\mathrm{B}(0, R_0)$ for $R_0 = \widetilde{\Theta}(R)$. Then, we achieve pure $W_2$ guarantee $W_2(R_{K,R_0}^{q_\infty^\tau}, q_0) \leq \epsilon_{W_2}$ with sample complexity $\widetilde{O}\left(\frac{dR^8(d+R\sqrt{d})^4}{\epsilon_{W_2}^{12}}\right)$.*

Since our drifted VESDE with $\tau = T^2$ and $\beta_t = t^2$ has a fast forward convergence rate $\exp(-T)/T^2$, $T$ becomes a logarithmic term and does not influence the discretization term, which is the source of the improved sample complexity. Furthermore, the requirement of $\epsilon_{\text{score}}$ has the same order with $\epsilon_{\text{TV}}$, which indicates the drifted VESDE balances the reverse beginning and approximated score error. In fact, we only require the forward convergence rate of drifted VESDE is $\exp(-T)$, which indicates a series of VE-based models can achieve this sample complexity. We use $\tau = T^2$ as an example. When considering the $\beta_t = t^{\alpha_1}$, we require $2 \geq \alpha_1 \geq 1 + \ln(T - \ln(T))/\ln(T)$ to enjoy $\exp(-T)$ forward convergence rate and achieve $K \leq \widetilde{O}\left(1/(\epsilon_{W_2}^8 \epsilon_{\text{TV}}^2)\right)$ sample complexity. For $\beta_t = t$ and $\tau = T$, we also obtain complexity $K \leq \widetilde{O}\left(1/\left(\epsilon_{W_2}^8 \epsilon_{\text{TV}}^2\right)\right)$ (Appendix C.1).

*Remark* 5.5. Recently, Lee et al. [2022] and Gao et al. [2023] consider the sample complexity of VESDE with reverse SDE under strong assumption. Lee et al. [2022] consider VESDE with $\sigma_t^2 = t$ and achieve $\tilde{O}(L^2/\epsilon_{\text{TV}}^4)$ result under the LSI assumption. Under the manifold hypothesis, the result is $\tilde{O}(1/\epsilon_{W_2}^8 \epsilon_{\text{TV}}^4)$, which is worse than Corollary 5.4. Gao et al. [2023] achieve pure $W_2$ guarantee $\widetilde{O}(1/\epsilon_{W_2}^{2.5})$ under the log-concave distribution, which is even stronger than LSI assumption and ignore the influence of $\delta$. Hence, they ignore an additional $1/\text{Poly}(W_2)$ (Detail in Appendix A.2).

## 6   The Tangent-based Analysis Framework

To deepen the understanding of VE-based models instead of the specific reverse process, we introduce the unified framework for VESDE with reverse SDE and PFODE. Similar to previous PFODE work [Chen et al., 2023d], we assume an accurate score and consider the other errors.

**Theorem 6.1.** *Assume Assumption 3.1 and 4.1, $\delta \leq 1/32$ and $\gamma_k \sup_{v \in [T-t_{k+1}, T-t_k]} \beta_v/\sigma_v^2 \leq 1/28$ for $\forall k \in \{0, ..., K-1\}$. Let $\gamma_K = \delta$. Then, for $\forall \tau \in [T, T^2]$:*

*(1) If $\eta = 1$ (the reverse SDE), choosing $\beta_t = t^2$, $W_1\left(R_K^{q_\infty^\tau}, q_0\right)$ is bounded by*

$$(\frac{R}{\tau} + \sqrt{d})\sqrt{\delta} + \exp\left(\frac{R^2}{2}(\frac{\bar{\beta}}{\delta^3} + \frac{1}{\tau})\right)\left(C_1(\tau)T\kappa_1^2(\tau)\left((\frac{\bar{\beta}}{\delta^3} + \frac{1}{\tau})\bar{\gamma}_K^{1/2} + 1\right)\bar{\gamma}_K^{1/2} + \frac{\bar{D}e^{-T/2}}{\sqrt{\tau}}\right),$$

*where $\kappa_1(\tau) = T^2(1/\tau + \bar{\beta}/\delta^3)$ and $C_1(\tau)$ is linear in $\tau^2$.*

*(2) If $\eta = 0$ (PFODE), choosing a conservative $\beta_t$ (Assumption 3.1), $W_1\left(R_K^{q_\infty^\tau}, q_0\right)$ is bounded by*

$$(\frac{R}{\tau} + \sqrt{d})\sqrt{\delta} + \exp\left(\frac{R^2}{2}(\frac{\bar{\beta}}{\delta^2} + \frac{1}{\tau})\right)\left(C_2(\tau)\kappa_2^2(\tau)T\left((\frac{\bar{\beta}}{\delta^2} + \frac{1}{\tau})\bar{\gamma}_K^{1/2} + 1\right)\bar{\gamma}_K^{1/2} + \frac{\bar{D}}{\sqrt{\tau}}\right),$$

*where $\kappa_2(\tau) = T\left(1/\tau + \bar{\beta}/\delta^2\right)$ and $C_2(\tau)$ is linear in $\tau^2$.*

Theorem 6.1 proves the first quantitative guarantee for VE-based models with reverse PFODE using the unified tangent-based framework. Correspondingly, the Girsanov-based method [Chen et al., 2023c,a] can not deal with reverse PFODE since the reverse process diffusion term is not well-defined. Recently, Chen et al. [2023d] employ the Restoration-Degradation framework to analyze VESDE with reverse PFODE, which also has exponential dependence on $R$ and $\delta$. Furthermore, their results have exponential dependence on $\beta_t$ ($g_{\max}$ in Chen et al. [2023d]), which corresponds to $\tau$. However, our dependence on $\tau$ appears in the polynomial term. Hence, our framework is a suitable unified framework. Furthermore, we emphasize that our tangent-based unified framework is not a simple extension of Bortoli [2022]. We carefully control the tangent process according to the variance exploding property of VESDE to avoid $\exp(T)$ term when considering PFODE (Section 6.1).

Theorem 6.1 has exponential dependence on $R$ and $\delta$, which is introduced by the tangent process. Similar to Bortoli [2022], if we assume the Hessian $\left\|\nabla^2 \log q_t(x_t)\right\| \leq \Gamma/\sigma_t^2$, we obtain a better control on the tangent process and replace the exponential dependence on $\delta$ by a polynomial dependence on $\delta$ and exponential dependence on $\Gamma$ when considering reverse PFODE.

**Corollary 6.2.** *Assume Assumption 3.1, 4.1 and $\left\|\nabla^2 \log q_t(x_t)\right\| \leq \Gamma/\sigma_t^2$. Let $\eta = 0$ (reverse PFODE), $\delta \in (0, 1/32), \tau = T^2, \beta_t = t$ and $\kappa_2(\tau), C_2(\tau)$ defined in Theorem 6.1, we have*

$$W_1\left(R_K^{q_\infty^\tau}, q_0\right) \leq (\frac{R}{\tau} + \sqrt{d})\sqrt{\delta} + \frac{\bar{\beta}^{\frac{\Gamma}{2}}}{\delta^\Gamma}\exp\left(\frac{\Gamma+2}{2}\right)\left(C_2(\tau)\kappa_2^2(\tau)T((\frac{\bar{\beta}}{\delta^2} + \frac{1}{\tau})\bar{\gamma}_K^{1/2} + 1)\bar{\gamma}_K^{1/2} + \frac{\bar{D}}{\sqrt{\tau}}\right).$$

Though the additional assumption is strong, many special cases, such as hypercube $\mathcal{M} = [-1/2, 1/2]^p$ satisfy this assumption. We emphasize that our analysis also holds for VESDE ($\sigma_t^2 = t^2$) with reverse PFODE, which means our results can explain the SOTA model in Karras et al. [2022].

## 6.1 The Discussion on the Unified Framework

In this section, we introduce the unified tangent-based framework for reverse SDE and PFODE and discuss key steps to achieve the quantitative guarantee for PFODE. Firstly, we decompose the goal $W_1\left(R_K^{q_\infty^\tau}, q_0\right)$ into three terms: $W_1\left(R_K^{q_\infty^\tau}, Q_{t_K}^{q_\infty^\tau}\right) + W_1\left(Q_{t_K}^{q_\infty^\tau}, Q_{t_K}^{q_0 P_T}\right) + W_1\left(Q_{t_K}^{q_0 P_T}, q_0\right)$.

These terms correspond to the discretization scheme, reverse beginning distribution, and the early stopping parameter $\delta$. We focus on most difficult discretization term and first recall the stochastic flow of the reverse process for any $x \in \mathbb{R}^d$ and $s, t \in [0, T]$ with $t \geq s$:

$$d\mathbf{Y}_{s,t}^x = \beta_{T-t}\left\{\mathbf{Y}_{s,t}^x/\tau + \left(1 + \eta^2\right)\nabla \log q_{T-t}\left(\mathbf{Y}_{s,t}^x\right)\right\}dt + \eta\sqrt{2\beta_{T-t}}d\mathbf{B}_t, \text{ where } \mathbf{Y}_{s,s}^x = x.$$

With $\nabla\mathbf{Y}_{s,s}^x = \mathbf{I}$, the corresponding tangent process is

$$d\nabla\mathbf{Y}_{s,t}^x = \beta_{T-t}\nabla\mathbf{Y}_{s,t}^x/\tau dt + \beta_{T-t}\left(1 + \eta^2\right)\nabla^2 \log q_{T-t}(\mathbf{Y}_{s,t}^x)\nabla\mathbf{Y}_{s,t}^x dt.$$

The key of the discretization error is to bound tangent process $\left\|\nabla\mathbf{Y}_{s,t_K}^x\right\|$. For this term, we consider the reverse SDE and PFODE simultaneously and propose a general version of Bortoli [2022].

**Lemma 6.3.** *Assume Assumption 3.1 and 4.1. For $\forall s \in [0, t_K]$ and $x \in \mathbb{R}^d$, we have*

$$\left\|\nabla\mathbf{Y}_{s,t_K}^x\right\| \leq \exp\left(\frac{R^2}{2\sigma_{T-t_K}^2} + \frac{(1-\eta^2)}{2}\int_0^{t_K}\frac{\beta_{T-u}}{\tau}du\right).$$

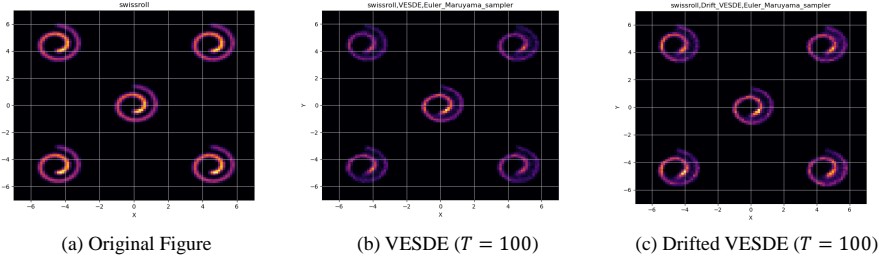

| (a) Original Figure | (b) VESDE ($T = 100$) | (c) Drifted VESDE ($T = 100$) |

Figure 1: Experiment results of Swiss roll with Euler Maruyama Method (Reverse SDE)

We emphasize that the general bound for the tangent process is the key to achieving the guarantee for VESDE with the reverse ODE. Recall that in the original lemma for the tangent processes, since $\tau$ is independent of $T$ and $\beta_t$ is bounded in a small interval $[1/\bar{\beta}, \bar{\beta}]$, $\int_0^{t_K} \beta_{T-u}/\tau \mathrm{d}u = \Theta(T)$, which means there is an additional $\exp(T)$ when considering VPSDE with revere PFODE. However, our tangent-based lemma makes use of the variance exploding property of VESDE to guarantee that $\int_0^T \beta_t/\tau \mathrm{d}t \leq C$ with a conservative $\beta_t = t$ when considering reverse PFODE. When $\eta = 1$, we choose aggressive $\beta_t = t^2$ since the choice of $\beta_t$ does not affect the bound of the tangent process.

For the early stopping term, it corresponds to $\delta$ and is smaller than $2(R/\tau + \sqrt{d})\sqrt{\delta}$. Since we can not use the data processing inequality in Wasserstein distance, the reverse beginning terms consists of the bound of tangent process term and the forward process term:

$$W_1\left(Q_{t_K}^{q_\infty^\tau}, Q_{t_K}^{q_0 P_T}\right) \leq \frac{\sqrt{m_T}\bar{D}}{\sigma_T} \exp\left(\frac{R^2}{2\sigma_{T-t_K}^2} + \frac{(1-\eta^2)}{2} \int_0^{t_K} \frac{\beta_{T-u}}{\tau} \mathrm{d}u\right).$$

One notable future work is introducing the short regularization technique [Chen et al., 2023b] and suitable corrector to remove the above exponential dependence.

## 7 Experiments

In this section, we show the power of the drifted VESDE forward process through experiments. Section 7.1 shows that aggressive one achieves good balance in different error terms. After that, we consider the approximated score and show that the conservative one can improve the quality of the generated distribution without training in the synethic and real-world setting.

### 7.1 The Aggressive Drifted VESDE Balances Errors

In this section, we do experiments on 2-D Gaussian to show that the aggressive drifted VESDE balances different errors. Since the ground truth score of the Gaussian can be directly calculated, we use the accurate score function to discuss the balance between the other two error terms. We show how to use approximated score in Section 7.2.

As shown in Figure 2, the process with aggressive $\beta_t = t^2$ achieves the best and second performance in EI and EM discretization, which supports our theoretical result (Corollary 5.4). The third best process is conservative $\beta_t = t$ with the small drift term. The reason is that though it can not achieve a $\exp(-T)$ forward guarantee, it also has a constant decay on prior information, which slightly reduces the effect of the reverse beginning error (Section 3.1). The worst process is pure VESDE since it is hard to balance different error sources. Our experimental results also show that EI is better than EM.

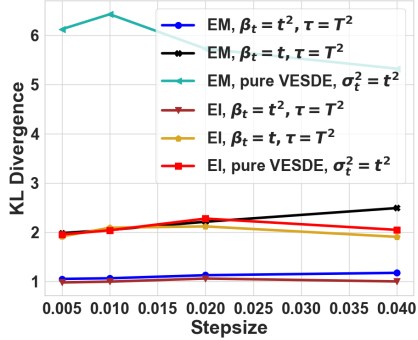

Figure 2: Results of 2-D Gaussian

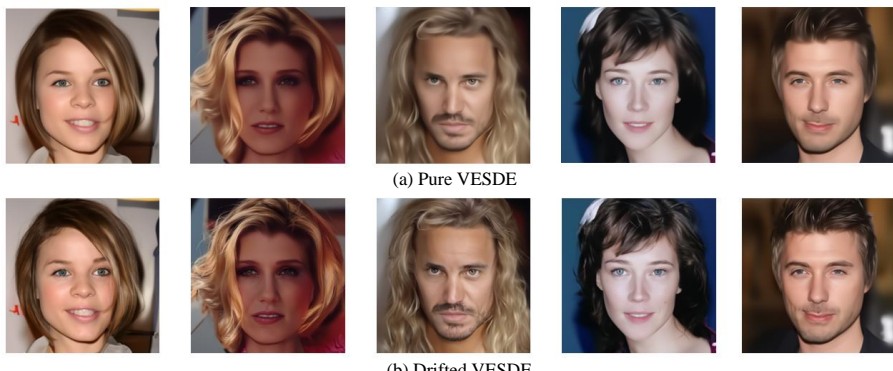

(a) Pure VESDE

(b) Drifted VESDE

Figure 3: Experiment results of CelebA dataset

## 7.2 The Conservative Drifted VESDE Benefits from VESDE without Training

As shown in Figure 2, the red and orange lines have similar trends. Hence, for conservative drifted VESDE, which satisfies (2) of Assumption 3.1, we can directly use the models trained by pure VESDE to improve the quality of generated distribution. We confirm our intuition by training the model with pure VESDE with $\sigma_t^2 = t$ and directly use the models to conservative drifted VESDE with $\beta_t = 1$ and $\tau = T$. From the experimental results (Figure 1), it is clear that pure VESDE has a low density on the Swiss roll except for the center one, which indicates pure VESDE can not deal with large dataset variance, as we discuss in Section 4. For conservative drift VESDE, as we discuss in the above section, it can reduce the influence of the dataset information. Figure 1 (c) supports our augmentation and shows that the density of the generated distribution is more uniform compared to pure VESDE, which means that the drift VESDE can deal with large dataset mean and variance.

Beyond the synthetic data, we show that our conservative drifted VESDE can improve the generated images of pure VESDE without training on the real-world CelebA256 dataset. From the qualitative perspective, as shown in Figure 3, the images generated by our drifted VESDE have more detail (such as hair and beard details). On the contrary, since pure VESDE can not deal with large variance, the images generated by pure VESDE appear blurry and unrealistic in these details. From the quantitative results, we use aesthetic score [Schuhmann et al., 2022] and Inception Score to measure the quality of generated images. Our drifted VESDE achieves aesthetic score 5.813, and IS 4.174, which is better than the results of baseline pure VESDE (aesthetic score 5.807 and IS 4.082). There are more examples on CelebA256 and more experiments on Swiss roll and 1D-GMM to explore different sampling methods (RK45, PFODE) and different $T$. We refer to Appendix G for more details.

## 8 Conclusion

In this work, we analyze the VE-based models under the manifold hypothesis. Firstly, we propose a new forward drifted VESDE process, which enjoys a faster forward convergence rate. Then, we show that with an aggressive $\beta_t$, the new process balances different errors and achieve the first efficient polynomial sample complexity for a series of VE-based models with reverse SDE.

After achieving the above results, we go beyond the reverse SDE and propose the tangent-based unified framework, which considers reverse SDE and PFODE at the same time. Under this framework, we make use of the variance exploding property of VESDE and achieve the first quantitative convergence guarantee for SOTA VE-based models with reverse PFODE. Finally, we show the power of the new drifted forward process through synthetic and real-world experiments.

**Future Work and Limitation.** This work proposes the first unified framework for VE-based models with an accurate score. After that, we plan to consider the approximated score error and provide a polynomial complexity for the VE-based models with reverse PFODE under the manifold hypothesis.

**Broader Impact.** Our work focuses on the convergence guarantee of the SOTA diffusion models and deepens the understanding of diffusion models. Therefore, this work can be viewed as a fundamental step in improving the quality of diffusion models and the societal impact is similar to general generative models [Mirsky and Lee, 2021].

## Acknowledgments and Disclosure of Funding

The author Bo Jiang is supported by National Natural Science Foundation of China (62072302).

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

# Appendix

## A    More Discussion on Drifted VESDE and Current Works

### A.1    The Drifted VESDE is Representative Enough.

In this work, we consider $\tau \in [T, T^2]$ and show that this choice is enough to represent the current VESDE. More specifically, since VESDE with $\sigma_t^2 = t$ has $q_T = \mathcal{N}(\mathbb{E}[q_0], \text{Cov}[q_0] + T\mathbf{I})$ and drifted VESDE with $\tau = T^2$ and $\beta_t \equiv 1/2$ has $q_T^\tau = \mathcal{N}(\exp(-\frac{1}{2T})\mathbb{E}[q_0], \exp(-\frac{1}{T})\text{Cov}[q_0] + (1 - \exp(-\frac{1}{T}))T^2\mathbf{I})$, these two setting is almost identical when $T \to +\infty$. The second choice of VESDE $\sigma_t^2 = t^2$, which achieves the state-of-the-art performance [Karras et al., 2022], is almost identical to $\tau = T^2$, $\beta_t = t$. The simulation experiments also show that VESDE and drifted VESDE with specific $\beta_t$ and $\tau$ have similar performance (Figure 2).

### A.2    The Detailed Calculation of Previous work

**The results of Lee et al. [2022]**    Lee et al. [2022] consider VESDE ($\sigma_t^2 = t$) with reverse SDE under the LSI assumption with parameter $C_{\text{LS}}$. The LSI assumption does not allow the presence of substantial non-convexity and is far away from the multi-modal real-world distribution. Furthermore, they use unrealistic assumption $\epsilon_{\text{score}} \leq 1/(C_{\text{LS}} + T)$ to avoid the effect of the approximated score, which is stronger than Assumption 5.1. Under the above strong assumption, Lee et al. [2022] achieve the polynomial sample complexity $\tilde{O}(L^2 d(d|c| + R)^2/\epsilon_{\text{TV}}^4)$. Under the manifold hypothesis, by Lemma E.2, we know that

$$L = R^2 d^2/\epsilon_{W_2}^4$$

Then, the result is $\tilde{O}(R^4 d^5(d|c| + R)^2/\epsilon_{W_2}^8 \epsilon_{\text{TV}}^4)$, which is worse than Corollary 5.4.

**The results of Gao et al. [2023].**    Gao et al. [2023] analyze a series of VESDE with reverse SDE and achieve $1/\epsilon_{W_2}^{2.5}$ sample compelxity for VESDE with $\sigma_t^2 = t^2$ in 2-Wasserstein distance. However, they assume the data distribution is log-concave, which is even stronger than LSI assumption. Furthermore, under this assumption, $\nabla \log q_{T-t}(\cdot)$ do not blow-up at the end of the reverse process, which do not match the empirical phenomenon and ignore the influence of early stopping parameters. To transfer their results to our the results under the manifold hypothesis, we need to consider the influence of $\delta$, which would introduce an additional $1/\text{Poly}(W_2)$ term.

## B    The Proof for the Faster Forward Process

**Lemma B.1.** *The minimization problem* $\min_{\bar{m}_t, V_t} \text{KL}\left(q_t^\tau \mid \mathcal{N}(\bar{m}_t, V_t)\right)$ *is minimized by* $\bar{m}_t = m_t \mathbb{E}[q_0]$ *and* $V_t = m_t^2 \text{Cov}[q_0] + \sigma_t^2 \mathbf{I}$, *where* $m_t$ *and* $\sigma_t$ *defined in Equation* (4).

**Proof.** For simplicity, we denote the mean and covariance of $q_0$ by $a$ and $C'$. We also define the optimize variable $n_t = \mathcal{N}(\bar{m}_t, C_t)$. We can directly compute the KL divergence $\text{KL}(q_t|n_t)$:

$$\text{KL}(q_t|n_t) = -H(q_t) - \int \log(n_t(x)) q_t(x) \mathrm{d}x$$

$$= -H(q_t) + \frac{d}{2}\log(2\pi) + \frac{1}{2}\log(\det(V_t)) + \frac{1}{2}\int (x - \bar{m}_t)^T V_t^{-1}(x - \bar{m}_t) q_t(x)\mathrm{d}x.$$

For the last term, we directly compute

$$\int (x - \bar{m}_t)^T V_t^{-1}(x - \bar{m}_t) p_t(x)\mathrm{d}x$$

$$= \mathbb{E}\left[(X_t - \bar{m}_t)^T V_t^{-1}(X_t - \bar{m}_t)\right] = \mathbb{E}\left[(m_t X_0 + \sigma_t Z - \bar{m}_t)^T V_t^{-1}(m_t X_0 + \sigma_t Z - \bar{m}_t)\right]$$

$$= \mathbb{E}\left[m_t^2(X_0 - a)^T V_t^{-1}(X_0 - a)\right] + (m_t a - \bar{m}_t)^T V_t^{-1}(m_t a - \bar{m}_t) + \sigma_t^2 \mathbb{E}\left[Z^T V_t^{-1} Z\right]$$

$$= m_t^2 \text{tr}\left(C' V_t^{-1}\right) + \sigma_t^2 \text{tr}\left(V_t^{-1}\right) + (m_t a - \bar{m}_t)^T V_t^{-1}(m_t a - \bar{m}_t),$$

where the second inequality follows that $X_t = m_t X_0 + \sigma_t Z$. It is clear that the optimal solution of $\bar{m}_t$ is $m_t a$. In the next step, we focus on the optimization problem for $V_t$:

$$L\left(V_t^{-1}\right) = \log\left(\det\left(V_t\right)\right) + \text{tr}\left(\left(m_t^2 C' + \sigma_t^2 \mathbf{I}\right) V_t^{-1}\right)$$
$$= -\log\left(\det\left(V_t^{-1}\right)\right) + \text{tr}\left(\left(m_t^2 C' + \sigma_t^2 \mathbf{I}\right) V_t^{-1}\right).$$

Since the above optimization is a convex optimization problem, we use the method similar to Pidstrigach [2022], we obtain that the optimal solution of $V_t$ is $m_t^2 C' + \sigma_t^2 \mathbf{I}$. $\blacksquare$

**Lemma B.2.** *Let $\bar{m}_t$ and $V_t$ be the optimal mean and covariance operator from Lemma B.1. Then*

$$\text{KL}\left(q_t | \mathcal{N}(\bar{m}_t, V_t)\right) \leq \frac{1}{2}\log\left(\frac{\prod_{i=1}^d \left(m_t^2 c_i + \sigma_t^2\right)}{(\sigma_t^2)^d}\right) + \frac{R^2 m_t}{\sigma_t^2}$$

$$\leq \frac{d m_t^2 c}{2\sigma_t^2} + \frac{R^2 m_t}{\sigma_t^2} + o(\frac{m_t^2 c}{\sigma_t^2}),$$

$$\text{KL}\left(\mathcal{N}(\bar{m}_t, V_t) | (\mathcal{N}(0, \sigma_t^2))\right) \leq \frac{m_t^2 \sum_{i=1}^d c_i}{2\sigma_t^2} + \frac{m_t^2 (\mathbb{E}[q_0])^2}{2\sigma_t^2} + \frac{1}{2}\log\left(\frac{(\sigma_t^2)^d}{\prod_{i=1}^d \left(m_t^2 c_i + \sigma_t^2\right)}\right)$$

$$\leq \frac{m_t^2 \sum_{i=1}^d c_i}{2\sigma_t^2} + \frac{m_t^2 (\mathbb{E}[q_0])^2}{2\sigma_t^2} + \frac{d m_t^2 c}{2\sigma_t^2} + o(\frac{m_t^2 c}{\sigma_t^2}),$$

*where $c_i$ are the eigenvalues of $\text{Cov}\,[q_0]$, and $c$ is the eigenvalue with the largest absolute value.*

**Proof.** For $t \geq 0$, we directly calculate the KL divergence for this term:

$$\text{KL}\left(q_t \mid \mathcal{N}(\bar{m}_t, V_t)\right) = -H\left(q_t\right) + \frac{1}{2}\log\left(\det\left(2\pi V_t\right)\right) + \frac{1}{2}\text{tr}\left(\left(m_t^2 C' + \sigma_t^2 \mathbf{I}\right) V_t^{-1}\right)$$

$$= -H\left(q_t\right) + \frac{1}{2}\log\left(\det\left(2\pi V_t\right)\right) + \frac{d}{2}$$

$$= -H\left(q_t\right) + \frac{d}{2}\log(2\pi) + \frac{1}{2}\log\left(\prod_{i=1}^d \left(m_t^2 c_i + \sigma_t^2\right)\right) + \frac{d}{2},$$

where $c_i$ are the eigenvalues of $\text{Cov}\,[q_0]$. Now, we only need to calculate $H(q_t)$:

$$-H\left(q_t\right) = \mathbb{E}_{X_t}\left[\log q_t\left(X_t\right)\right] = \mathbb{E}_{X_t}\left[\log\left(\mathbb{E}_{X_0}\left[(2\pi\sigma_t^2)^{-d/2}\exp\left(-\frac{1}{2\sigma_t^2}\|X_t - X_0\|^2\right)\right]\right)\right].$$

By Assumption 4.1, it is clear that

$$\exp\left(-\frac{1}{2\sigma_t^2}\|X_t - X_0\|^2\right) \leq \exp\left(-\frac{1}{2\sigma_t^2}\left(\|X_t\|^2 + 2\langle X_t, X_0\rangle\right)\right).$$

Then, we know that

$$\mathbb{E}\left[\log\left(\mathbb{E}_{X_0}\left[(2\pi\sigma_t^2)^{-d/2}\exp\left(-\frac{1}{2\sigma_t^2}\|X_t - X_0\|^2\right)\right]\right)\right]$$

$$\leq \mathbb{E}\left[\log\left((2\pi\sigma_t^2)^{-d/2}\right) - \frac{1}{2\sigma_t^2}\left(\|X_t\|^2 + 2\langle X_t, X_0\rangle\right)\right]$$

$$\leq -\frac{d}{2}\log(2\pi) - \frac{1}{2}\log\left((\sigma_t^2)^d\right) - \frac{1}{2\sigma_t^2}\mathbb{E}\left[\|X_t\|^2\right] + \frac{R^2 m_t}{\sigma_t^2}.$$

we also know that

$$\mathbb{E}\left[\|X_t\|^2\right] = m_t^2\mathbb{E}\left[\|X_0\|^2\right] + \sigma_t^2\mathbb{E}\left[\|Z\|^2\right] = \mathbb{E}\left[\|X_0\|^2\right] + t\mathbb{E}\left[\|Z\|^2\right] = \bar{m}_0^2 + V_0 + \sigma_t^2 d.$$

Finally, put these terms together, we have:

$$\text{KL}\left(q_t | \mathcal{N}(\bar{m}_t, V_t)\right) \leq \frac{1}{2}\log\left(\frac{\prod_{i=1}^d \left(m_t^2 c_i + \sigma_t^2\right)}{(\sigma_t^2)^d}\right) + \frac{R^2 m_t}{\sigma_t^2},$$

where $c_i$ are the eigenvalues of $\text{Cov}\,[q_0]$. Then by choosing the largest absolute value eigenvalue largest absolute value, we can use the Taylor expansion to obtain the first results of this lemma. For the second result of this lemma, we directly compute the KL divergence between $\mathcal{N}(\bar{m}_t, V_t)$ and $\mathcal{N}(0, \sigma_t^2)$ to obtain the final results. $\blacksquare$

**Theorem 4.2.** *Assume Assumption 4.1 and 3.1. Let $q_\infty^\tau = \mathcal{N}(0, \sigma_T^2 \mathbf{I})$. With $m_T, \sigma_T$ defined in Equation (4), we have $\mathrm{TV}(q_T^\tau, q_\infty^\tau) \leq \sqrt{m_T} \bar{D} / \sigma_T$, where $\bar{D} = d|c| + \mathbb{E}[q_0] + R$ and $c$ is the eigenvalue of $\mathrm{Cov}[q_0]$ with the largest absolute value.*

**Proof.** We know that

$$
\|q_T - q_\infty^\tau\|_{\mathrm{TV}}
$$
$$
\leq \|q_T - \mathcal{N}(m_T \mathbb{E}[q_0], m_T^2 \mathrm{Cov}[q_0] + \sigma_T^2 \mathbf{I})\|_{\mathrm{TV}} + \|\mathcal{N}(m_T \mathbb{E}[q_0], m_T^2 \mathrm{Cov}[q_0] + \sigma_T^2 \mathbf{I}) - q_\infty^\tau\|_{\mathrm{TV}} .
$$

By directly using the Pinsker's inequality and Lemma B.2, we complete the proof. ∎

## C The Proof of the Polynomial Complexity for Reverse SDE

In this section, we prove Corollary 5.4. First, we recall the Girsanov's Theorem [Le Gall, 2016] used in Chen et al. [2023c]:

**Lemma C.1** (Girsanov's theorem). *Let $P_T$ and $Q_T$ be two probability measures on path space $\mathcal{C}\left([0, T]; \mathbb{R}^d\right)$. Suppose that under $P_T$, the process $(X_t)_{t \in [0,T]}$ follows*

$$
\mathrm{d}X_t = \tilde{b}_t \, \mathrm{d}t + \alpha_t \, \mathrm{d}\tilde{B}_t
$$

*where $\tilde{B}$ is a $P_T$-Brownian motion, and under $Q_T$, the process $(X_t)_{t \in [0,T]}$ follows*

$$
\mathrm{d}X_t = b_t \, \mathrm{d}t + \alpha_t \, \mathrm{d}B_t
$$

*where $B$ is a $Q_T$-Brownian motion. We assume that for each $t > 0$, $\alpha_t$ is a $d \times d$ symmetric positive definite matrix. Then, provided that Novikov's condition holds,*

$$
\mathbb{E}_{Q_T} \exp\left(\frac{1}{2} \int_0^T \left\|\alpha_t^{-1}\left(\tilde{b}_t - b_t\right)\right\|^2 \, \mathrm{d}t\right) < \infty,
$$

*we have that*

$$
\frac{\mathrm{d}P_T}{\mathrm{d}Q_T} = \exp\left(\int_0^T \alpha_t^{-1}\left(\tilde{b}_t - b_t\right) \mathrm{d}B_t - \frac{1}{2} \int_0^T \left\|\alpha_t^{-1}\left(\tilde{b}_t - b_t\right)\right\|^2 \, \mathrm{d}t\right) .
$$

If the Novikov's condition is satisfied, we apply the Girsanov theorem by choosing $P_T = R_K^{q_T^\tau}, Q_T = Q_{t_K}^{q_T^\tau}, \tilde{b}_t = \beta_{T-t}\left\{\frac{1}{\tau}\widetilde{\mathbf{Y}}_t + 2\mathbf{s}(T - t_k, \widetilde{\mathbf{Y}}_t)\right\}$ (for $t \in [t_k, t_{k+1}]$), $b_t = \beta_{T-t}\left\{\frac{1}{\tau}\mathbf{Y}_t + \left(1 + \eta^2\right)\nabla \log q_{T-t}(\mathbf{Y}_t)\right\}$, and $\alpha_t = \sqrt{2\beta_{T-t}}\mathbf{I}_d$.

Then, similar to Chen et al. [2023c], we have the following lemma.

**Lemma C.2.** *Assuming that $R_K^{q_T^\tau}$ and $Q_{t_K}^{q_T^\tau}$ satisfy Novikov's condition, it holds that*

$$
\mathrm{KL}\left(Q_{t_K}^{q_T^\tau} \| R_K^{q_T^\tau}\right)
$$
$$
= \mathbb{E}_{Q_{t_K}^{q_T^\tau}} \ln \frac{\mathrm{d}Q_{t_K}^{q_T^\tau}}{\mathrm{d}R_K^{q_T^\tau}} = \sum_{k=0}^{K-1} \mathbb{E}_{Q_{t_K}^{q_T^\tau}} \int_{t_k}^{t_{k+1}} 2\beta_{T-t} \|\mathbf{s}(T - t_k, \mathbf{Y}_{t_k}) - \nabla \ln q_{T-t}(\mathbf{Y}_t)\|^2 \, \mathrm{d}t .
$$

Before using the Girsanov's Theorem, we need to check the Novikov's condition. We use almost the same proof process compare to Chen et al. [2023c]. The key proof of the Novikov's condition is Lemma 19 of Chen et al. [2023c]. Hence, we give a complete proof of Lemma 19 in Chen et al. [2023c] under our drifted VESDE. Before the proof, we first introduce a smooth cuttoff function for truncating the drift terms.

**Lemma C.3** (lemma 17 of Chen et al. [2023c]). *For any $\bar{R} > 0$, there is a smooth function $\phi_R : \mathbb{R}^d \to [0, 1]$ satisfying: 1. $\phi_{\bar{R}}(x) = 1$ for all $\|x\| \leq \bar{R}$, 2. $\phi_{\bar{R}}(x) = 0$ for all $\|x\| \geq 2\bar{R}$, 3. $\phi_{\bar{R}}$ is $O(1/\bar{R})$-Lipschitz.*

Note that $\bar{R}$ is not $R$ in Assumption 4.1 and will goes to $+\infty$ in the proof of Chen et al. [2023c]. Similar to Chen et al. [2023c], we also introduce a $L_\infty$ and a modified process with truncation argument for $t \in [t_k, t_{k+1}]$. Define the bad set

$$B_t := \{\|s_t - \nabla \ln q_t\| \geq \varepsilon_{\text{score},\infty}\},$$

where $\varepsilon_{\text{score},\infty} > 0$ is a parameter to be chosen later. We define the $L^\infty$-accurate score estimate to be

$$s_t^\infty := s_t \mathbf{1}_{B_t^c} + \nabla \ln q_t \mathbf{1}_{B_t}.$$

We note that $\|s_t^\infty - \nabla \ln q_t\| \leq \varepsilon_{\text{score},\infty}$.

The modified process with truncation argument for $t \in [t_k, t_{k+1}]$ is

$$d\mathbf{Y}_t^\infty = \beta_{T-t}\left\{\mathbf{Y}_t^\infty/\tau + 2\nabla \log q_{T-t}^\tau(\mathbf{Y}_t^\infty)\right\} dt + \sqrt{2\beta_{T-t}}\, d\mathbf{B}_t,$$
$$d\widetilde{\mathbf{Y}}_t^\infty = \beta_{T-t}\left\{\widetilde{\mathbf{Y}}_t^\infty/\tau + 2\mathbf{s}(T-t_k, \widetilde{\mathbf{Y}}_t^\infty)\right\} dt + \sqrt{2\beta_{T-t}}\, d\mathbf{B}_t,$$

where $\mathbf{Y}_0^\infty = \widetilde{\mathbf{Y}}_0^\infty$ is obtained by sampling $\mathbf{Y}_0^\infty \sim q_T^\tau$ and setting $\widetilde{\mathbf{Y}}_0^\infty = \mathbf{X}_T$ if $\|\mathbf{X}_T\| \leq \bar{R}$ and setting $\widetilde{\mathbf{Y}}_0^\infty = 0$ otherwise. Then, we ara ready to prove the following lemma.

**Lemma C.4** (Modified key lemma for Novikov's condition).

$$\mathbb{E}_{Q_{t_K}^{q_T^\tau,\infty}} \exp\left(\sum_{k=0}^{K-1} \int_{t_k}^{t_{k+1}} \beta_{T-t_k} \left\|\phi_{\bar{R}}\left(\mathbf{Y}_{T-t_k}^\infty\right) \mathbf{s}_{T-t_k}^\infty\left(\mathbf{Y}_{T-t_k}^\infty\right) - \phi_{\bar{R}}\left(\mathbf{Y}_{T-t_k}^\infty\right) \nabla \ln q_{T-t}\left(\mathbf{Y}_{T-t_k}^\infty\right)\right\|^2 dt\right) < \infty$$

**Proof.** We note that due to the manifold hypothesis (Assumption 4.1), if $\bar{R} \geq \sqrt{dT^2 + R^2}$, then the marginal distribution of $\mathbf{Y}_{T-t_k}^\infty$ is exactly the same compared to $\mathbf{X}_{t_k}$. We also recall that $\bar{R} \to +\infty$ in Chen et al. [2023c] (Theorem 21 of Chen et al. [2023c]). Hence, we can use Lemma E.1 to prove that

$$\left\|\sqrt{\beta_{T-t_k}}\phi_{\bar{R}}\left(\mathbf{Y}_{T-t_k}^\infty\right) \mathbf{s}_{T-t_k}^\infty\left(\mathbf{Y}_{T-t_k}^\infty\right)\right\| \leq \sup_{t^* \in [0, T-\delta]} \sqrt{\beta_{T-t^*}} \left\|\mathbf{s}_{T-t^*}^\infty\left(\mathbf{Y}_{T-t^*}\right)\right\| =: A' < \infty.$$

and

$$\left\|\sqrt{\beta_{T-t_k}}\phi_{\bar{R}}\left(\mathbf{Y}_{T-t_k}^\infty\right) \nabla \ln q_{T-t}\left(\mathbf{Y}_{T-t_k}^\infty\right)\right\| \leq \sup_{t^* \in [0, T-\delta]} \sqrt{\beta_{T-t^*}} \left\|\nabla \ln q_{T-t^*}\left(\mathbf{Y}_{T-t^\star}\right)\right\| =: B' < \infty.$$

Then, the left hand of this lemma is at most $\exp\left(2T\left(A'^2 + B'^2\right)\right) < \infty$ as claimed. ∎

After obtaining the above inequality, the remaining proof for Novikov's condition are exactly compared to Chen et al. [2023c].

Since we assume the accurate score function in this work, this lemma need to control

$$\sup_{x^* \in \mathbb{B}(0,R), t^* \in [0, T-\delta]} 2\beta_{T-t^*} \|\nabla \ln q_{T-t^*}(x^*)\| =: B < \infty.$$

As we shown in Lemma E.1, we know that with the early stopping parameter $\delta$, $\|\nabla \ln q_{T-t^*}(x^*)\|$ is controlled. By using Assumption 4.1, we know that $\frac{1}{\beta_{T-t^*}} \leq \bar{\beta}$. Finally, with similar process to Chen et al. [2023c], we can proof that the Novikov's condition is satisfied. The following lemma show the discretization error for our drifted VESDE with reverse SDE.

**Lemma C.5** (Discretization). *Suppose that Assumption 4.1 and Assumption 5.1 holds. Let $\bar{\gamma}_K = \arg\max_{k \in \{0,\ldots,K-1\}} \gamma_k, \gamma_K = \delta$. If $\tau \in [1, T^2]$ and $\beta_t \in [1, t^2]$, then with $Q_{t_K}^{q_T^\tau}$ and $R_K^{q_T^\tau}$ defined in Lemma C.2,*

$$\text{TV}\left(R_K^{q_T^\tau}, Q_{t_K}^{q_T^\tau}\right)^2 \lesssim \frac{R^4 T\tau\beta_T d}{\sigma_\delta^8}\bar{\gamma}_K + \frac{R^6 T\tau\beta_T}{\sigma_\delta^8}\bar{\gamma}_K^2 + \epsilon_{score}^2 T\beta_T.$$

**Proof.** First, we control the discretization error in an interval $t \in [t_k, t_{k+1}]$:

$$\mathbb{E}_{Q_{t_K}^{q_T^\tau}} \left[ \| \mathbf{s}\left(T - t_k, \mathbf{Y}_{t_k}\right) - \nabla \ln q_{T-t}\left(\mathbf{Y}_t\right) \|^2 \right]$$

$$\lesssim \epsilon_{\text{score}}^2 + \mathbb{E}_{Q_{t_K}^{q_T^\tau}} \left[ \| \nabla \ln q_{T-t_k}\left(\mathbf{Y}_{t_k}\right) - \nabla \ln q_{T-t}\left(\mathbf{Y}_{t_k}\right) \|^2 \right]$$

$$+ \mathbb{E}_{Q_{t_K}^{q_T^\tau}} \left[ \| \nabla \ln q_{T-t}\left(\mathbf{Y}_{t_k}\right) - \nabla \ln q_{T-t}\left(\mathbf{Y}_t\right) \|^2 \right]$$

$$\lesssim \mathbb{E}_{Q_{t_K}^{q_T^\tau}} \left[ \left\| \nabla \ln \frac{q_{T-t_k}}{q_{T-t}}\left(\mathbf{Y}_{t_k}\right) \right\|^2 \right] + L^2 \mathbb{E}_{Q_{t_K}^{q_T^\tau}} \left[ \| \mathbf{Y}_{t_k} - \mathbf{Y}_t \|^2 \right] + \epsilon_{\text{score}}^2$$

$$\lesssim \tau L^2 d\bar{\gamma}_K + \tau L^2 \bar{\gamma}_K^2 \left( d\tau + R^2 \right) + \tau L^3 \bar{\gamma}_K^2 + L^2 (\beta_T d\bar{\gamma}_K + R^2 \bar{\gamma}_K^2) + \epsilon_{\text{score}}^2$$

$$\lesssim \tau L^2 d\bar{\gamma}_K + \tau L^2 R^2 \bar{\gamma}_K^2 + \epsilon_{\text{score}}^2 ,$$

where $L = \max_{t \in [0, T-\delta]} \| \nabla^2 \log q_{T-t}\left(\mathbf{Y}_t\right) \| \leq \left( 1 + R^2 \right) / \sigma_\delta^4$ and the third inequality follows Lemma E.5. Then, we know that

$$\sum_{k=0}^{K-1} \mathbb{E}_{Q_{t_K}^{q_T^\tau}} \int_{t_k}^{t_{k+1}} 2\beta_{T-t} \| s\left(T - t_k, \mathbf{Y}_{t_k}\right) - \nabla \ln q_{T-t}\left(\mathbf{Y}_t\right) \|^2 \, dt$$

$$\lesssim \tau T \beta_T L^2 d\bar{\gamma}_K + L^2 R^2 \tau T \beta_T \bar{\gamma}_K^2 + \epsilon_{\text{score}}^2 T \beta_T$$

$$\lesssim \frac{R^4 \tau T \beta_T d}{\sigma_\delta^8} \bar{\gamma}_K + \frac{R^6 \tau T \beta_T}{\sigma_\delta^8} \bar{\gamma}_K^2 + \epsilon_{\text{score}}^2 T \beta_T .$$

After obtaining the general discretization error for our drifted VESDE, we focus on two special cases. For $\tau = T^2$ and $\beta_t = t^2$, we have that

$$\sum_{k=0}^{K-1} \mathbb{E}_{Q_{t_K}^{q_T^\tau}} \int_{t_k}^{t_{k+1}} 2\beta_{T-t} \| s\left(T - t_k, \mathbf{Y}_{t_k}\right) - \nabla \ln q_{T-t}\left(\mathbf{Y}_t\right) \|^2 \, dt$$

$$\lesssim T^3 \beta_T L^2 d\bar{\gamma}_K + L^2 R^2 T^3 \beta_T \bar{\gamma}_K^2 + \epsilon_{\text{score}}^2 T \beta_T$$

$$\lesssim \frac{R^4 T^3 \beta_T d}{\sigma_\delta^8} \bar{\gamma}_K + \frac{R^6 T^3 \beta_T}{\sigma_\delta^8} \bar{\gamma}_K^2 + \epsilon_{\text{score}}^2 T \beta_T$$

$$= \frac{R^4 T^5 d}{\sigma_\delta^8} \bar{\gamma}_K + \frac{R^6 T^5}{\sigma_\delta^8} \bar{\gamma}_K^2 + \epsilon_{\text{score}}^2 T^3 .$$

For $\tau = T$ and $\beta_t = t$, we know that

$$\sum_{k=0}^{K-1} \mathbb{E}_{Q_{t_K}^{q_T^\tau}} \int_{t_k}^{t_{k+1}} 2\beta_{T-t} \| s\left(T - t_k, \mathbf{Y}_{t_k}\right) - \nabla \ln q_{T-t}\left(\mathbf{Y}_t\right) \|^2 \, dt$$

$$\lesssim T^3 L^2 d\bar{\gamma}_K + L^2 R^2 T^3 \bar{\gamma}_K^2 + \epsilon_{\text{score}}^2 T^2$$

$$\lesssim \frac{R^4 T^3 d}{\sigma_\delta^8} \bar{\gamma}_K + \frac{R^6 T^3}{\sigma_\delta^8} \bar{\gamma}_K^2 + \epsilon_{\text{score}}^2 T^2 .$$

∎

Combined with the reversing beginning error controlled by Theorem 4.2, we can obtain the convergence guarantee for the general drifted VESDE with reverse SDE.

**Theorem 5.2.** *Assume Assumption 3.1, 4.1, 5.1. Let $\bar{D}$ defined in Theorem 4.2, $\bar{\gamma}_K = \text{argmax}_{k \in \{0, \ldots, K-1\}} \gamma_k$, $\tau = T^2$ and $\beta_t \in [1, t^2]$. Then, we have that*

$$\text{TV}\left( R_K^{q_\infty^\tau}, q_\delta \right) \leq \frac{\bar{D}\sqrt{m_T}}{\sigma_T} + \frac{R^2 \sqrt{d}}{\sigma_\delta^4} \sqrt{\bar{\gamma}_K \beta_T \tau T} + \epsilon_{\text{score}} \sqrt{\beta_T T} .$$

**Proof.** By the data processing inequality, we know that

$$\text{TV}\left(R_K^{q_\infty^\tau}, q_\delta\right) \le \text{TV}\left(R_K^{q_\infty^\tau}, R_K^{q_T^\tau}\right) + \text{TV}\left(R_K^{q_T^\tau}, Q_{t_K}^{q_T^\tau}\right)$$

$$\le \text{TV}\left(q_T^\tau, q_\infty^\tau\right) + \text{TV}\left(R_K^{q_\infty^\tau}, Q_{t_K}^{q_T^\tau}\right).$$

Combined with Theorem 4.2 and Lemma C.5, we achieve the final result. ∎

## C.1 The Sample Complexity for Drifted VESDE

As shown in Theorem 5.2, the general convergence guarantee is

$$\frac{\bar{D}\sqrt{m_T}}{\sigma_T} + \frac{R^2\sqrt{d}}{\sigma_\delta^4}\sqrt{\bar{\gamma}_K \beta_T \tau T} + \epsilon_{\text{score}}\sqrt{\beta_T T}.$$

In this section, we provide the sample complexity for different $\beta_t$ and $\tau$.

**The results of $\tau = 1$ and $\beta_t = 1$.** When considering $\beta_t = 1$ and $\tau = 1$, the drifted VESDE becomes VPSDE and $m_T = \exp\left(-T\right)$, which indicates $T$ is a logarithmic term and the dominated term of the convergence guarantee is the discretization term $\widetilde{O}(R^2\sqrt{d\bar{\gamma}_K}/\sigma_\delta^4)$. To make this term smaller than $\epsilon_{\text{TV}}$, we require $\bar{\gamma}_K \le \sigma_\delta^8 \epsilon_{\text{TV}}^2/(R^4 d)$. To make sure that $W_2^2\left(q_0, q_\delta\right) \le \epsilon_{W_2}^2$, we require $\sigma_\delta^2 \le \frac{\epsilon_{W_2}^2}{d + R\sqrt{d}}$. Then, we achieve the final sample complexity

$$K \le \widetilde{O}\left(\frac{dR^4(d + R\sqrt{d})^4}{\epsilon_{W_2}^8 \epsilon_{\text{TV}}^2}\right).$$

We note that this results is exactly the same with Chen et al. [2023c], which means our drifted VESDE covers VPSDE setting.

### C.1.1 The results of $\tau = T^2$ with different $\beta_t$

In this part, we analyze the influence of $\beta_t$ under setting $\tau = T^2$ and show the power of our drifted VESDE.

**The aggressive $\beta_t$ ($\tau = T^2$).** When considering aggressive $\beta_t = t^{\alpha_1}$ where $2 \ge \alpha_1 \ge 1 + \ln(T - \ln(T))/\ln(T)$, $\sqrt{m_T}/\sigma_T \ge \exp\left(-T/2\right)$, which means $T$ is a logarithmic term and can be ignored. After that, the analysis process is exactly the same with the above VPSDE setting, and we achieve the sample complexity

$$K \le \widetilde{O}\left(\frac{dR^4(d + R\sqrt{d})^4}{\epsilon_{W_2}^8 \epsilon_{\text{TV}}^2}\right).$$

Defined by $R_{K,R_0}^{q_\infty^\tau}$ the output $R_K^{q_\infty^\tau}$ projected onto $\text{B}\left(0, R_0\right)$ for $R_0 = \widetilde{\Theta}(R)$. Following exactly the same proof process of Chen et al. [2023c] (Corollary 5), we have that

$$W_2(R_{K,R_0}^{q_\infty^\tau}, q_0) \le \epsilon_{W_2}$$

with sample complexity

$$K \le \widetilde{O}\left(\frac{dR^8(d + R\sqrt{d})^4}{\epsilon_{W_2}^{12}}\right).$$

**The conservative $\beta_t = t$.** In this case, the first term is $\bar{D}/T$. To make this term smaller than $\epsilon_{\text{TV}}$, we require $T \ge \bar{D}/\epsilon_{\text{TV}}$. For the stepsize, we require $\bar{\gamma}_K \le \sigma_\delta^8 \epsilon_{\text{TV}}^2/(R^4 d T^4)$, which means the sample complexity is

$$K \le O\left(\frac{dR^4(d + R\sqrt{d})^4 \bar{D}^5}{\epsilon_{W_2}^8 \epsilon_{\text{TV}}^7}\right).$$

**The most conservative $\beta_t = 1$.** In this case, $m_T = \exp(-1/T)$ and $\sigma_t^2 = \tau(1 - m_t^2) = T^2(1 - \exp(-2/T))$. When $T$ is large enough, $m_T = \Theta(1)$ and $\sigma_T^2 = T$, which indicates the first term is $\bar{D}/\sqrt{T}$. To make this term smaller than $\epsilon_{\mathrm{TV}}$, we require $T \geq \bar{D}^2/\epsilon_{\mathrm{TV}}^2$. For the second term, we require $\bar{\gamma}_K \leq \sigma_\delta^8 \epsilon_{\mathrm{TV}}^2/(R^4 d T^3)$ Then, the final complexity is

$$K \leq O\left(\frac{dR^4(d + R\sqrt{d})^4 \bar{D}^8}{\epsilon_{W_2}^8 \epsilon_{\mathrm{TV}}^{10}}\right).$$

### C.1.2 The results of $\tau = T$ with different $\beta_t$

At the remaining part, we show the sample complexity of setting $\tau = T$ with different $\beta_t$.

**The results for setting $\tau = T$ and $\beta_t = t$.** For this setting, as shown in Lemma C.5, the discretization error is

$$\mathrm{TV}\left(R_K^{q_T^\tau}, Q_{t_K}^{q_T^\tau}\right)^2 \lesssim \frac{R^4 T^3 d}{\sigma_\delta^8}\bar{\gamma}_K + \frac{R^6 T^3}{\sigma_\delta^8}\bar{\gamma}_K^2 + \epsilon_{\mathrm{score}}^2 T^2.$$

Furthermore, we choose an aggressive $\beta_t$, which indicates $T$ is a logarithmic term. Then, by choosing $\sigma_\delta^2 \leq \frac{\epsilon_{W_2}^2}{(d+R\sqrt{d})}$ and $\bar{\gamma}_K \leq \sigma_\delta^8 \epsilon_{\mathrm{TV}}^2 \ln^3\left(\bar{D}/\epsilon_{\mathrm{TV}}\right)/\left(R^4 d\right)$, we obtain the sample complexity

$$K = \frac{T}{\bar{\gamma}_K} \leq \widetilde{O}\left(\frac{dR^4(d + R\sqrt{d})^4}{\epsilon_{W_2}^8 \epsilon_{\mathrm{TV}}^2}\right).$$

**The results for setting $\tau = T$ and $\beta_t = 1$.** In this setting, the reverse beginning error is bounded by $\bar{D}/\sqrt{T}$, which indicates $T \geq \bar{D}^2/\epsilon_{\mathrm{TV}}^2$. For the discretization term, we require $\bar{\gamma}_K \leq \sigma_\delta^8 \epsilon_{\mathrm{TV}}^2/(R^4 d T^2)$. Then, the sample complexity is bounde by

$$K = \frac{T}{\bar{\gamma}_K} \leq O\left(\frac{dR^4(d + R\sqrt{d})^4 \bar{D}^6}{\epsilon_{W_2}^8 \epsilon_{\mathrm{TV}}^8}\right).$$

## D The Proof of the Convergence Guarantee in the Unified Framework

In this work, we introduce an indicator $i \in \{1, 2\}$ for $\sigma_{T-t_K}$ to represent different $\beta_t$. We use $\tau = T^2$ as an example. When $\beta_t = t^2$ is aggressive, we choose $i = 1$, $\eta = 1$ and $\sigma_{T-t_K}^{-2}(i = 1) \leq \frac{1}{\tau} + \frac{\bar{\beta}}{\delta^3}$. When $\beta_t = t$ is conservative, we choose $i = 2$, $\eta \in [0, 1)$ and $\sigma_{T-t_K}^{-2}(i = 2) \leq \frac{1}{\tau} + \frac{\bar{\beta}}{\delta^2}$. In the proof process of Lemma 6.3, Lemma D.1, Lemma D.2 and Lemma D.3, we ignore the indicator $i$ since this lemma does not involve the specific value of $\sigma_{T-t_K}^2(i)$. Before the proof of this section, we first recall the stochastic flow of the reverse process for any $x \in \mathbb{R}^d$ and $s, t \in [0, T]$ with $t \geq s$:

$$d\mathbf{Y}_{s,t}^x = \beta_{T-t}\left\{\mathbf{Y}_{s,t}^x/\tau + \left(1 + \eta^2\right)\nabla \log q_{T-t}\left(\mathbf{Y}_{s,t}^x\right)\right\}dt + \eta\sqrt{2\beta_{T-t}}d\mathbf{B}_t, \qquad \mathbf{Y}_{s,s}^x = x,$$

and the interpolation of its discretization for any $k \in \{0, ..., K\}$ and $t \in [s_k, t_{k+1}]$:

$$d\bar{\mathbf{Y}}_{s,t}^x(k) = \beta_{T-t}\left\{\bar{\mathbf{Y}}_{s,t}^x/\tau + \left(1 + \eta^2\right)\mathbf{s}\left(T - s_k, \bar{\mathbf{Y}}_{s,t}^x\right)\right\}dt + \eta\sqrt{2\beta_{T-t}}d\mathbf{B}_t, \qquad \bar{\mathbf{Y}}_{s,s}^x = x,$$

where $s_k = \max(s, t_k)$. To deal with the discretization error, we use the approximation technique used in Bortoli [2022]. Hence, we introduce the tangent process:

$$d\nabla\mathbf{Y}_{s,t}^x = \beta_{T-t}\left\{\mathbf{I}/\tau + \left(1 + \eta^2\right)\nabla^2 \log q_{T-t}(\mathbf{Y}_{s,t}^x)\right\}\nabla\mathbf{Y}_{s,t}^x dt, \qquad \nabla\mathbf{Y}_{s,s}^x = \mathbf{I}.$$

Then, we discuss the interpolation formula, which is used to control the discretization error.

**Proposition 1.** For $s, t \in [0, T)$ with $s < t$, any $k \in \{0, ..., K\}$ and $(\omega_v)_{v \in [s,T]}$, we define that

$$b_u(\omega) = \beta_{T-u}(\frac{1}{\tau}\omega_u + (1 + \eta^2)\nabla \log q_{T-u}(\omega_u)),$$

$$\bar{b}_u(\omega) = \beta_{T-u}(\frac{1}{\tau}\omega_u + (1 + \eta^2)\mathbf{s}(T - s_k, \omega_{s_k})), \quad \Delta b_u(\omega) = b_u(\omega) - \bar{b}_u(\omega),$$

where $s_k = \max(s, t_k)$ and $u \in [s_k, t_{k+1})$. Then, for any $x \in \mathbb{R}^d$, we have that

$$\mathbf{Y}_{s,t}^x - \bar{\mathbf{Y}}_{s,t}^x = \int_s^t \nabla \mathbf{Y}_{u,t}^x \left( \bar{\mathbf{Y}}_{s,u}^x \right)^\top \Delta b_u \left( \left( \bar{\mathbf{Y}}_{s,v}^x \right)_{v \in [s,T]} \right) du,$$

where for any $u \in [0, T)$, there exists a $k \in \{0, ..., K\}$ satisfies $u \in [s_k, t_{k+1})$.

For reverse SDE, the augmentation is similar to Bortoli [2022] (Appendix E). When $\eta = 0$, the stochastic extension of the Alekseev–Gröbner formula [Del Moral and Singh, 2022] degenerates into the original version [Alekseev, 1961]. After that, we control the tangent process.

**Lemma 6.3.** Assume Assumption 3.1 and 4.1. For $\forall s \in [0, t_K]$ and $x \in \mathbb{R}^d$, we have

$$\|\nabla \mathbf{Y}_{s,t_K,i}^x\| \leq \exp \left( \frac{R^2}{2\sigma_{T-t_K}^2} + \frac{(1-\eta^2)}{2} \int_0^{t_K} \frac{\beta_{T-u}}{\tau} du \right).$$

If $\left\| \nabla^2 \log q_t(x_t) \right\| \leq \Gamma/\sigma_t^2$, $\|\nabla \mathbf{Y}_{s,t_K,i}^x\| \leq \sigma_{T-t_K}^{-(1+\eta^2)\Gamma} \exp \left( \left( (1+\eta^2) \Gamma + 2 \right) \int_0^{t_K} \frac{\beta_{T-u}}{\tau} du \right).$

**Proof.** Using the definition of the tangent process and Lemma E.1, we have

$$d \left\| \nabla \mathbf{Y}_{s,t}^x \right\|^2$$
$$\leq 2\beta_{T-t} \left( \frac{1}{\tau} \left\| \nabla \mathbf{Y}_{s,t}^x \right\|^2 - (1+\eta^2) \left( 1 - m_{T-t}^2 R^2 / \left( 2\sigma_{T-t}^2 \right) \right) / \sigma_{T-t}^2 \left\| \nabla \mathbf{Y}_{s,t}^x \right\|^2 \right) dt.$$

Using Lemma F.1, we have

$$\int_s^t \beta_{T-u} \left( \frac{1}{\tau} - (1+\eta^2) / \sigma_{T-u}^2 + (1+\eta^2) m_{T-u}^2 R^2 / 2\sigma_{T-u}^4 \right) du$$

$$\leq \left( (1+\eta^2) R^2/4 \right) \left( \sigma_{T-t}^{-2} - \sigma_{T-s}^{-2} \right) + \frac{1-\eta^2}{2} \int_s^t \frac{\beta_{T-u}}{\tau} du$$

$$\leq \frac{(1+\eta^2) R^2}{4\sigma_{T-t}^2} + \frac{1-\eta^2}{2} \int_s^t \frac{\beta_{T-u}}{\tau} du.$$

Note that $\nabla \mathbf{Y}_{s,s} = \mathbf{I}$, we get

$$\|\nabla \mathbf{Y}_{s,t_K}^x\|^2 \leq \exp \left[ \frac{(1+\eta^2) R^2}{2\sigma_{T-t}^2} + (1-\eta^2) \int_0^{t_K} \frac{\beta_{T-u}}{\tau} du \right].$$

When we assume $\left\| \nabla \log q_t^2(x_t) \right\| \leq \Gamma/\sigma_t^2$, we know that

$$d \left\| \nabla \mathbf{Y}_{s,t}^x \right\|^2 \leq 2\beta_{T-t} \left( \frac{1}{\tau} - \frac{(1+\eta^2) \Gamma}{\sigma_{T-t}^2} \right) \left\| \nabla \mathbf{Y}_{s,t}^x \right\|^2 dt.$$

Using Lemma F.1, we have

$$2 \int_s^t \beta_{T-u} / \sigma_{T-u}^2 du$$

$$\leq \log \left( \exp \left[ 2 \int_0^{T-s} \frac{\beta_{T-u}}{\tau} du \right] - 1 \right) - \log \left( \exp \left[ 2 \int_0^{T-t} \frac{\beta_{T-u}}{\tau} du \right] - 1 \right)$$

$$\leq \log \left( \sigma_{T-s}^2 \right) - \log \left( \sigma_{T-t}^2 \right) + \int_{T-t}^{T-s} \frac{\beta_u}{\tau} du.$$

Then we have

$$\|\nabla \mathbf{Y}_{s,t_K}^x\|^2 \leq \sigma_{T-t_K}^{-(1+\eta^2)\Gamma} \exp \left[ \left( (1+\eta^2) \Gamma + 2 \right) \int_0^{t_K} \frac{\beta_{T-u}}{\tau} du \right].$$

Thus we complete our proof. $\blacksquare$

After bounding the gradient of the tangent process, the remaining term is $\|\Delta b\|$:

$$\|\Delta b\| \le \|\Delta^{(a,b)} b\| + \|\Delta^{(b,c)} b\| + \|\Delta^{(c,d)} b\|, \tag{7}$$

where $b^{(a)} = b$ and $b^{(d)} = \bar{b}$. Moreover,

$$b_u^{(b)}(\omega) = \beta_{T-u}(\frac{1}{\tau}\omega_u + (1+\eta^2)\nabla \log q_{T-s_k}(\omega_u)),$$

$$b_u^{(c)}(\omega) = \beta_{T-u}(\frac{1}{\tau}\omega_u + (1+\eta^2)\nabla \log q_{T-s_k}(\omega_{s_k})),$$

$$\Delta_b^{a,b} = b^{(a)} - b^{(b)},\ \Delta_b^{b,c} = b^{(b)} - b^{(c)},\ \Delta_b^{c,d} = b^{(c)} - b^{(d)}.$$

We then control $\|\Delta^{(a,b)} b\|, \|\Delta^{(b,c)} b\|, \|\Delta^{(c,d)} b\|$ separately. In this section, $\|\Delta^{(c,d)} b\| = 0$ since we assume that the accurate score function is achieved. For $\left\|\Delta^{(a,b)} b_u(\omega)\right\|$, we have the following lemma.

**Lemma D.1.** *For $s, u \in [0, T)$ such that $u \ge s, u \in [s_k, t_{k+1})$ and $\omega = (\omega_v)_{v \in [s,T]}$ we have*

$$\|\Delta^{(a,b)} b_u(\omega)\|$$
$$\le (1 + \eta^2)\beta_{T-u} \sup_{v \in [T-u, T-t_k]} (\beta_v/\sigma_v^6)(2 + R^2)(R + \|\omega_u\|)\gamma_k.$$

**Proof.** Without loss of generality, we assume $s \le t_k$. Then

$$\|\Delta^{(a,b)} b_u(\omega)\| \le (1 + \eta^2)\beta_{T-u}\|\nabla \log q_{T-u}(\omega_u) - \nabla \log q_{T-t_k}(\omega_u)\|$$
$$\le (1 + \eta^2)\beta_{T-u}\gamma_k \sup_{v \in [T-u, T-t_k]} \|\partial_v \nabla \log q_{T-v}(\omega_u)\|.$$

Then by Lemma E.4, we have

$$\|\Delta^{(a,b)} b_u(\omega)\|$$
$$\le (1 + \eta^2)\beta_{T-u} \sup_{v \in [T-u, T-t_k]} (\beta_v/\sigma_v^6)(2 + R^2)(R + \|\omega_u\|)\gamma_k.$$

∎

For $\left\|\Delta^{(b,c)} b_u(\omega)\right\|$, we have the following lemma.

**Lemma D.2.** *For $s, u \in [0, T)$ such that $u \ge s, u \in [s_k, t_{k+1})$ and $\omega = (\omega_v)_{v \in [s,T]}$ we have*

$$\|\Delta^{(b,c)} b_u(\omega)\| \le (1 + \eta^2)(\beta_{T-u}/\sigma_{T-u}^4)(1 + R^2)\|\omega_u - \omega_{s_k}\|.$$

**Proof.** Without loss of generality, we assume $s \le t_k$. In this case $s_k = t_k$, Then

$$\|\Delta^{(b,c)} b_u(\omega)\| \le (1 + \eta^2)\beta_{T-u}\|\nabla \log q_{T-t_k}(\omega_{t_k}) - \nabla \log q_{T-t_k}(\omega_u)\|$$
$$\le (1 + \eta^2)\beta_{T-u} \sup_{v \in [u, T-t_k]} \|\nabla^2 \log q_{T-t_k}(\omega_v)\|\|\omega_u - \omega_{t_k}\|.$$

Using Lemma E.2, we have that

$$\|\Delta^{(b,c)} b_u(\omega)\| \le (1 + \eta^2)(\beta_{T-u}/\sigma_{T-u}^4)(1 + R^2)\|\omega_u - \omega_{t_k}\|.$$

Then the proof is complete. ∎

We need to control the reverse process when dealing with $\Delta b$. The following lemma shows an upper bound for the reverse $Y_k$.

**Lemma D.3.** *Assume Assumption 3.1 ,Assumption 4.1, and there exists $\delta > 0$ such that $\frac{\gamma_k \beta_{T-t_k}}{\sigma_{T-t_k}^2} \le \delta \le 1/28$ for any $k \in \{0, \cdots, K\}$, then we have*

$$\mathbb{E}[\|Y_k\|^2] \le U(\tau) = \tau d + B(1/A + \delta),$$

*where*

$$A = 4\eta^2 + 2 - 2\delta - 4(1+\eta^2)(1+\delta)\mu R$$

$$B = 4(1+\eta^2)R^2\delta + 2(1+\eta^2)(1+\delta)\frac{R}{\mu} + 4\eta^2\tau d$$

*and $\mu$ is an arbitrary positive number which makes $A > 0$. In particular, if $\delta \leq 1/28$, then*

$$\mathbb{E}[\|Y_k\|^2] \leq U_0(\tau) = 111R^2 + 13\tau d.$$

**Proof**. Recall the discretization of the backward process (the explicit form of Equation (6))

$$Y_{k+1} = Y_k + \gamma_{1,k}\left(\frac{1}{\tau}Y_k + (1+\eta^2)\mathbf{s}\left(T - t_k, Y_k\right)\right) + \eta\sqrt{2\gamma_{2,k}}Z_k,$$

$$\gamma_{1,k} = \exp\left[\int_{T-t_{k+1}}^{T-t_k}\beta_s\,\mathrm{d}s\right] - 1, \quad \gamma_{2,k} = \left(\exp\left[2\int_{T-t_{k+1}}^{T-t_k}\beta_s\,\mathrm{d}s\right] - 1\right)/2,$$

where $\{Z_k\}_{k\in K}$ are independent Gaussian random variables. It is clear that $\gamma_{1,k} \leq \gamma_{2,k} \leq 2\gamma_{1,k}$, and using Lemma E.1 we have

$$\langle x_t, \mathbf{s}(t, x_t)\rangle = \langle x_t, \nabla \log q_t(x_t)\rangle$$
$$\leq -\|x_t\|^2/\sigma_t^2 + m_t R\|x_t\|/\sigma_t^2$$
$$\leq (-1 + \mu m_t R)\|x_t\|^2/\sigma_t^2 + (m_t R/\mu)/\sigma_t^2,$$

where the first equality follows that we assume the accurate score function. For any $\mu > 0$. Again using Lemma E.1, we have

$$\|\mathbf{s}(t, x_t)\|^2 = \|\nabla \log q_t(x_t)\|^2$$
$$\leq 2\|x_t\|^2/\sigma_t^4 + 2m_t^2 R^2/\sigma_t^4.$$

Combining the results above, we have

$$\mathbb{E}[\|Y_{k+1}\|^2] = (1 + \frac{\gamma_{1,k}}{\tau})^2\mathbb{E}[\|Y_k\|^2] + (1+\eta^2)^2\gamma_{1,k}^2\mathbb{E}[\|s(T - t_k, Y_k)\|^2]$$
$$+ 2(1+\eta^2)(1 + \frac{\gamma_{1,k}}{\tau})\gamma_{1,k}\mathbb{E}[\langle Y_k, s(T - t_k, Y_k)\rangle] + 2\eta^2\gamma_{2,k}d$$
$$\leq ((1 + \frac{\gamma_{1,k}}{\tau})^2 + 2(1+\eta^2)^2\gamma_{1,k}^2/\sigma_{T-t_k}^4$$
$$+ 2(1+\eta^2)(1 + \frac{\gamma_{1,k}}{\tau})\gamma_{1,k}(-1 + \mu m_{T-t_k}R)/\sigma_{T-t_k}^2)\mathbb{E}[\|Y_k\|^2]$$
$$+ \frac{2m_{T-t_k}^2 R^2}{\sigma_{T-t_k}^4}(1+\eta^2)^2\gamma_{1,k}^2 + \frac{m_{T-t_k}R}{\mu\sigma_{T-t_k}^2}(1+\eta^2)(1 + \frac{\gamma_{1,k}}{\tau})\gamma_{1,k} + 4\eta^2\gamma_{1,k}d.$$

If we denote $\delta_k = \gamma_{1,k}/\sigma_{T-t_k}^2$ and notice the fact that $m_t \in [0, 1], \sigma_t^2 \in [0, \tau], \eta \in [0, 1]$, then we have

$$\mathbb{E}[\|Y_{k+1}\|^2] \leq (1 + 2\delta_k + \delta_k^2)\mathbb{E}[\|Y_k\|^2] + 8\delta_k^2\mathbb{E}[\|Y_k\|^2]$$
$$+ 2(1 + \delta_k)\delta_k(-1 + \mu R)\mathbb{E}[\|Y_k\|^2] + 8R^2\delta_k^2 + \frac{2R}{\mu}\delta_k(1 + \delta_k) + 4\tau\delta_k d.$$

We also have that

$$\gamma_{1,k} = \exp[\int_{T-t_{k+1}}^{T-t_k}\beta_s\mathrm{d}s] - 1 \leq \exp[\beta_{T-t_k}\gamma_k] - 1 \leq 2\beta_{T-t_k}\gamma_k,$$

where the last inequality follows that $\gamma_k = \exp(-T)$, $\beta_{T-t_k}\gamma_k \leq 1/2$ for small enough stepsize, and $e^\omega - 1 \leq 2\omega$ for any $\omega \in [0, 1/2]$. We get $\delta_k \leq 2\gamma_k\beta_{T-t_k}/\sigma_{T-t_k}^2 \leq 2\delta$. Thus

$$\mathbb{E}[\|Y_{k+1}\|^2] \leq (1 + 2\delta_k + 2\delta_k\delta)\mathbb{E}[\|Y_k\|^2] + 16\delta_k\delta\mathbb{E}[\|Y_k\|^2]$$
$$+ 4(1 + \delta)(-1 + \mu R)\delta_k\mathbb{E}[\|Y_k\|^2] + 16R^2\delta_k\delta + 4(1 + \delta)\frac{R}{\mu}\delta_k + 4\tau d\delta_k.$$

Hence, we have

$$\mathbb{E}[\|Y_{k+1}\|^2] \le (1 + \delta_k[-2 + 14\delta + 4(1 + \delta)\mu R])\mathbb{E}[\|Y_k\|^2]$$
$$+ \delta_k[16R^2\delta + 4(1 + \delta)\frac{R}{\mu} + 4\tau d].$$

We denote $A = 2 - 14\delta - 4(1 + \delta)\mu R$ and $B = 16R^2\delta + 4(1 + \delta)\frac{R}{\mu} + 4\tau d$, then

$$\mathbb{E}[\|Y_{k+1}\|^2] \le (1 - \delta_k A)\mathbb{E}[\|Y_k\|^2] + \delta_k B.$$

Notice that $\mathbb{E}[\|Y_0\|^2] = d\tau$ and if $\mathbb{E}[\|Y_k\|^2] \ge B/A$ it is decreasing, if $\mathbb{E}[\|Y_k\|^2] \le B/A$ we have $\mathbb{E}[\|Y_{k+1}\|^2] \le B/A + \delta B$. so

$$\mathbb{E}[\|Y_k\|^2] \le \tau d + B(1/A + \delta).$$

Notice that when $\delta \le 1/28$, if we choose $\mu = 1/(4(1 + \delta)R)$, $A \ge 1/2$, and

$$B \le 37R^2 + 4\tau d.$$

Then, the proof is complete. ∎

The following lemma shows a discretization error in the $k$-the interval.

**Lemma D.4.** *Assume Assumption 3.1,Assumption 4.1 and $\gamma_k\beta_{T-t_k}/\sigma^2_{T-t_k} \le 1/28$ for any $k \in \{0, \cdots, K-1\}$. Then for any $k$, $t \in [t_k, t_{k+1}]$ and $i \in \{1, 2\}$, we have that*

$$\mathbb{E}[\|\bar{\mathbf{Y}}_t - \bar{\mathbf{Y}}_{t_k}\|^2] \le L_i(\tau)\beta_{T-t_k}\gamma_k,$$

*where $L_i(\tau) = \bar{\gamma}_K\kappa_i(\tau)(\frac{64}{\sigma^2_{T-t_K}(i)} + \frac{8}{\tau})U_0(\tau) + 64R^2\frac{\bar{\gamma}_K\kappa_i(\tau)}{\sigma^2_{T-t_K}(i)} + 4d$, $\bar{\gamma}_K$, $\kappa_i(\tau)$ is defined in Lemma D.5 and $U_0(\tau)$ is defined in Lemma D.3.*

**Proof.** Recall the discretized backward process

$$\bar{\mathbf{Y}}_t = \bar{\mathbf{Y}}_{t_k} + (\exp[\int_{T-t}^{T-t_k} \beta_s\mathrm{d}s] - 1)(\frac{1}{\tau}\bar{\mathbf{Y}}_{t_k} + (1 + \eta^2)\mathbf{s}(T - t_k, \bar{\mathbf{Y}}_{t_k}))$$
$$+ \eta(\exp[2\int_{T-t}^{T-t_k} \beta_s\mathrm{d}s - 1])^{1/2}Z,$$

where $Z$ is a standard Gaussian random variable. By directly calculating, we have that

$$\mathbb{E}[\|\bar{\mathbf{Y}}_t - \bar{\mathbf{Y}}_{t_k}\|^2] = 2(\exp[\int_{T-t}^{T-t_k} \beta_s\mathrm{d}s] - 1)^2(\frac{1}{\tau^2}\mathbb{E}[\|\bar{\mathbf{Y}}_{t_k}\|^2] + (1 + \eta^2)^2\mathbb{E}[\|\mathbf{s}(T - t_k, \bar{\mathbf{Y}}_{t_k})\|^2])$$
$$+ \eta^2(\exp[2\int_{T-t}^{T-t_k} \beta_s\mathrm{d}s] - 1)d.$$

By Lemma E.1 and accurate score function assumption,

$$\|\mathbf{s}(T - t_k, \bar{\mathbf{Y}}_{t_k})\|^2 \le 2\|\bar{\mathbf{Y}}_{t_k}\|^2/\sigma^4_{T-t_k}(i) + 2m^2_{T-t_k}R^2/\sigma^4_{T-t_k}(i).$$

So we have that

$$\mathbb{E}[\|\bar{\mathbf{Y}}_t - \bar{\mathbf{Y}}_{t_k}\|^2] \le 2(\exp[\int_{T-t}^{T-t_k} \beta_s\mathrm{d}s] - 1)^2((\frac{8}{\sigma^4_{T-t_k}(i)} + \frac{1}{\tau^2})\mathbb{E}[\|\bar{\mathbf{Y}}_{t_k}\|^2] + \frac{8R^2}{\sigma^4_{T-t_k}(i)})$$
$$+ (\exp[2\int_{T-t}^{T-t_k} \beta_s\mathrm{d}s] - 1)d.$$

By $e^{2w} - 1 \le 1 + 4w$ for any $w \in [0, 1/2]$ and $\gamma_k \sup_{v \in [T-t_{k+1}, T-t_k]} \beta_v/\sigma^2_v \le 1/28$ for any $k \in \{0, ..., K-1\}$, we have

$$\exp[\rho\int_{T-t}^{T-t_k} \beta_s\mathrm{d}s] - 1 \le 2\rho\beta_{T-t_k}\gamma_k.$$

for $\rho = 1, 2$. And using Lemma D.3 and Lemma F.2 we have

$$\mathbb{E}[\|\bar{\mathbf{Y}}_t - \bar{\mathbf{Y}}_{t_k}\|^2]$$
$$\leq \left(\frac{64\gamma_k}{\sigma_{T-t_k}^4(i)} + \frac{8\beta_{T-t_k}\gamma_k}{\tau^2}\right)U_0(\tau)\beta_{T-t_k}\gamma_k + 64R^2\frac{\gamma_k}{\sigma_{T-t_k}^4(i)}\beta_{T-t_k}\gamma_k + 4d\beta_{T-t_k}\gamma_k.$$

We denote $L_i(\tau) = \bar{\gamma}_K\kappa_i(\tau)\left(\frac{64}{\sigma_{T-t_K}^2(i)} + \frac{8}{\tau}\right)U_0(\tau) + 64R^2\frac{\bar{\gamma}_K\kappa_i(\tau)}{\sigma_{T-t_K}^2(i)} + 4d$ for $i \in \{1, 2\}$ and the proof is complete. ∎

**Lemma D.5.** *Assume Assumption 3.1 and Assumption 4.1, $\gamma_k \sup_{v\in[T-t_{k+1}, T-t_k]}\beta_v/\sigma_v^2 \leq 1/28$ for any $k \in \{0, ..., K-1\}$. Let $\bar{\gamma}_K = argmax_{k\in\{0,...,K-1\}}\gamma_k$, $\kappa_i(\tau) = \max\{\bar{\beta}, \frac{T^2}{T^{-1+i}}\}\sigma_{T-t_K}^{-2}(i)$, and*

$$C_i(\tau) = 2(2 + R^2)(R + U_0^{1/2}(\tau)) + 2L_i^{1/2}(\tau)\tau^{3/2}(1 + R^2),$$

*for $i \in \{1, 2\}$. Then, for any $s, u \in [0, t_K]$ with $u \geq s$ and $i \in \{1, 2\}$, we have*

$$\mathbb{E}[\|\Delta b_{u,i}((\bar{\mathbf{Y}}_{s,v})_{v\in[s,T]})\|] \leq C_i(\tau)[\kappa_i^2(\tau)\sigma_{T-t_K}^{-2}(i)\bar{\gamma}_K^{1/2} + \kappa_i^2(\tau)]\bar{\gamma}_K^{1/2},$$

*where $\bar{\mathbf{Y}}_{s,s} \sim N(0, \mathbf{I})$.*

**Proof.** Combining Lemma D.1, Lemma D.2 and the exact score function, we get

$$\|\Delta b_{u,i}(\omega)\| \leq (1 + \eta^2) \sup_{v\in[T-t_{k+1}, T-t_k]} (\beta_v^2/\sigma_v^6(i))(2 + R^2)(\text{diam}(\mathcal{M} + \|\omega_u\|))\gamma_k$$
$$+ (1 + \eta^2)(\beta_{T-u}/\sigma_{T-u}^4(i))(1 + \text{diam}(\mathcal{M}^2))\|\omega_u - \omega_{s_k}\|.$$

For any $u \in [T - t_K, T]$, using Lemma F.3 we have $\beta_u/\sigma_u^2(i) \leq \kappa_i(\tau)$. Hence,

$$\|\Delta b_{u,i}(\omega)\| \leq (1 + \eta^2) \sup_{v\in[T-t_{k+1}, T-t_k]} (\beta_v^2/\sigma_v^6(i))(2 + \text{diam}(\mathcal{M}^2))(R + \|\omega_u\|)\gamma_k$$
$$+ (1 + \eta^2)(\beta_{T-u}/\sigma_{T-u}^4(i))(1 + \text{diam}(\mathcal{M}^2))(\|\omega_u - \omega_{t_k}\|)$$
$$\leq (1 + \eta^2)(\kappa_i^2(\tau)/\sigma_{T-t_{k+1}}^2(i))\gamma_k(2 + \text{diam}(\mathcal{M}^2))(R + \|\omega_u\|)$$
$$+ (1 + \eta^2)\kappa_i^2(\tau)(1 + R^2)\|\omega_u - \omega_{t_k}\|/\beta_{T-u}.$$

Combining this with Lemma D.3 and Lemma D.4,

$$\mathbb{E}[\|\Delta b_{u,i}((\bar{\mathbf{Y}}_{s,v})_{v\in[s,T]})\|] \leq (1 + \eta^2)(\kappa_i^2(\tau)/\sigma_{T-t_{k+1}}^2(i))\bar{\gamma}_K(2 + R^2)(R + U_0^{1/2}(\tau))$$
$$+ (1 + \eta^2)\kappa_i^2(\tau)(1 + R^2)L_i^{1/2}(\tau)\max\{\bar{\beta}, \tau\}^{3/2}\bar{\gamma}_K^{1/2}.$$

We denote $C_i(\tau) = 2(2 + R^2)(R + U_0^{1/2}(\tau)) + 2L_i^{1/2}(\tau)\tau^{3/2}(1 + R^2)$, for $i \in \{1, 2\}$, then we have

$$\mathbb{E}[\|\Delta b_{u,i}((\bar{\mathbf{Y}}_{s,v})_{v\in[s,T]})\|] \leq C_i(\tau)((\kappa_i^2(\tau)/\sigma_{T-t_{k+1}}^2)\bar{\gamma}_K + \kappa_i^2(\tau)\bar{\gamma}_K^{1/2}).$$

∎

**Lemma D.6.** *Assume Assumption 3.1 and Assumption 4.1, $\gamma_k \sup_{v\in[T-t_{k+1}, T-t_k]}\beta_v/\sigma_v^2 \leq 1/28$ for any $k \in \{0, ..., K-1\}$. Let $\bar{\gamma}_K = argmax_{k\in\{0,...,K-1\}}\gamma_k$, $\gamma_K = \delta$, and $\delta \leq 1/32$. Then*

$$W_1\left(R_K^{q_\tau}, Q_{t_K}^{q_\tau}\right) \leq C_i(\tau)\kappa_i^2(\tau)T\exp\left[\frac{R^2}{2\sigma_{T-t_K}^2(i)} + \frac{(1 - \eta^2)}{2}\right]\left[\frac{\bar{\gamma}_K^{1/2}}{\sigma_{T-t_K}^2(i)} + 1\right]\bar{\gamma}_K^{1/2},$$

*where $C_i(\tau), \kappa_i(\tau)$ for $i \in \{1, 2\}$ are the same terms to Theorem 6.1.*

**Proof.** By **Proposition 1** we have

$$\|\mathbf{Y}_{t_K} - Y_K\| = \|\mathbf{Y}_{t_K} - \bar{\mathbf{Y}}_{t_K}\| \leq \int_0^{t_K} \|\nabla \mathbf{Y}_{u,t_K,i}(\bar{\mathbf{Y}}_{0,u})\| \|\Delta b_{u,i}((\bar{\mathbf{Y}}_{0,v})_{v\in[0,T]})\| \mathrm{d}u.$$

$$\|\mathbf{Y}_{t_K} - Y_K\|$$
$$\leq \exp\left[\frac{(1+\eta^2)R^2}{4\sigma_{T-t}^2(i)} + \frac{(1-\eta^2)}{2}\int_0^{t_K}\frac{\beta_{T-u}}{\tau}\mathrm{d}u\right]\int_0^{t_K}\|\Delta b_{u,i}((\bar{\mathbf{Y}}_{0,v})_{v\in[0,T]})\|\mathrm{d}u\,.$$

Then by definition of Wasserstein distance, we have

$$W_1(q_\infty Q_{t_K}, q_\infty R_K)$$
$$\leq \mathbb{E}[\|\mathbf{Y}_{t_K} - Y_K\|]$$
$$\leq \exp\left[\frac{(1+\eta^2)R^2}{4\sigma_{T-t_K}^2(i)} + \frac{(1-\eta^2)}{2}\int_0^{t_K}\frac{\beta_{T-u}}{\tau}\mathrm{d}u\right]\int_0^{t_K}\mathbb{E}[\|\Delta b_{u,i}((\bar{\mathbf{Y}}_{0,v})_{v\in[0,T]})\|]\mathrm{d}u$$
$$\leq C_i(\tau)T\exp\left[\frac{(1+\eta^2)R^2}{4\sigma_{T-t_K}^2(i)} + \frac{(1-\eta^2)}{2}\right][\kappa_i^2(\tau)\sigma_{T-t_K}^{-2}(i)\bar{\gamma}_K^{1/2} + \kappa_i^2(\tau)]\bar{\gamma}_K^{1/2}\,.$$

■

**Theorem 6.1.** *Assume Assumption 3.1 and 4.1, $\delta \leq 1/32$ and $\gamma_k \sup_{v\in[T-t_{k+1},T-t_k]}\beta_v/\sigma_v^2 \leq 1/28$ for $\forall k \in \{0,...,K-1\}$. Let $\gamma_K = \delta$. Then, for $\forall \tau \in [T, T^2]$:*

*(1) If $\eta = 1$ (the reverse SDE), choosing $\beta_t = t^2$, $W_1\left(R_K^{q_\infty^\tau}, q_0\right)$ is bounded by*

$$(\frac{R}{\tau} + \sqrt{d})\sqrt{\delta} + \exp\left(\frac{R^2}{2}(\frac{\bar{\beta}}{\delta^3} + \frac{1}{\tau})\right)\left(C_1(\tau)T\kappa_1^2(\tau)\left((\frac{\bar{\beta}}{\delta^3} + \frac{1}{\tau})\bar{\gamma}_K^{1/2} + 1\right)\bar{\gamma}_K^{1/2} + \frac{\bar{D}e^{-T/2}}{\sqrt{\tau}}\right),$$

*where $\kappa_1(\tau) = T^2(1/\tau + \bar{\beta}/\delta^3)$ and $C_1(\tau)$ is linear in $\tau^2$.*

*(2) If $\eta = 0$ (PFODE), choosing a conservative $\beta_t$ (Assumption 3.1), $W_1\left(R_K^{q_\infty^\tau}, q_0\right)$ is bounded by*

$$(\frac{R}{\tau} + \sqrt{d})\sqrt{\delta} + \exp\left(\frac{R^2}{2}(\frac{\bar{\beta}}{\delta^2} + \frac{1}{\tau})\right)\left(C_2(\tau)\kappa_2^2(\tau)T\left((\frac{\bar{\beta}}{\delta^2} + \frac{1}{\tau})\bar{\gamma}_K^{1/2} + 1\right)\bar{\gamma}_K^{1/2} + \frac{\bar{D}}{\sqrt{\tau}}\right),$$

*where $\kappa_2(\tau) = T\left(1/\tau + \bar{\beta}/\delta^2\right)$ and $C_2(\tau)$ is linear in $\tau^2$.*

**Proof.** To obtain the convergence guarantee, we need to control three error terms:

$$W_1\left(R_K^{q_\infty^\tau}, q_0\right) \leq W_1\left(R_K^{q_\infty^\tau}, Q_{t_K}^{q_\infty^\tau}\right) + W_1\left(Q_{t_K}^{q_\infty^\tau}, Q_{t_K}^{q_0 P_T}\right) + W_1\left(Q_{t_K}^{q_0 P_T}, q_0\right)\,.$$

For term $W_1\left(R_K^{q_\infty^\tau}, Q_{t_K}^{q_\infty^\tau}\right)$, we use Lemma D.6.

For the second term, we define $\left(\mathbf{Y}_{0,t}^x\right)_{t\in[0,T]}$ and $\left(\mathbf{Y}_{0,t}^y\right)_{t\in[0,T]}$ be the reverse processes with initial condition $x$ and $y$. Then we have

$$\|\mathbf{Y}_{0,t}^x - \mathbf{Y}_{0,t}^y\| \leq \|x - y\|\int_0^1 \|\nabla \mathbf{Y}_{0,t}^{z_\lambda}\|d\lambda\,,$$

where $z_\lambda = \lambda x + (1-\lambda)y$. In this work, we choose $x \sim q_\infty^\tau$ and $y \sim q_0 P_T$. Combined with the above inequality, Theorem 4.2 and Lemma 6.3, we know that:

$$W_1\left(Q_{t_K}^{q_\infty^\tau}, Q_{t_K}^{q_0 P_T}\right)$$
$$\leq \exp\left[\frac{R^2}{2\sigma_{T-t_K}^2(i)} + \frac{(1-\eta^2)}{2}\int_0^{t_K}\frac{\beta_{T-u}}{\tau}\mathrm{d}u\right]\|q_0 P_T - q_\infty^\tau\|$$
$$\leq \frac{\sqrt{m_T}\bar{D}}{\sigma_T}\exp\left[\frac{R^2}{2\sigma_{T-t_K}^2(i)} + \frac{(1-\eta^2)}{2}\int_0^{t_K}\frac{\beta_{T-u}}{\tau}\mathrm{d}u\right]\,.$$

For the last term, we use exactly the same process with Bortoli [2022] with bounded $\sigma_{T-t_K}^2$:

$$W_1\left(Q_{t_K}^{q_0 P_T}, q_0\right) \leq \mathbb{E}\left[\|X - m_{T-t_K}X + \sigma_{T-t_K}Z\|\right]$$

$$\leq (\frac{R}{\tau} + \sqrt{d})\sigma_{T-t_K}$$

$$\leq 2(\frac{R}{\tau} + \sqrt{d})\sqrt{\delta},$$

where the second inequality follows that $\sigma_{T-t_K}^2 + \tau m_{T-t_K} = \tau$. ∎

In the end of the section, we provide the proof of Corollary 6.2.

**Corollary D.7.** *Assume Assumption 3.1, 4.1 and $\left\|\nabla^2 \log q_t(x_t)\right\| \leq \Gamma/\sigma_t^2$. Let $\eta = 0$ (reverse PFODE), $\delta \in (0, 1/32), \tau = T^2, \beta_t = t$ and $\kappa_2(\tau), C_2(\tau)$ defined in Theorem 6.1, we have*

$$W_1\left(R_K^{q_\infty^\tau}, q_0\right) \leq (\frac{R}{\tau} + \sqrt{d})\sqrt{\delta} + \frac{\bar{\beta}^{\frac{\Gamma}{2}}}{\delta^\Gamma} \exp\left(\frac{\Gamma+2}{2}\right)\left(C_2(\tau)\kappa_2^2(\tau)T((\frac{\bar{\beta}}{\delta^2} + \frac{1}{\tau})\bar{\gamma}_K^{1/2} + 1)\bar{\gamma}_K^{1/2} + \frac{\bar{D}}{\sqrt{\tau}}\right).$$

**Proof.** The proof of this corollary is almost identical to the proof of Theorem 6.1. We just need to replace the first bound for the tangent process in Lemma 6.3 by the second bound. ∎

# E   Lemmas for the Logarithmic Density

In this section, we introduce auxiliary lemmas to control the gradient and Hessian of the logarithmic density under the manifold hypothesis. Lemma E.1, Lemma E.2 and Lemma E.3 come from Lemma C.1, Lemma C.2, and Lemma C.5 of Bortoli [2022]. Since these lemmas do not involve the relationship between $m_t$ and $\sigma_t$, we can directly use the results from Bortoli [2022]. Following Bortoli [2022], we also define a empirical version of $q_0$ with $N$ datapoints, i.e. $q_0^N = (1/N)\sum_{k=1}^N X^k$, with $\{X^k\}_{k=1}^N \sim q_0^{\otimes N}$. We denote by $(q_t^N)_{t>0}$ such that for any $t > 0$ the density w.r.t. the Lebesgue measure of the distribution of $\mathbf{X}_t^N$, and when $N \to +\infty$, $q_t^N = q_t$.

**Lemma E.1.** *Assume Assumption 4.1. Then for any $t \in (0, T]$ and $x_t \in \mathbb{R}^d$ we have that*

$$\langle \nabla \log q_t(x_t), x_t \rangle \leq -\|x_t\|^2/\sigma_t^2 + m_R\|x_t\|/\sigma_t^2.$$

*In addition, we have*

$$\|\nabla \log q_t(x_t)\|^2 \leq 2\|x_t\|^2/\sigma_t^4 + 2m_t^2 R^2/\sigma_t^4.$$

**Lemma E.2.** *Assume Assumption 4.1. Then for any $t \in (0, T]$, $x_t \in \mathbb{R}^d$ and $M \in \mathcal{M}_d\left(\mathbb{R}^d\right)$*

$$\langle M, \nabla^2 \log q_t(x_t) M \rangle \leq -\left(1 - m_t^2 R^2/\left(2\sigma_t^2\right)\right)/\sigma_t^2\|M\|^2.$$

*In addition, we have*

$$\left\|\nabla^2 \log q_t(x_t)\right\| \leq \left(1 + R^2\right)/\sigma_t^4.$$

The following lemma shows that the derivatives up to the fourth order are uniformly bounded since $\tau \in [T, T^2]$. Thus we can use the stochastic extension of the Alekseev–Gröbner formula [Del Moral and Singh, 2022].

**Lemma E.3.** *Assume Assumption 4.1. Then, there exists $\bar{C} \geq 0$ such that for any $t \in (0, T]$ we have*

$$\left\|\nabla^2 \log q_t(x)\right\| + \left\|\nabla^3 \log q_t(x)\right\| + \left\|\nabla^4 \log q_t(x)\right\| \leq \bar{C}/\sigma_t^8.$$

The following lemma shows that $\|\partial_t \nabla \log q_t(x_t)\|$ is bounded. The proof before using the relationship between $\sigma_t$ and $m_t$ is identical compared to Lemma C.3 in Bortoli [2022]. For the sake of completeness, we also give the proof process of this part.

**Lemma E.4.** *Assume Assumption 4.1. Then for any $t \in (0, T]$ and $x_t \in \mathbb{R}^d$ we have*

$$\|\partial_t \nabla \log q_t(x_t)\| \leq \left(\beta_t/\sigma_t^6\right)\left(2 + R^2\right)(R + \|x_t\|).$$

**Proof.** Let $N \in \mathbb{N}$ and $t \in (0, T]$. We denote for any $x \in \mathbb{R}^d$, $q_t^N(x) = \bar{q}_t^N(x) / \left(2\pi\sigma_t^2\right)^{d/2}$ with

$$\bar{q}_t^N(x) = (1/N) \sum_{k=1}^N e_t^k(x), \qquad e_t^k(x) = \exp\left[-\|x - m_t X^k\|^2 / \left(2\sigma_t^2\right)\right].$$

Next we denote $f_t^k \triangleq \log e_t^k$. Then we have

$$\partial_t \log \bar{q}_t^N(x) \sum_{k=1}^N \partial_t f_t^k(x) e_t^k(x) / \sum_{k=1}^N e_t^k(x).$$

Therefore we have

$$\partial_t \nabla \log \bar{q}_t^N(x)$$

$$= \sum_{k=1}^N \partial_t \nabla f_t^k(x) e_t^k(x) / \sum_{k=1}^N e_t^k(x) + \sum_{k=1}^N \partial_t f_t^k(x) \nabla f_t^k(x) e_t^k(x) / \sum_{k=1}^N e_t^k(x)$$

$$\quad - \sum_{k,j=1}^N \partial_t f_t^k(x) \nabla f_t^j(x) e_t^k(x) e_t^j(x) / \sum_{k,j=1}^N e_t^k(x) e_t^j(x)$$

$$= \sum_{k=1}^N \partial_t \nabla f_t^k(x) e_t^k(x) / \sum_{k=1}^N e_t^k(x)$$

$$\quad + (1/2) \sum_{k,j=1}^N \left(\partial_t f_t^k(x) - \partial_t f_t^j(x)\right) \left(\nabla f_t^k(x) - \nabla f_t^j(x)\right) e_t^k(x) e_t^j(x) / \sum_{k,j=1}^N e_t^k(x) e_t^j(x).$$

In what follows, we provide upper bounds for $|\partial_t f_t^k - \partial_t f_t^j|$, $\|\nabla f_t^k - \nabla f_t^j\|$ and $\partial_t \nabla f_t^k$. First we notice that $\nabla f_t^k(x) = -\left(x - m_t X^k\right)/\sigma_t^2$, and using $m_t \leq 1$ we get

$$\|\nabla f_t^k(x) - \nabla f_t^j(x)\| \leq m_R/\sigma_t^2 \leq R/\sigma_t^2.$$

and

$$\partial_t f_t^k(t) = \partial_t \sigma_t^2 / \left(2\sigma_t^4\right) \|x - m_t X^k\|^2 + \partial_t m_t / \sigma_t^2 \left\langle X^k, x - m_t X^k\right\rangle.$$

Notice the fact that $\partial_t \sigma_t^2 = -2\tau m_t \partial_t m_t = 2\beta_t m_t^2$ and $\partial_t m_t = -\dfrac{\beta_t}{\tau} m_t$, combined with the above equality, we know that

$$\partial_t f_t^k(t) = -\beta_t m_t / \sigma_t^2 \left[-\left(m_t/\sigma_t^2\right)\|x - m_t X^k\|^2 + \frac{1}{\tau}\left\langle x - m_t X^k, X^k\right\rangle\right]$$

$$= -\beta_t m_t / \sigma_t^2 \left\langle x - m_t X^k, -\left(m_t/\sigma_t^2\right)\left(x - m_t X^k\right) + \frac{1}{\tau} X^k\right\rangle$$

$$= -\beta_t m_t / \sigma_t^4 \left\langle x - m_t X^k, -m_t x + \left(m_t^2 + \frac{\sigma_t^2}{\tau}\right) X^k\right\rangle$$

$$= \beta_t m_t / \sigma_t^4 \left(m_t\|x\|^2 + m_t \left\|X^k\right\|^2 + \left(1 + m_t^2\right)\left\langle x, X^k\right\rangle\right),$$

where the last equality holds that $\tau m_t^2 + \sigma_t^2 = \tau$. The rest of the proof is identical to the Lemma C.3 in Bortoli [2022].

So using $m_t \leq 1$ we have

$$\left|\partial_t f_t^k(x) - \partial_t f_t^j(x)\right| \leq 2\beta_t m_t^2 R^2 / \sigma_t^4 + \beta_t m_t \left(1 + m_t^2\right) R\|x\| / \sigma_t^4$$

$$\leq 2 \left(\beta_t / \sigma_t^4\right) R(R + \|x\|)$$

Now we compute $\nabla \partial_t f_t^k(x)$ for any $x \in \mathbb{R}^d$

$$\nabla \partial_t f_t^k(x) = 2\beta_t m_t^2 / \sigma_t^4 x + \left(\beta_t m_t / \sigma_t^4\right)\left(1 + m_t^2\right) X^k.$$

So we can bound the norm of it by

$$\|\partial_t \nabla f_t^k(x)\| \le 2\left(\beta_t/\sigma_t^4\right)(R + \|x\|).$$

Combining results above we get for any $x \in \mathbb{R}^d$

$$\left\|\partial_t \nabla \log \bar{q}_t^N(x)\right\| \le 2\left(\beta_t/\sigma_t^4\right)(R + \|x\|) + \left(\beta_t/\sigma_t^6\right)R^2(R + \|x\|)$$
$$\le \left(\beta_t/\sigma_t^6\right)\left(2 + R^2\right)(R + \|x\|)$$

Note that

$$\lim_{N \to +\infty} \partial_t \nabla \log q_t^N(x_t) = \partial_t \nabla \log q_t$$

and the proof is complete. ∎

In the following lemma, similar to Chen et al. [2023c], we obtain a better control on the time discretization error instead of controlling $\|\partial_t \nabla \log q_t(x_t)\|$ for $\forall x_t \in \mathbb{R}^d$.

**Lemma E.5.** *Assume Assumption 4.1 and $X_t$ satisfies the forward process Equation (3). Define $L = \max_{t \in [0, T-\delta]} \|\nabla^2 \log q_{T-t}(\mathbf{Y}_t)\| \le \left(1 + R^2\right)/\sigma_\delta^4$, then we have that*

$$\mathbb{E}_{Q_{t_K}^{q_T^\tau}}\left[\left\|\nabla \ln \frac{q_{T-t_k}}{q_{T-t}}(\mathbf{Y}_{t_k})\right\|^2\right]$$
$$\lesssim \tau L^2 d\bar{\gamma}_K + \tau L^2 \bar{\gamma}_K^2(d\tau + R^2) + \tau L^3 \bar{\gamma}_K^2 + \tau L^4 \bar{\gamma}_K^2(\beta_T d\bar{\gamma}_K + R^2 \bar{\gamma}_K^2).$$

**Proof.** Due to the property of the forward process, we know that if $S : \mathbb{R}^d \to \mathbb{R}^d$ is the mapping $S(x) := \exp(-(t - t_k))x$, then $q_{T-t_k} = S_\# q_{T-t} * \text{normal}\left(0, \tau\left(1 - \exp(-2\int_{t_k}^{t_k+1} \beta_s/\tau ds)\right)\right)$
Similar to Chen et al. [2023c], we define $\alpha = \exp\left[\int_{t_k}^{t_k+1} \frac{\beta_s}{\tau} ds\right] = 1 + O(\bar{\gamma}_K)$ and $\sigma^2 = \tau\left(1 - \exp(-2\int_{t_k}^{t_k+1} \beta_s/\tau ds)\right) = O(\tau\bar{\gamma}_K)$. Then we can use Lemma C.12 of Lee et al. [2022] to obtain

$$\mathbb{E}_{Q_{t_K}^{q_T^\tau}}\left[\left\|\nabla \ln \frac{q_{T-t_k}}{q_{T-t}}(\mathbf{Y}_{t_k})\right\|^2\right]$$
$$\lesssim \tau L^2 d\bar{\gamma}_K + \tau L^2 \bar{\gamma}_K^2 \|\mathbf{Y}_{t_k}\|^2 + \tau L^2 \bar{\gamma}_K^2 \|\nabla \ln q_{T-t}(\mathbf{Y}_{t_k})\|^2$$
$$\lesssim \tau L^2 d\bar{\gamma}_K + \tau L^2 \bar{\gamma}_K^2(d\tau + R^2) + \tau L^3 \bar{\gamma}_K^2 + \tau L^4 \bar{\gamma}_K^2(\beta_T d\bar{\gamma}_K + R^2 \bar{\gamma}_K^2).$$

The last inequality follows Lemma F.4 and the fact that

$$\|\nabla \ln q_{T-t}(\mathbf{Y}_{t_k})\|^2 \lesssim \|\nabla \ln q_{T-t}(\mathbf{Y}_t)\|^2 + \|\nabla \ln q_{T-t}(\mathbf{Y}_{t_k}) - \nabla \ln q_{T-t}(\mathbf{Y}_t)\|^2$$
$$\lesssim \|\nabla \ln q_{T-t}(\mathbf{Y}_{t_k})\|^2 + L^2(\beta_T d\bar{\gamma}_K + R^2 \bar{\gamma}_K^2)$$
$$\lesssim L + L^2(\beta_T d\bar{\gamma}_K + R^2 \bar{\gamma}_K^2).$$

∎

# F   Auxiliary Lemmas

**Lemma F.1.** *For any $s, t \in [0, T]$ we have*

$$\int_s^t \beta_{T-u}/\sigma_{T-u}^2 du = \left[-\frac{1}{2}\log\left(\exp\left[2\int_0^{T-u} \frac{\beta_v}{\tau} dv\right] - 1\right)\right]_s^t,$$
$$\int_s^t \beta_{T-u} m_{T-u}^2/\sigma_{T-u}^4 du = \left[(1/2\tau)/\left(1 - \exp\left[-2\int_0^{T-u} \frac{\beta_v}{\tau} dv\right]\right)\right]_s^t.$$

**Proof.** We directly compute

$$\int_s^t \beta_{T-u}/\sigma_{T-u}^2 \mathrm{d}u = \frac{1}{\tau}\int_s^t \beta_{T-u}/\left(1 - \exp\left[-2\int_0^{T-u}\frac{\beta_v}{\tau}\mathrm{d}v\right]\right)\mathrm{d}u$$

$$= \frac{1}{\tau}\int_s^t \beta_{T-u}\exp\left[2\int_0^{T-u}\frac{\beta_v}{\tau}\mathrm{d}v\right]\Big/\left(\exp\left[2\int_0^{T_u}\frac{\beta_v}{\tau}\mathrm{d}v\right]-1\right)\mathrm{d}u$$

$$= -\frac{1}{2}\int_s^t \partial_u\log\left(\exp\left[2\int_0^{T-u}\frac{\beta_v}{\tau}\mathrm{d}v\right]-1\right)\mathrm{d}u\,.$$

Similarly

$$\int_s^t \beta_{T-u}m_{T-u}^2/\sigma_{T-u}^4$$

$$= \frac{1}{\tau^2}\int_s^t \beta_{T-u}\exp\left[-2\int_0^{T-u}\frac{\beta_v}{\tau}\mathrm{d}v\right]\Big/\left(1-\exp\left[-2\int_0^{T-u}\frac{\beta_v}{\tau}\mathrm{d}v\right]\right)^2\mathrm{d}u$$

$$= (1/2\tau)\int_t^s \partial_u\left(1-\exp\left[-2\int_0^{T-u}\frac{\beta_v}{\tau}\mathrm{d}v\right]\right)^{-1}\mathrm{d}u.$$

∎

**Lemma F.2.** *Assume Assumption 3.1. For $i \in \{1,2\}$, we have $\sigma_{T-t_K}^2(i) \leq 2\delta$ and $\sigma_u^{-2}(i) \leq \sigma_{T-t_K}^{-2}(i) \leq \frac{1}{\tau} + \frac{\bar\beta}{\delta^{4-i}}, \forall u \in [T - t_K, T]$.*

**Proof.**

$$\sigma_{T-t_K}^2(i) = \tau\left(1 - \exp\left[-2\int_0^{T-t_K}\frac{\beta_s}{\tau}\,\mathrm{d}s\right]\right)$$

$$\leq 2\int_0^{T-t_K}\beta_s\,\mathrm{d}s \leq 2\delta\,,$$

where the first inequality follows from for any $a \geq 0, \exp[-a] \geq 1 - a$; the second inequlity follows from Assumption 3.1 and $\delta \leq 1$.

$$\sigma_{T-t_K}^{-2}(i) = \frac{1}{\tau}\left(1-\exp\left[-2\int_0^{T-t_K}\frac{\beta_s}{\tau}\,\mathrm{d}s\right]\right)^{-1} \leq \frac{1}{\tau}\left(1+\left(2\int_0^{T-t_K}\frac{\beta_s}{\tau}\,\mathrm{d}s\right)^{-1}\right)$$

$$\leq \frac{1}{\tau} + \frac{\bar\beta}{\delta^{4-i}}\,,$$

where the first inequality follows from for any $a \geq 0, 1/(1+\exp[-a]) \leq 1+1/a$, the second inequality follows from Assumption 3.1. It is easy to check that $\sigma_u^{-2}(i) \leq \sigma_{T-t_K}^{-2}(i), \forall u \in [T - t_K, T]$.

∎

Using the bound on $\sigma_{T-t_K}^{-2}(i)$ immediately yields the following control of $\beta_u/\sigma_u^2(i)$.

**Lemma F.3.** *Assume Assumption 3.1. Then, we have for any $u \in [T - t_K, T]$: (1) if $i = 1$, then*

$$\frac{\beta_u}{\sigma_u^2(i=1)} \leq \kappa_1(\tau) = \max\{\bar\beta, T^2\}\left(\frac{1}{\tau} + \frac{\bar\beta}{\delta^3}\right);$$

*(2) if $i = 2$, then*

$$\frac{\beta_u}{\sigma_u^2(i=2)} \leq \kappa_2(\tau) = \max\{\bar\beta, T\}\left(\frac{1}{\tau} + \frac{\bar\beta}{\delta^2}\right).$$

Generally speaking, $T \geq \bar{\beta} \geq 1$. Hence, We can further simplify the above inequality by removing max.

In the rest of this section, we provide the useful lemma to achieve polynomial sample complexity for VE-based models with reverse SDE. As shown in Lemma E.1, we also need to control $\mathbb{E}[\|\mathbf{X}_t\|^2]$ in the forward process. The following lemmas shows that this term is bounded by the $R^2$ and exploding variance.

**Lemma F.4.** *Suppose that Assumption 4.1hold. Let $(\mathbf{X}_t)_{t \in [0,T]}$ denote the forward process Equation* (3). *Then, for all $t \geq 0$,*

$$\mathbb{E}\left[\|\mathbf{X}_t\|^2\right] \leq d\sigma_t^2 \vee R^2 .$$

**Proof.** As shown in Equation (4),

$$\mathbb{E}\left[\|\mathbf{X}_t\|^2\right] \leq \mathbb{E}\left[\|\mathbf{X}_0\|^2\right] + \sigma_t^2 d \leq d\sigma_t^2 \vee R^2 .$$

∎

**Lemma F.5** (movement bound for VESDE). *Let $(\mathbf{X}_t)_{t \in [0,T]}$ denote the forward process Equation* (3). *For $0 \leq s < t$ with $\delta := t - s$, if $\delta \leq 1$, then*

$$\mathbb{E}\left[\|\mathbf{X}_t - \mathbf{X}_s\|^2\right] \lesssim 2\beta_t \delta d + \delta^2 R^2 .$$

**Proof.**

$$\mathbb{E}\left[\|\mathbf{X}_t - \mathbf{X}_s\|^2\right] \lesssim \mathbb{E}\left[\left\|\sqrt{2\beta_t}\left(B_t - B_s\right)\right\|^2\right] + \delta \int_s^t \mathbb{E}\left[\|\mathbf{X}_r\|^2\right] dr \lesssim 2\beta_t \delta d + \delta^2 R^2 .$$

∎

Similar to Chen et al. [2023c], we can also show that if we do forward process for time $\delta$, $q_\delta$ will be close to $q_0$ in $W_2$ distance.

**Lemma F.6.** *Suppose Assumption 4.1 holds. Let $\epsilon_{W_2} > 0$. If $\beta_t^2 = t^2$ and $\tau = T^2$, we choose the early stopping parameter $\delta \leq \frac{\epsilon_{W_2}^{2/3}}{(d+R\sqrt{d})^{1/3}}$. If $\beta_t = t$ and $\tau = T$, we choose $\delta \leq \frac{\epsilon_{W_2}}{(d+R\sqrt{d})^{1/2}}$. If consider pure VESDE (SMLD) (Equation (2)) with $\sigma_t^2 = t$, we choose $\delta \leq \frac{\epsilon_{W_2}^2}{d}$. Then we have $W_2(q_\delta, q_0) \leq \epsilon_{W_2}$.*

**Proof.** For the forward process Equation (3), we know that $\mathbf{X}_t := m_t \mathbf{X}_0 + \sigma_t Z$, where $Z \sim$ normal $(0, I_d)$ is independent of $X_0$ and $m_t \leq 1$. Hence, for $\delta \lesssim 1$,

$$W_2^2(q_0, q_\delta) \leq (1 - m_t)^2 \mathbb{E}\left[\|\mathbf{X}_0\|^2\right] + \mathbb{E}\left[\|\sigma_\delta Z\|^2\right] .$$

For $\beta_t = t^2$ and $\tau = T^2$, we have that

$$W_2^2(q_0, q_\delta) \leq \delta^3 d + \frac{R^2 \delta^6}{T^2}$$

Hence, we can take $\delta \leq \frac{\epsilon_{W_2}^{2/3}}{(d+R\sqrt{d})^{1/3}}$. For $\beta_t = t$ and $\tau = T$, we have that

$$W_2^2(q_0, q_\delta) \leq \delta^2 d + \frac{R^2 \delta^4}{T^2}$$

Hence, we can take $\delta \leq \frac{\epsilon_{W_2}}{(d+R\sqrt{d})^{1/2}}$. For pure VESDE (Equation (2)) with $\sigma_t = t$, we have

$$W_2^2(q_0, q_\delta) \leq \delta d .$$

∎

# G   Additional Synthetic Experiments

In this section, we do synthetic experiments to show the power of our new forward process with small drift term in different setting.

## G.1   The Synthetic experiments with accurate score function

In this section, we do numerical experiments on 2-dimension Gaussian distribution to show the power of our new VESDE forward process in balancing different error sources.

**Experiment Setting.**   We set the mean of target distribution $\mathbb{E}[q_0] = [6, 8]$, the covariance matrix $\text{Cov}[q_0] = \begin{bmatrix} 25 & 5 \\ 5 & 4 \end{bmatrix}$, the diffusion time $T = 2$, $\tau = T^2$ and the reverse beginning distribution is $\mathcal{N}(0, T^2 \mathbf{I})$. We choose uniform stepsize $\gamma_k = h, \forall k \in [K]$ where $h \in \{0.005, 0.01, 0.02, 0.04\}$. For score functions, we directly calculate the ground truth score function instead of learning it by the score matching objective. We calculate the KL divergence between the generation distribution and target distribution $q_0$ as the experiments.

**The implementable algorithm.**   We choose three different VESDE forward processes in the experiments: (1) aggressive $\beta_t = t^2$ with $\tau = T^2$; (2) conservative $\beta_t = t$ with $\tau = T^2$ and (3) VESDE without drift term Equation (2) with $\sigma_t^2 = t^2$. After determining the forward process, we run the reverse SDE with the above $\gamma_k, k \in [K]$. For the discretization scheme, we choose two common method: exponential integrator (EI) [Zhang and Chen, 2022] and Euler-Maruyama (EM) discretization [Ho et al., 2020].

**Observations.**   The experimental results are shown in Figure 2. We note that the red line (EI, VESDE without drift, $\sigma_t^2 = t^2$) and orange line (conservative drift VESDE, $\beta_t = t$ and $\tau = T^2$) has a similar trend. Furthermore, the conservative drift VESDE has better performance compared to pure VESDE without drift term. Hence, our new forward process is representative enough to represent current VESDE, as discussed in Section 3.1.

The experimental results also support our theoretical results and show the power of the new forward process in balancing different error terms. As shown in Figure 2, the process with aggressive $\beta_t = t^2$ with small drift term achieves the best and second performance in EI and EM discretization since it can balance the reverse beginning and discretization. The third best process is conservative $\beta_t = t$ with the small drift term. The reason is that though it can not achieve a $\exp(-T)$ forward process guarantee, it also has a constant decay on prior information, as shown in Section 3.1. This decay slightly reduces the effect of the reverse beginning error. The worse process is VESDE without drift term since it is hard to balance different error sources. Our experimental results also show that EI discretization is better than EM discretization.

## G.2   The Synthetic experiments with approximated score function

In this section, instead of using an accurate score function, we train an approximated score function on the pure VESDE (Equation (2)) without drift term on two synthetic datasets: multiple Swiss rolls and 1-D GMM. Then, for the drift VESDE, we do not train the approximated score corresponding to Equation (3); we directly use the approximated score learned by pure VESDE and show that the drift VESDE can improve the generated distribution without the training process.

**Datasets.**   The 1-D GMM distribution contains three modes:

$$\frac{3}{10} \mathcal{N}(-8, 0.01) + \frac{3}{10} \mathcal{N}(-4, 0.01) + \frac{4}{10} \mathcal{N}(3, 1) \ .$$

For multiple Swiss rolls, we use a similar code compared to Listing 2 of Lai et al. [2023], except Line 6. We change Line 6. to data /=10. to obtain a larger variance dataset. Each dataset contains 50000 datapoints.

**The implementable algorithm.**   In this subsection, we choose two forward processes: (1) conservative $\beta_t = 1$ with $\tau = T$; (2) pure VESDE without drift term (Equation (2)) with $\sigma_t^2 = t$. To match

Table 1: The KL divergence for pure VESDE (Equation (2)) and conservative drift VESDE with different sampling method.

| Forward Process | 1-D GMM | | Swiss roll | |
|---|---|---|---|---|
| | Reverse SDE | PFODE | Reverse SDE | PFODE |
| Pure VESDE ($T = 100$) | 0.082 | 0.434 | 9.58 | 21.05 |
| Drift VESDE ($T = 100$) | 0.043 | 0.249 | 8.71 | 7.77 |
| Pure VESDE ($T = 625$) | 0.027 | 0.057 | 8.00 | 8.20 |
| Drift VESDE ($T = 625$) | **0.025** | 0.031 | 7.95 | **7.21** |

our analysis, we choose two sampling methods for the reverse process: Euler-Maruyama method for reverse SDE and RK45 ODE solver for the reverse PFODE method.

We note that although aggressive setting $\beta_t = t$ and $\tau = T$ has shown its power in theory (Lemma C.2) and the experiments with accurate score (Figure 2), other sampling issues may arise in practice. We leave the experimental exploration for drift VESDE with aggressive $\beta_t$ as a future work.

**The training detail.** For each dataset, we train a score function with pure VESDE (Equation (2), $\sigma_t^2 = t$). We train for 200 epochs with batch size 200 and learning rate $10^{-4}$. For both training and inference, the start time is $\delta = 10^{-5}$. For the conservative VESDE, we directly adapt the checkpoint learned by the pure VESDE since the conservative drift VESDE has a similar trend compared to pure VESDE, as shown in Figure 2. The above experiments are runned over 5 random seed and we present the average over these seeds in Table 1.

The above experiments are conduct on a GeForce RTX 4090. It takes 25 minutes to train a score function of pure VESDE.

**Observation.** We do experiments with $T = 100$ and lager $T = 625$ and these two choice show similar phenomenon. In this paragraph, we first use $T = 100$ as an example to discuss the results. As shown in Table 1, the conservative drift VESDE has smaller KL divergence compared to pure VESDE under all sampling methods and datasets. From Figure 1 and Figure 4, it is clear that pure VESDE has low density on the Swiss roll except the center one, which means that though pure VESDE can deal with small $\mathbb{E}[q_0]$, it is hard to deal with large dataset variance $\text{Cov}[q_0]$, as we discuss in Section 4. For conservative drift VESDE ($\beta_t = 1$ and $\tau = T$), as we discuss in Section 3.1, there is a constant decay on the prior information $\mathbb{E}[q_0]$ and $\text{Cov}[q_0]$, which is helpful in deal with large dataset mean and variance. The experimental results support our augmentation. Figure 1 (c), Figure 4 (c) and Figure 5 (c) show that the density of the generated distribution is more uniform compared to pure VESDE, which means that the drift VESDE can deal with large dataset mean and variance.

We also do experiments with larger $T = 625$. As we discuss in Section 4, larger $T$ will reduce the influence of the prior data information and have greater generated distribution, as shown in Figure 4 (c) and Figure 4 (e). The experiments of 1D-GMM (Figure 5) show a similar phenomenon compared to the multi Swiss rolls.

### G.3 The Real-World Experiments on CelebA 256

After achieving great performance under the synthetic data, we show that our conservative drifted VESDE can improve the results of pure VESDE without training.

**Setting.** In this experiment, we adapt well-known VESDE implementation [Song et al., 2020b] and do experiments on CelebA datasets (size: $256 * 256 * 3$). More specifically, we use ve/celebahq_256_ncsnpp_continuous checkpoints provided by [Song et al., 2020b] and modify the sampling process strictly according to our drifted VESDE. To do a fair comparsion, we fix the random seed and use the reverse PFODE process. Then, we generate 10000 face images to calculate the metrics. We note that when using this checkpoint and pure VESDE pipeline provided by [Song et al., 2020b], the models would generate almost pure noise with a certain probability. Hence, we use an aesthetic predictor [Schuhmann et al., 2022] (aesthetic score $\geq 5.5$) to filter the generated images to ensure that the images are clear faces.

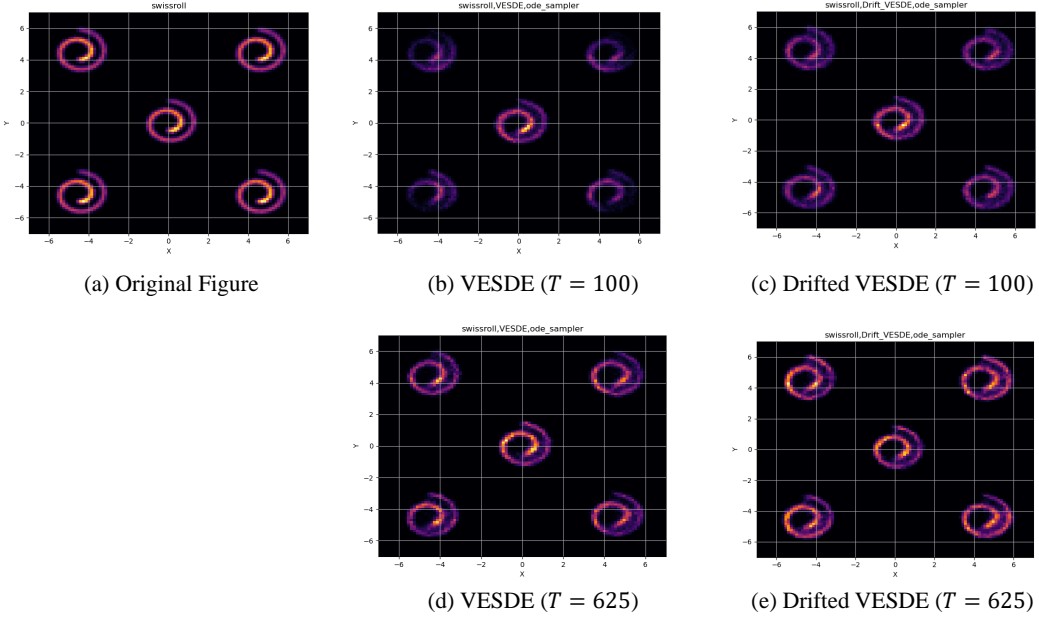

(a) Original Figure      (b) VESDE ($T = 100$)      (c) Drifted VESDE ($T = 100$)

(d) VESDE ($T = 625$)      (e) Drifted VESDE ($T = 625$)

Figure 4: Experiment results of Swiss roll with reverse PFODE

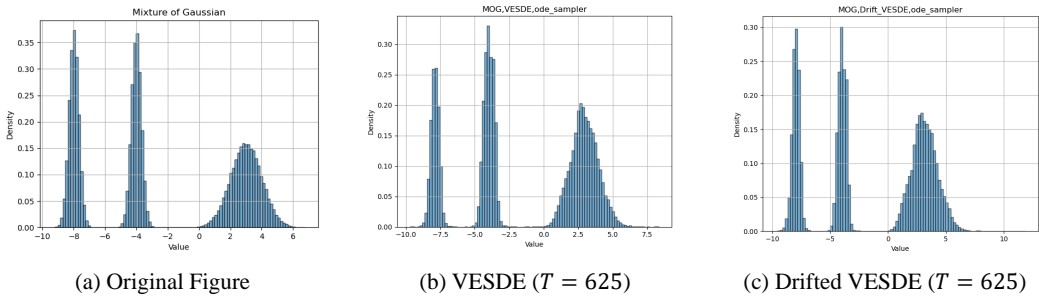

(a) Original Figure      (b) VESDE ($T = 625$)      (c) Drifted VESDE ($T = 625$)

Figure 5: Experiment results of 1D-GMM with reverse PFODE

**Discussion.** From the qualitative perspective, as shown in Figure 3 (Figure 6 and 7), the images generated by our drifted VESDE have more detail (such as hair and beard details). On the contrary, since pure VESDE can not deal with large variance, the images generated by pure VESDE appear blurry and unrealistic in these details. From the quantitative results, our drifted VESDE achieves aesthetic score 5.813, and IS 4.174, which is better than the results of baseline pure VESDE (aesthetic score 5.807 and IS: 4.082). In conclusion, the real-world experiments show the potential of our drifted VESDE.

We note that the goal of these experiments is to show that our conservative drifted VESDE is plug-and-play without training instead of achieving a SOTA performance. Hence, we focus on the relative improvement compared to the baseline [Song et al., 2020b]. There are two interesting empirical future works. For the conservative drifted VESDE, we will do experiments on the SOTA pure VESDE models [Karras et al., 2022] and improve their results without training. For the aggressive drifted VESDE, since this process makes a larger modification compared with the conservative one, we need to train a new score function instead of directly using a pre-train one to achieve better results.

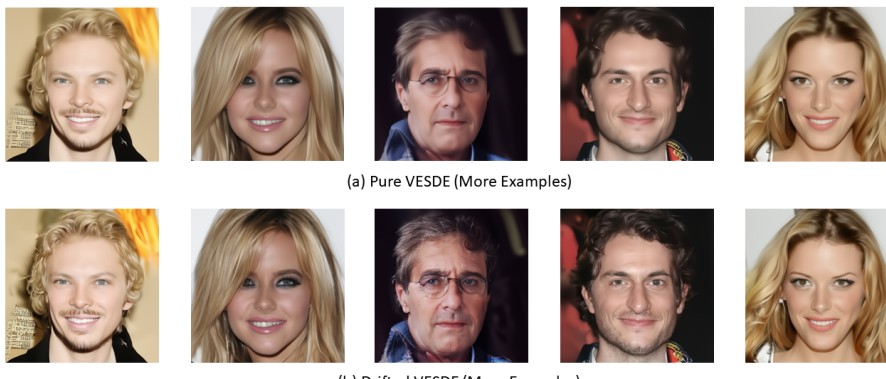

(a) Pure VESDE (More Examples)

(b) Drifted VESDE (More Examples)

Figure 6: The real-world experiments on CelebA256 dataset (More examples)

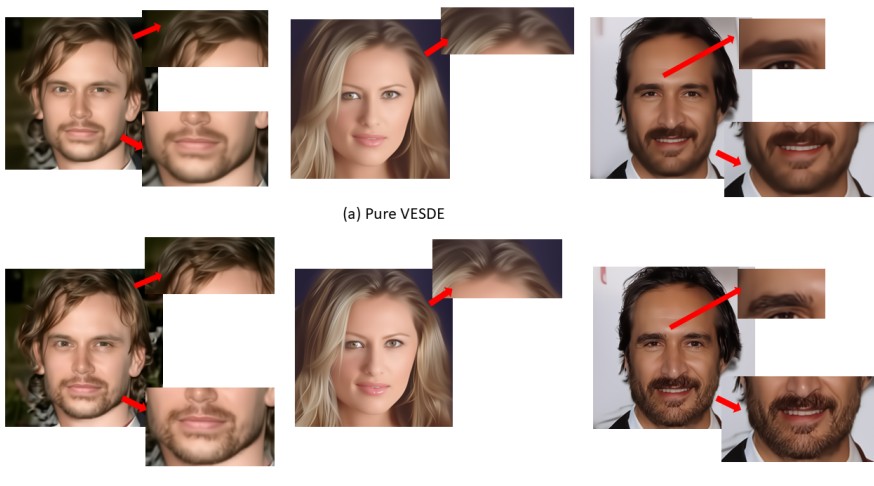

(a) Pure VESDE

(b) Drifted VESDE

Figure 7: The real-world experiments on CelebA256 dataset (Detail)

