# OpenReview forum: "Leveraging Drift to Improve Sample Complexity of Variance Exploding Diffusion Models"
_NeurIPS.cc/2024/Conference — NeurIPS 2024 poster_

### Official Review · Reviewer_bHCt · 2024-07-11

**Soundness:** 3
**Presentation:** 3
**Contribution:** 3
**Rating:** 4
**Confidence:** 3

**Summary:**

Diffusion models are powerful tools in generative modeling.  As the paper pointed out, very few theoretical works in the literature
consider variance exploding diffusion models. Among those works, forward convergence rate $1/\text{poly}(T)$ is achieved compared to $\exp(-T)$ from the variance preserving models.  This paper proposes a drifted variance exploding diffusion model that allows a faster $\exp(-T)$ forward convergence rate.  With this process, the polynomial sample complexity for a series of variance exploding models is achieved under the manifold hypothesis.  In addition to the reverse SDE, probability flow ODE is a popular alternative in the literature. The paper considers a more general setting and proves a convergence guarantee with probability flow ODE.

**Strengths:**

(1) The paper proposes a so-called drifted variance exploding forward process and to the best of my knowledge this is new.

(2) When the data is supported on a compact set, the paper manages to derives convergence guarantees in total variation distance, and 1-Wasserstein distance.

(3) The analysis seems to be rigorous.

**Weaknesses:**

(1) The model seems to be an interplay between the VE-SDE model and VP-SDE model  in the literature in the sense that when $\tau=1$, it recovers the VP-SDE model and then $\tau=\infty$, it recovers the VE-SDE model. To better compare the model with the VP-SDE and VE-SDE models in the literature, I am wondering whether it is better to assume $\tau=[1,\infty]$ and for a fixed $\tau$, to see what is the convergence guarantees and how it depends on $\tau$? If you do that, will you find the optimal range of $\tau$ to be $[T,T^{2}]$ which is what proposed in the paper? My guess is that for given $\beta_{t}$, the optimal choice of $\tau$ should depend on $\beta_{t}$ instead of being in the range $[T,T^{2}]$.

(2) Since the model proposed is a new model, it would be even more convincing if the paper can include some experiments beyond synthetic experiments.

**Questions:**

(1) You studied both total variation distance and 1-Wasserstein distance. Since your data is supported on a compact domain, would it be possible to study more general $p$-Wasserstein distance guarantees?

(2) After Assumption 3.1. you wrote that the choice $\tau\in[T,T^{2}]$ is used to guarantee the exploding variance of the forward process. This is not that accurate. I understand when $\tau$ is large, the mean-reverting effect of the forward SDE gets very weak, and the variance tends to grow. But that does not explain why you need $\tau\leq T^{2}$. I think you should also add some discussions on why you need $\tau\leq T^{2}$.

**Limitations:**

Limitations have been discussed in the conclusion section.

---

> ### Author Rebuttal · Authors · 2024-08-07
>
> Thank you for your valuable comments and suggestions. We provide our response to each question below.
>
> **Weakness 1: The universe error analysis for general $\tau$ and $\beta_t$.**
>
> In this part,  we first provide a universe complexity for general $\tau \in [1,+\infty)$ and $\beta_t\in [1,t^2]$ under the reverse SDE, which covers current diffusion models. Then, we discuss the influence of different $\tau$ and $\beta_t$ in detail.
> $$
> \begin{align}
>     \frac{\bar{D} \sqrt{m_t}}{\sigma_T}+\frac{R^2\sqrt{d}}{\sigma_\delta^4}\sqrt{\bar{\gamma}_K\beta_T\tau T}+\epsilon\_{\text{score}}\sqrt{\beta_TT},
> \end{align}
> $$
> where
> $$
> \begin{align}
> m_t=\exp \left(-\int_0^t \beta_s/\tau \mathrm{~d} s\right)\, \text{and } \sigma_t^2=\tau\left(1-m_t^2\right)\,.
> \end{align}
> $$
> (a) The general formula covers current diffusion models.
>
> In this paragraph, we show that our general formula can cover current models and provide the sample complexity. The key part of the sample complexity is balancing the first two terms: the reverse beginning and the discretization term. When $\beta_t=1$ and $\tau=1$, the drifted VESDE becomes VPSDE and $m_t=\exp{(-T)}$, which leads to a logarithmic $T$ and would not influence discretization term heavily. Then, we achieve $\tilde{O}(1/\epsilon\_{W_2}^8\epsilon\_{TV}^2)$, which has the same order with [1]. When $\beta_t=1$ and $\tau =T$, our formula is similar but slightly better (as shown in Figure 2 and our real-world experiments) to pure VESDE (\sigma_t^2= t) and we can achieve $\tilde{O}(1/\epsilon\_{W_2}^8\epsilon\_{TV}^6)$ results ([2] achieve slightly better results $1/\epsilon\_{W_2}^8\epsilon\_{TV}^4$ since they assume strong LSI assumption). When considering $\beta_t=t$ and $\tau =T^2$, the general formula is similar to pure SOTA VESDE (\sigma_t^2=t^2) and achieve the first polynomial sample complexity $\tilde{O}(1/\epsilon\_{W_2}^8\epsilon\_{TV}^6)$ under the manifold hypothesis. We also note that the above results holds for pure VESDE with $\sigma_t^2= t$ and $t^2$.
>
> (b)  When given a fixed $\beta_t$, the optimal $\tau$ has the same order with $\beta_T$.
>
> As shown in (a), pure VESDE has a worse $\epsilon_{TV}$ dependence compared to VPSDE, which comes from large reverse beginning terms (the first term). For example, when considering $\beta_t=t$ and $\tau =T^2$, $m\_T= e^{-1/2}$ and $\sigma\_T^2=(1-e^{-1})T$, which leads a polynomial $T$ and heavily influence the second discretization term. Hence, the optimal choice of $\tau=T$ instead of $T^2$. With $\tau = T$, $m\_T=\exp{(-T)}$ and the complexity is $\tilde{O}(1/\epsilon\_{W_2}^8\epsilon\_{TV}^2)$ (Thm. 5.2), which has the same with VPSDE (this result also shows that the choice of VPSDE is optimal.). For $\beta_t=t^2$, the optimal $\tau$ is $T^2$, which has the same order with $\beta_T$.
>
> We will make the above discussion clearer in the next version.
>
> **Weakness 2: The real-world experiments on CelebA 256.**
>
> We do experiments on the CelebA 256 dataset (a common face dataset) and show that our drifted VESDE can improve the results of pure VESDE **without training** from the quantitative and qualitative perspectives. Please see the experiment detail, discussion, and generated images in the **global rebuttal** part.
>
>
>
> **Q1: The general $W_p$ distance.**
>
> For the reverse SDE setting, similar to Corollary 5 of [1], by using the projection technique, we can achieve pure $W_2$ guarantee $\tilde{O}(1/\epsilon\_{W\_2}^{12})$, which has the same order with [1].
>
> For the reverse PFODE setting, similar to the tangent-based method for VPSDE [3], our results can extend to $W_p$ for any $p\ge 1$. We will make it clearer in the next version.
>
>
>
> **Q2: The choice of $\tau$.**
>
> Thanks for the helpful comment on our general formula. As shown in Weakness 1, our general formula can consider $\tau\in [1,+\infty)$ and achieve polynomial sample complexity under the reverse SDE. We also show that our formula can recover current VPSDE and pure VESDE models and go beyond with $\tau \in [1,T^2]$.
>
> For the reverse PFODE, as shown in Lem. 6.3, if considering   $\tau =1$ (the VPSDE setting), there would be an additional $\exp{(T)}$ term. To avoid this term, we need to use the variance exploding property of VESDE. Hence, we choose $\tau\in [T,T^2]$, which represents two common VESDE choices under reverse PFODE (We note that Thm. 6.2 and Coro. 6.3 also hold for pure VESDE.)
>
> We will make it clearer in the next version.
>
>
>
> [1] Chen, S., Chewi, S., Li, J., Li, Y., Salim, A., & Zhang, A. R. (2022). Sampling is as easy as learning the score: theory for diffusion models with minimal data assumptions. *arXiv preprint arXiv:2209.11215*.
>
> [2] Lee, H., Lu, J., & Tan, Y. (2022). Convergence for score-based generative modeling with polynomial complexity. *Advances in Neural Information Processing Systems*, *35*, 22870-22882.
>
> [3] De Bortoli, V. (2022). Convergence of denoising diffusion models under the manifold hypothesis. *arXiv preprint arXiv:2208.05314*.

---

> > ### Author Response · Authors · 2024-08-13
> >
> > We thank you once again for your careful reading of our paper and your constructive comments and suggestions. As the discussion period approaches its end,  we will appreciate it very much if you could let us know whether all your concerns are addressed. We are also more than happy to discuss our work and answer any further questions.

---

> ### Author Response · Authors · 2024-08-11
>
> Thanks again for your insightful suggestions and comments! According to your helpful comments, we improve our work from the empirical and theoretical perspectives. From the empirical perspective, we do experiments on the real-world CelebA 256 dataset and show that our drifted VESDE is a plug-and-play method without training. More specifically, the images generated by our drifted VESDE are more detailed than those of the pure VESDE baseline (shown in the PDF of the global rebuttal). From the theoretical perspective, we show that our drifted VESDE covers common diffusion models (including VP and VESDE) and goes beyond. More details and discussions are shown in the rebuttal part. We will add the above discussion to our next version and are more than happy to answer any further questions.

---

### Official Review · Reviewer_Duca · 2024-07-12

**Soundness:** 3
**Presentation:** 2
**Contribution:** 3
**Rating:** 7
**Confidence:** 2

**Summary:**

The paper analyzes the Variance Exploding diffusion model under the manifold hypothesis. By a slight modification to the VESDE process, the authors propose a method whose convergence guarantees are better than prior best known rates in this regime.

**Strengths:**

The rate obtained is state-of-the-art for Variance Exploding models, which are notorious for being difficult to analyze.

Although the paper borrows parts of its analytic framework from the work of Bortoli et al. (2022), the adaptation of the analysis in this context appears to be non-trivial.

The paper is clearly written and the exposition of the proof is overall quite clear.

**Weaknesses:**

The experiments in the main text and appendix are not particularly thorough. This is acceptable since the primary claimed contribution by this article is theoretical, but some kind of practical benchmark might also be beneficial.

The dimension and diameter dependence are quite severe. Is there any sense of how close these parameters are to their optimal values?

**Questions:**

The final rates are only given in total variation. Is it possible to improve the metric to KL or to state W2 bounds with similar complexities? What is the analytical challenge otherwise?

Theorems 6.1 and Corollary 6.2 could have their exposition simplified in the main text; the intuition provided in the subsequent section is much clearer.

Minor:

The font used for "KL" is not consistent in the equations in the appendix.

97: about data -> about the data

102: score function -> the score function

107: with strong LSI -> with a strong LSI

108: assume the Lipschitz score -> assume the score is Lipschitz

108: first work focus -> first work to focus

138: introduces -> introduce

187: "reversing" does not make sense here. Perhaps "initial distribution of the reverse process" is meant

216: "does reverse" -> "which reverse the process"

345: "support" -> "supports"

364: "support our", "show that" -> "supports our", "shows that"

**Limitations:**

The authors have already addressed all major limitations of their work.

---

> ### Author Rebuttal · Authors · 2024-08-07
>
> Thank you for your valuable comments and suggestions. We provide our response to each question below.
>
> **W1: The real-world experiments on CelebA 256.**
>
> We do experiments on the CelebA 256 dataset (a common face dataset) and show that our drifted VESDE can improve the results of pure VESDE **without training** from the quantitative and qualitative perspectives. Please see the experiment detail, discussion, and generated images in the **global rebuttal** part.
>
> **W2: The discussion on the diameter.**
>
> Since there is no existing work to discuss the lower bound of the diameter of VE-based models, we start from the VP-based models and discuss the improvement space of VE-based models to achieve the same order results compared with VP-based models.
>
> For the reverse SDE, VP-based models achieve an optimal $d$ dependence by using the stochastic localization technique and exponential-decay stepsize [1]. As a first step, we use a uniform stepsize for VE-based models, which leads to a slightly worse dependence on $R$ and $d$. It is an interesting future work to use a more refined, time-dependent stepsize to achieve improved results for VESDE and show that VESDE performs better than VPSDE from the theoretical perspective.
>
> For the reverse PFODE,  since the Wasserstein distance can not use the data processing inequality, Thm. 6 has an exponential dependence on $R$. As discussed at the end of Sec. 6, it is possible to introduce a suitable corrector (such as the Underdamped Langevin process in [2]) to inject some small noise into the PFODE predictor, which allows the use of the data processing inequality and replaces the exponential $R$ with a polynomial one.
>
> We will add a discussion paragraph on the diameter to make it clearer.
>
> **Q1: The guarantee under stronger metric.**
>
> (a) The pure $W_2$ guarantee.
>
> Since we assume the manifold hypothesis, we first show how to obtain a pure $W_2$ guarantee at each setting. For the reverse SDE setting, similar to Corollary 5 of [3], by using the projection technique, we can achieve pure $W_2$ guarantee $\tilde{O}(1/\epsilon\_{W\_2}^{12})$, which has the same order with [3]. We note that the slightly worse $\epsilon$ dependence of our work and [3]  is due to  the relationship between $W_2$ and $\mathrm{TV}$ [4]:
>
> $$
> W\_2(R\_K^{q^{\tau}_{\infty}},q\_{\delta})\leq R\sqrt{\mathrm{TV}(R\_K^{q^{\tau}\_{\infty}},q\_{\delta})}+R\exp{(-R)}.
> $$
> For the reverse PFODE setting, similar to the tangent-based method for VPSDE [5], our results can extend to $W_p$ for any $p\ge 1$.
>
> (b) The $\mathrm{KL}+W_2$ guarantee.
>
> When considering reverse SDE setting, similar to [6], we can use the chain rule of $\mathrm{KL}$ divergence instead of the triangle inequality to obtain a $\mathrm{KL}+W_2$ guarantee $\tilde{O}(1/\epsilon_{\mathrm{KL}}^2\epsilon\_{W\_2}^8)$.
>
> We will discuss the guarantee under stronger metrics in detail.
>
> **Q2 and Minor Question: the presentation.**
>
> Thanks for your helpful comments on the presentation. For our tangent-based unified framework part, we will simplify the formula of Thm. 6.1 and Coro. 6.2 and highlight the technique novelty. For the typos, we will polish our presentation according to your comments.
>
>
>
> [1] Benton, J., De Bortoli, V., Doucet, A., & Deligiannidis, G. (2023). Linear convergence bounds for diffusion models via stochastic localization. *arXiv preprint arXiv:2308.03686*.
>
> [2] Chen, S., Chewi, S., Lee, H., Li, Y., Lu, J., & Salim, A. (2024). The probability flow ode is provably fast. *Advances in Neural Information Processing Systems*, *36*.
>
> [3] Chen, S., Chewi, S., Li, J., Li, Y., Salim, A., & Zhang, A. R. (2022). Sampling is as easy as learning the score: theory for diffusion models with minimal data assumptions. *arXiv preprint arXiv:2209.11215*.
>
> [4] Rolland, P. T. Y. (2022). *Predicting in uncertain environments: methods for robust machine learning* (No. 9118). EPFL.
>
> [5] De Bortoli, V. (2022). Convergence of denoising diffusion models under the manifold hypothesis. *arXiv preprint arXiv:2208.05314*.
>
> [6] Chen, H., Lee, H., & Lu, J. (2023, July). Improved analysis of score-based generative modeling: User-friendly bounds under minimal smoothness assumptions. In *International Conference on Machine Learning* (pp. 4735-4763). PMLR.

---

> > ### Comment · Reviewer_Duca · 2024-08-13
> >
> > Thank you for your detailed response. In light of the numerous improvements shown both in your response to me and to the other reviewers, I will raise my score by 1.

---

> > > ### Author Response · Authors · 2024-08-13
> > >
> > > Thank you for your positive feedback and support! We will add the discussion and polish our presentation according to your comments. In case you have any other questions, please don't hesitate to let us know.

---

### Official Review · Reviewer_a6C8 · 2024-07-12

**Soundness:** 3
**Presentation:** 3
**Contribution:** 3
**Rating:** 6
**Confidence:** 2

**Summary:**

This paper focuses on variance exploding (VE) based diffusion models and proposes a drifted VESDE forward process with an unbounded diffusion coefficient. This choice of coefficients allows an exponential-decay forward convergence rate, and the authors establish the first polynomial sample complexity for VE-based models with reverse PFODE. Moreover, the authors propose a tangent-based unified analysis framework with reverse SDE and PFODE and prove the first quantitative guarantee for SOTA VE-based models with reverse PFODE.

**Strengths:**

In terms of originality, this paper proposes a new variance exploding (VE) based diffusion model and establishes the corresponding convergence guarantees. The theoretical results are solid. Moreover, this paper is well-organized and clearly written.

**Weaknesses:**

1. The convergence guarantees for VE-based models with reverse PFODE are relatively weak. For example, in Assumption 3.1, the choice of $\beta_t$ is more conservative for reverse PFODE; In Theorem 6.2 part (2), the last term is $\bar{D}/\tau$ instead of $\bar{D} e^{-T/2} / \sqrt{\tau}$ in part (1). Does this mean that the $e^{-T}$ forward convergence rate can only achieved by the reverse SDE?

2. The numerical results only include synthetic experiments.

**Questions:**

1. In Assumption 3.1, why for reverse SDE and PFODE, the choices of $\beta_t$ are different? Could the authors provide some intuitive explanation?

2. In Theorem 5.2, the sample complexity has the same dependence on $\epsilon_{W_2}$ and $\epsilon_{TV}$ as that in Chen et al. (2023c). However, the dependence on $d$ is worse. Is it a consequence of a more aggressive $\beta_t$?

3. Could the theoretical results reflect the superiority of VE-based diffusion models over VP-based diffusion models?

**Limitations:**

The authors discuss future work and limitations in Section 8.

---

> ### Author Rebuttal · Authors · 2024-08-07
>
> Thank you for your valuable comments and suggestions. We provide our response to each question below.
>
> **W1 & Q1: The different $\beta_t$ for reverse SDE and PFODE: the balance between different error terms.**
>
> (a) We first recall the reverse beginning error term when considering the unified tangent-based framework:
> $$
> \begin{align}
> W_1\left(Q_{t_K}^{q_{\infty}^{\tau}},Q_{t_K}^{q_0P_T}\right)\leq \frac{\sqrt{m_T}\bar{D}}{\sigma_T}\exp\left(\frac{R^2}{2\sigma_{T-t_K}^{2}}+ \frac{(1-\eta^2)}{2}\int_0^{t_K}\frac{\beta_{T-u}}{\tau}\mathrm{d}u\right)\,.
> \end{align}
> $$
> We note that the forward and reverse processes determine the above bound simultaneously, where the exponential terms come from the bound of the tangent process (Lem. 6.3, reverse process), and the first part comes from Thm. 4.2 (forward process).
>
> For the reverse SDE ($\eta=1$), the exponential term becomes $\exp\left(\frac{R^2}{2\sigma_{T-t_K}^{2}}\right)$, which is independent with $\beta\_t$. Hence, choosing an aggressive $\beta_t=t^2$  (here we use $\tau =T^2$ as an example) will introduce $\frac{\bar{D}e^{-T/2}}{\tau} \exp\left(\frac{R^2}{2\sigma\_{T-t_K}^{2}}\right)$ without introducing additional terms.
>
> For the reverse PFODE ($\eta=0$), we can also choose an aggressive $\beta_t=t^2$, which leads to a $\bar{D}e^{-T/2}/\tau$ for $\frac{\sqrt{m_T}\bar{D}}{\sigma_T}$. However, since $\eta = 0$, $\exp{\int_0^{t_K}\frac{\beta\_{T-u}}{\tau}\mathrm{d}u}$ term  will introduce an additional $e^{\frac{T}{6}}$ to the bound of the tangent process. **We note that the key part of the convergence guarantee is to balance the discretization, reverse beginning, and early stopping error terms.** Though the reverse beginning still enjoys an $e^{-T/4}\exp\left(\frac{R^2}{2\sigma\_{T-t_K}^{2}}\right)$, the aggressive $\beta_t$ will introduce an additional  $e^{\frac{T}{6}}$ in the final result (since the tangent process also influences the discretization error term.). Hence, a better choice for the reverse PFODE is a conservative $\beta_t$.
>
> (b) An interesting future work: the PFODE predictor and suitable corrector.
>
> The above discussion shows that since the reverse beginning error is determined by the forward and reverse process at the same time, an aggressive $\beta_t$ can not be used under the PFODE setting. The complex dependency is due to the fact that data processing inequality is forbidden when considering the Wasserstein distance. As a next step, we discuss how to improve the results of Thm. 6.1 with an aggressive $\beta_t$ (the PFODE setting).
>
> We first recall the data processing inequality: Consider a channel that produces $Y$ given $X$  based on the law $P_{Y \mid X}$. Let $P_Y$ be the distribution of $Y$ when $X$ is generated by $P_X$ and $Q_Y$ be the distribution of $Y$ when $X$ is generated by $Q_X$. Then we know that for any $f$-divergence $D_f(\cdot \| \cdot)$, $D_f\left(P_Y \| Q_Y\right) \leq D_f\left(P_X \| Q_X\right)$.
>
> When choosing $f(x)=\frac{1}{2}|x-1|$, the $f$-divergence is TV distance. Hence, by viewing $Q_{t_K}$ as the channel, the inequality $\operatorname{TV}\left(Q\_{t_K}^{q\_{\infty}^\tau}, Q\_{t_K}^{q_T^\tau}\right)\leq \operatorname{TV}\left(q\_T^\tau, q\_{\infty}^\tau\right)$ holds, which indicates that for the $\mathrm{TV}$ distance, the influence of reverse process can be ignored when considering the reverse beginning error term. However, the data processing inequality does not hold for Wasserstein distance. To overcome this problem, an interesting future work is to introduce a suitable corrector (such as the Underdamped Langevin process in [1]) to inject some small noise into the PFODE predictor, which allows the use of the data processing inequality and achieve a polynomial sample complexity.
>
> We will add a discussion to make it clearer.
>
> **W2: The real-world experiments on CelebA 256.**
>
> We do experiments on the CelebA 256 dataset (a common face dataset) and show that our drifted VESDE can improve the results of pure VESDE **without training** from the quantitative and qualitative perspectives. Please see the experiment detail, discussion, and generated images in the **global rebuttal** part.
>
> **Q2: The dependence on $d$ and $R$.**
>
> We recall that the result shown in [2] is $\tilde{O}\left(\frac{d^3 R^4(R \vee \sqrt{d})^4}{\varepsilon_{\mathrm{TV}}^2 \varepsilon_{W_2}^8}\right)$ and our results is $\tilde{O}\left(\frac{dR^4(d+R\sqrt{d})^4}{\epsilon\_{W_2}^{8}\epsilon_{\text{TV}}^2}\right)$. Since the image datasets are usually normalized into $[-1,1]^d$, $R\leq \sqrt{d}$. Hence, our results have the same order as [2]. We will add a discussion part about the dependence on $d$ and $R$.
>
> **Q3: The superiority of VE-based models over VP-based models.**
>
> When considering the reverse PFODE, we show the superiority of VE-based models over VP-based models. More specifically, Lem. 6.3 contains $\exp{(\int_{0}^{t_K}\beta\_{T-u}/\tau du)}$ for the reverse PFODE. For VPSDE ($\beta_t =1$ and $\tau =1$), there is an additional $\exp{(T)}$. On the contrary, VE-based models make use of the variance exploding property of VESDE, avoid this term (for example, the above term is a constant for $\beta\_t=t$ and $\tau = T^2$), and achieve polynomial $T$ dependence in final results. We note that our results also hold for VESDE ($\sigma\_t^2= t^2$), the SOTA models proposed by [3]. We will add a discussion part in the next version.
>
>
>
> [1] Chen, S., Chewi, S., Lee, H., Li, Y., Lu, J., & Salim, A. (2024). The probability flow ode is provably fast. *Advances in Neural Information Processing Systems*, *36*.
>
> [2] Chen, S., Chewi, S., Li, J., Li, Y., Salim, A., & Zhang, A. R. (2022). Sampling is as easy as learning the score: theory for diffusion models with minimal data assumptions. *arXiv preprint arXiv:2209.11215*.
>
> [3] Karras, T., Aittala, M., Aila, T., & Laine, S. (2022). Elucidating the design space of diffusion-based generative models. *Advances in neural information processing systems*, *35*, 26565-26577.

---

> > ### Comment · Reviewer_a6C8 · 2024-08-12
> >
> > Thank you for your detailed response.
> >
> > I have an additional concern. It seems that Theorem 6.1 implies an exponential dependence of sample complexity on the inverse of the target accuracy. Is it correct to understand Section 6 as providing a more general analysis of VE-based models to encompass the reverse PFODE, whereas the analysis in Section 5 is more refined for the reverse SDE?
> > Additionally, could the authors elaborate on the dependence of the sample complexity on the target accuracy for the reverse PFODE under the extra assumption in Corollary 2?

---

> > > ### Author Response · Authors · 2024-08-12
> > >
> > > Thanks for your effort and time. We discuss each question in detail below.
> > >
> > > (Question 1) The understanding in the response is correct. Section 6 provides a unified tangent-based framework for the VE-based models (including reverse SDE and PFODE). Hence, we achieve a slightly worse convergence guarantee compared to the results in Section 5 when considering the reverse SDE. In Section 5, we focus on the reverse SDE setting and give a more refined analysis. The more refined analysis shows that our drifted VESDE can balance different error terms and achieve better complexity results compared to pure VESDE. We will add the above discussion and polish our presentation according to your helpful comments.
> > >
> > >
> > >
> > > (Question 2) In this part, we provide the sample complexity of Corollary 6.2. More specifically, by choosing  $\delta \leq \epsilon_{W_1}^2/d$, $T\ge \frac{\bar{D}\exp{(\Gamma)}\beta^{\Gamma/2}}{\delta^{\Gamma}\epsilon\_{W_1}}$ and $\bar{\gamma}\_K\leq \frac{\epsilon\_{W_1}^2\delta^{2\Gamma}}{C_2^2(\tau)\kappa_2^4(\tau)T^2\exp{(2\Gamma)}\beta^{\Gamma}}$, we have $W_1\left(R_K^{q_{\infty}^\tau}, q_0\right)\leq \epsilon_{W_1}$ with the sample complexity
> > > $$
> > > K\leq \frac{\bar{D}\exp{(3\Gamma)}\beta^{3\Gamma/2}C_2^2(\tau)\kappa_2^4(\tau)T^2}{\delta^{3\Gamma}\epsilon_{W_1}^3}.
> > > $$
> > > We note that compared with Thm. 6.1, the above complexity replace the exponential dependence on $\delta$ by a polynomial $\delta$ and an exponential $\Gamma$. Since $\delta$ is related to the $\epsilon_{W_1}$ and $\Gamma$ is only determined by the data structure, this result improves Thm.6.1. As discussed at the end of Section 6, an interesting future work is to introduce a suitable corrector to inject some small noise to the PFODE sampler and achieve a polynomial sample complexity (w.r.t all problem parameters) under the manifold hypothesis. We will add the above result and discuss it in Section 6.
> > >
> > > We hope the above discussion can address your concerns. We are more than happy to discuss our work in detail and answer any further questions in the rebuttal phase.

---

> > > > ### Comment · Reviewer_a6C8 · 2024-08-12
> > > >
> > > > Thank you for your detailed response again. I will raise my score to 6.

---

> > > > > ### Author Response · Authors · 2024-08-12
> > > > >
> > > > > Thank you for your positive feedback and support! In case you have any other questions, please don't hesitate to let us know.

---

### Official Review · Reviewer_ZwJR · 2024-07-13

**Soundness:** 3
**Presentation:** 3
**Contribution:** 3
**Rating:** 6
**Confidence:** 4

**Summary:**

In this paper, the authors propose an analysis of the convergence of diffusion models under the manifold hypothesis in a similar setting as [1]. The main contribution is the extension of the analysis to the case of VESDE (Variance Exploding SDE) contrary to [1] which is limited to VPSDE (Variance Preserving). The rates obtained by the authors are better than the ones obtained in [1] (although in a different context). They also extend their analysis to ODE samplers which are notably more difficult to deal with than SDE samplers from a theoretical point of view. The decomposition of the error is the same as in [1] but with a more careful analysis of the tangent process (see Section 6 "The Tangent-based Analysis Framework"). Experiments in toy settings are presented.

[1] De Bortoli -- Convergence of denoising diffusion models under the manifold hypothesis

**Strengths:**

* These results are the first results obtained for the convergence of diffusion models in the VESDE setting under the manifold hypothesis.

* The analysis of the tangent process represents an improvement over the results of [1]. This is an interesting development in itself.

* The introduction of the drifted VESDE (Equation 5) is interesting and represents a good avenue for future studies of the VESDE process.


[1] De Bortoli -- Convergence of denoising diffusion models under the manifold hypothesis

**Weaknesses:**

* Experiments are only toyish. I actually don't think they benefit the paper. This is mostly theoretical work and I'm struggling to understand what point is made here. If this is to illustrate the validity of the samplers this is already well established. If the point of the experiment is to illustrate the benefit of drifted VESDE then I would have appreciated a more challenging setting (like CIFAR10 or Imagenet in image processing or larger models like Stable Diffusion). This does not require pretraining a large model since this is a modification of the sampler.

* There is actually not a lot of discussion on how drifted VESDE relate to VESDE. Can one obtain convergence results for  VESDE (classical) based on drifted VESDE?

* As of now, the analysis is limited to drifted VESDE. It would be interesting to analysis if the improved results can transfer to the VPSDE framework and improve on the results of [2].

* The paper is easy to follow and clearly presented.

[1] Karras et  al. -- Elucidating the Design Space of Diffusion-Based Generative Models

[2] De Bortoli -- Convergence of denoising diffusion models under the manifold hypothesis

**Questions:**

* l.34 "Furthermore, Karras et al. [2022] unify two processes and show that the optimal parameters of the general formula correspond to VESDE." --> Not clear what the authors are referring to here.

* l.48 "leads to a large reverse beginning error" --> I disagree as one could argue that most of the error in diffusion models arises from the approximation of the score. The "large reverse beginning error" is a strong statement here.

* One relevant work that is not discussed is [1]

* l.296 "Furthermore, we emphasize that our tangent-based unified framework is not a simple extension of Bortoli [2022]." --> Can the authors provide more details here?

* Since there is a one-to-one mapping between VESDE and VPSDE could the authors have leveraged this connection? There is also a connection with Stochastic Localization as pointed out by [2]. Below we explicit the connection between VESDE and VPSDE.

Assume that $(X_t)_{t \geq 0}$ satisfies a VESDE $\mathrm{d} X_t = g(t) \mathrm{d} B_t$ then we have that $(Y_t)_{t \geq 0}$ given for any $t \geq 0$ by $Y_t = \exp[F(\phi(t))] X_{\phi(t)}$ satisfies $\mathrm{d} Y_t  = F(\phi)'(t) Y_t \mathrm{d}_t  + \exp[F(\phi(t))] \phi'(t)^{1/2} \mathrm{d} B_t$.

[1] Conforti et al. -- "Score diffusion models without early stopping: finite Fisher information is all you need"

[2] Montanari -- "Sampling, Diffusions, and Stochastic Localization"

**Limitations:**

Limitations are not really addressed in Section 8 ("Conclusion") in the paragraph "Future Work and Limitation". I think a more in depth discussion of the benefits of VESDE and VPSDE is needed.

---

> ### Author Rebuttal · Authors · 2024-08-07
>
> Thanks for your valuable comments and suggestions. We provide our response to each question below.
>
> **W1: The real-world experiments.**
>
> We do experiments on the CelebA 256 and show that our drifted VESDE improves pure VESDE **without training** from the quantitative and qualitative perspectives. Please see the experiment details in the **global rebuttal**.
>
> **W2: The link between drifted VESDE and VESDE and universe error analysis.**
>
> (i) For reverse PFODE, our analysis holds for pure VESDE with $\sigma_t^2=t$ or $t^2$, which covers the SOTA VESDE.
>
> (ii) For reverse SDE, we extend our general drifted VESDE formula and provide a universe complexity for general $\tau \in [1,+\infty)$ and $\beta_t\in [1,t^2]$.
>
> $$
> \begin{align}
>     \frac{\bar{D} \sqrt{m_t}}{\sigma_T}+\frac{R^2\sqrt{d}}{\sigma_\delta^4}\sqrt{\bar{\gamma}_K\beta_T\tau T}+\epsilon\_{\text{score}}\sqrt{\beta_TT},
> \end{align}
> $$
> where $m_t=\exp \left(-\int_0^t \beta_s/\tau \mathrm{~d} s\right)$ and $\sigma_t^2=\tau\left(1-m_t^2\right)$.
>
> (a) The general formula covers current models (including VE and VPSDE).
>
> When $\beta_t=1$ and $\tau=1$, the drifted VESDE becomes VP and $m_t=\exp{(-T)}$, which leads to a logarithmic $T$ and achieve $\tilde{O}(1/\epsilon\_{W_2}^8\epsilon\_{TV}^2)$ (the same with [1]). When $\beta_t=1$ and $\tau =T$, our formula is similar but slightly better (Fig. 2 and real-world experiments) to pure VESDE (\sigma_t^2= t) and achieves $1/\epsilon\_{W_2}^8\epsilon\_{TV}^6$ results ([2] achieve $1/\epsilon\_{W_2}^8\epsilon\_{TV}^4$ since they assume strong LSI holds). For $\beta_t=t$ and $\tau =T^2$, the formula is similar to SOTA  pure VESDE ($\sigma_t^2=t^2$) and achieves the first polynomial results $1/\epsilon\_{W_2}^8\epsilon\_{TV}^6$ under the manifold hypothesis. We also note that the above results holds for pure VESDE with $\sigma_t^2= t$ and $t^2$.
>
> (b)  Go beyond: When given a $\beta_t$, the optimal $\tau$ has the same order with $\beta_T$ for reverse SDE.
>
> The key part of analysis is balancing the reverse beginning and discretization error (please see approximated score in Q2). As in (a), pure VESDE has a worse $\epsilon_{TV}$ than VP, which comes from the large reverse beginning term. For example, if $\beta_t=t$ and $\tau =T^2$, $m\_T= e^{-1/2}$ and $\sigma\_T^2=(1-e^{-1})T$, which leads a polynomial $T$ and heavily influence the discretization term. Hence, the optimal choice of $\tau=T$ instead of $T^2$ and $m\_T=e^{-T}$. Then, we achieve the same guarantee with VPSDE [1]. Similarly, the optimal $\tau$ is $T^2$ for $\beta_t=t^2$ (Thm. 5.2).
>
> **W3: The improved results for [3].**
>
> [3] considers VPSDE with the reverse SDE and achieves a guarantee with an exponential term $\exp{(1/\delta)}$. [1] achieve a pure $W_2$ guarantee $1/\epsilon_{W_2}^{12}$ by using a projection technique=, which is a direct improvement of [2]. Our general formula can also recover this result (W1), although the main point of our work is not VPSDE.
>
> **Q1: Discussion on Karras et al.**
>
> This work unifies the reverse PFODE of VP and VESDE (Eq. 4 of their work) and proves that the ODE solution trajectory of VESDE ($\sigma_t^2=t^2$) is linear and directly towards the data manifold (Fig. 3 of their work). On the contrary, the trajectories of VP and VESDE ($\sigma_t^2=t$) are not linear in most regions, which makes the denoise process difficult.
>
> **Q2: The error terms.**
>
> The reverse beginning, discretization, and approximated score error are both important. Since the sampling and learning process are relatively independent, current works usually decouple these parts. For the sample process, previous works assume an $L_2$ accuracy score [1] [3]. For the learning process, some works analyze how to use a NN to learn score [4]. We will add a discussion about error terms.
>
> **Q3: The discussion of Conforti et al.**
>
> This work considers VPSDE with reverse SDE under the finite fisher information assumption. Though this work relies heavily on the property of OU process, it is an interesting future work to analyze whether the connection in Q5 can be used to improve the results of VESDE. We also note that our work analyzes a broader area (reverse SDE and PFODE). We will add a discussion.
>
> **Q4: The novelty of our tangent-based method.**
>
> The technique novelty is our tangent-based lemma. For the PFODE, Lem. 6.3 contains $\exp{(\int_{0}^{t_K}\beta_{T-u}/\tau du)}$. For VPSDE ($\beta_t =1$ and $\tau =1$), there is an additional $\exp{(T)}$, which indicates that the previous lemma can't deal with reverse PFODE even under the VPSDE setting. To avoid this term, we use the variance exploding property of VESDE (for example, the above term is a constant for $\beta_t=t$ and $\tau = T^2$) and achieve polynomial $T$ in final results.
>
> **Q5: The connection between VP and VESDE.**
>
> As shown in Montanari, VE and VPSDE are equivalent to a change of time, which indicates the discretization analysis of these models is similar under the reverse SDE. Our universal analysis (W2) also reflects this phenomenon. However, the other key point is the balance of the first two terms, and pure VESDE performs badly. Hence, we further propose the general drifted VESDE formula, prove the optimal choice of $\tau$ faces a $\beta_t$, and provide better results. We will make the discussion of W1 and Q5 clearer.
>
> **Limitation.**
>
> Thanks for the comments on Limitation. We will discuss the benefit of our drifted VESDE formula (including VPSDE and pure VESDE) and its potential to achieve the SOTA performance.
>
> [1] Chen et al. Sampling is as easy as learning the score: theory for diffusion models with minimal data assumptions. ICLR 2023.
>
> [2] Lee et al. Convergence for score-based generative modeling with polynomial complexity. NeurIPS 2022.
>
> [3] De Bortoli, V. Convergence of denoising diffusion models under the manifold hypothesis. TMLR.
>
> [4] Chen et al.. Score approximation, estimation and distribution recovery of diffusion models on low-dimensional data. ICML 2023.

---

> > ### Comment · Reviewer_ZwJR · 2024-08-12
> >
> > Thank you for your answer and the additional experiments. I would like to keep my score to  6.
> >
> > I would also like to point out that I think there was a misunderstanding regarding W2.
> > When I am talking about VESDE (Brownian motion) I am talking about the following dynamics.
> >
> > $$ \mathrm{d} \mathbf{X}_t = g(t) \mathrm{d} \mathbf{B}_t . $$
> >
> > To the best of my knowledge this is not described by (3). I have no doubt that VPSDE can be recovered. Regarding VESDE I am less sure. For example in the bound provided by the authors in the rebuttal to get to VESDE (I insist on having a zero drift), I need $\tau \to +\infty$. Unless, I am mistaken, this means that the bound provided by the authors blow up.
> >
> > This is the discussion I was asking for.
> >
> > I also think that the authors missed my point in Q2. My point was that improving on the discretization errors in SDE models might not be the right term to look at. Indeed, choosing VPSDE, VESDE (drifted or not), implies different sampling bounds (as illustrated by this work), but some of these choices also affect the learning of the score. It is hard to disentangle those parts.
> >
> > Regarding Q1, I don't think that Karras et al. "prove" the optimality of VESDE (and a reference to a Figure 3 is not sufficient).

---

> > > ### Author Response · Authors · 2024-08-12
> > >
> > > Thanks again for your effort in reading our real-world experiments and rebuttal. We discuss each question in detail below.
> > >
> > > (Question 1.) As shown at the end of W2 (a), we can obtain a polynomial sample complexity for the pure VESDE (with zero drift). We use pure VESDE ($\sigma_t^2=t^2$) as an example
> > > $$
> > > \mathrm{d} \mathbf{X}_t=\sqrt{2t} \mathrm{d} \mathbf{B}_t.
> > > $$
> > > In this case, the convergence guarantee is
> > > $$
> > > \begin{align}
> > >     \frac{\bar{D} }{T}+\frac{R^2\sqrt{d}}{\delta^4}\sqrt{\bar{\gamma}_KT^4}+\epsilon\_{\text{score}}\sqrt{T^2},
> > > \end{align}
> > > $$
> > > whose bound does not blow up. However, the above bound still has difficulty in balancing the reverse beginning and discretization error term. Hence, we propose our drifted VESDE to balance these two terms.
> > >
> > > For the drifted VESDE, as mentioned in your response,  the formula can not recover pure drifted VESDE by setting $\tau \rightarrow +\infty$, and we need the above independent theorem to give a guarantee for the pure VESDE. However, we also note that with a conservative $\beta_t$ (for example, $\beta_t=t$ when $\tau =T^2$), the performance of conservative VESDE is similar but better than pure VESDE (The red and brown line of our Sec. 7.1, Fig 2 and our real-world experiments). Hence, we present the sample of drifted VESDE with reverse SDE for the sake of coherence. We will add the above results of pure VESDE as an independent theorem and make our presentation clearer according to your comments in the next version.
> > >
> > > (Question 2.) The learning process of the score function is an important part of the analysis of diffusion models. However, when considering the sample complexity, most current theoretical works assume a $L_2$ accuracy score function [1] [2] [3] [4] [5], and we follow this standard assumption in our work. In this work, we take the first step in analyzing the great performance of VE-based models from the sample complexity perspective. As mentioned in the response, the choice of different processes will influence the score learning process. Hence, an end-to-end analysis (considering the sampling and learning process simultaneously) for the VE-based models is a really interesting future work, and we will add a detailed discussion in our future paragraph.
> > >
> > > (Question 3.) When discussing Karras et al., we want to show that the linear solution trajectory of VESDE ($\sigma_t^2=t^2$) is more friendly compared with the one of VPSDE when considering the sampling phase. The great performance has been shown in many areas, such as the one-step consistency model (including the follow-up works) [6]  and video generation models [7]. For the consistency models, they use the forward process proposed by Karras et al. due to the linear trajectory. For the Stable Video Diffusion, they also use the noise schedule to obtain a pre-trained base model (Sec. 4.1 of their paper). We will improve our presentation according to your comments and add the above discussion to our introduction paragraph.
> > >
> > >
> > >
> > > [1] Chen et al. Sampling is as easy as learning the score: theory for diffusion models with minimal data assumptions. ICLR 2023.
> > >
> > > [2] Chen, S., Chewi, S., Lee, H., Li, Y., Lu, J., & Salim, A. (2024). The probability flow ode is provably fast. *Advances in Neural Information Processing Systems*, *36*.
> > >
> > > [3] De Bortoli, V. Convergence of denoising diffusion models under the manifold hypothesis. TMLR.
> > >
> > > [4] Benton, J., Bortoli, V. D., Doucet, A., & Deligiannidis, G. (2024). Nearly d-linear convergence bounds for diffusion models via stochastic localization.
> > >
> > > [5] Lee et al. Convergence for score-based generative modeling with polynomial complexity. NeurIPS 2022.
> > >
> > > [6] Song, Y., Dhariwal, P., Chen, M., & Sutskever, I. (2023). Consistency models. *arXiv preprint arXiv:2303.01469*.
> > >
> > > [7] Blattmann, A., Dockhorn, T., Kulal, S., Mendelevitch, D., Kilian, M., Lorenz, D., ... & Rombach, R. (2023). Stable video diffusion: Scaling latent video diffusion models to large datasets. *arXiv preprint arXiv:2311.15127*.

---

### Author Rebuttal · Authors · 2024-08-07

# The Real-World Experiments and Discussion (CelebA 256)

Once again, we thank all reviewers for their valuable suggestions on real-world experiments. In this part, we show that our conservative drifted VESDE can improve the quantitative results (IS (higher is better), and Aesthetic score [1] (1-10, higher is better)) of pure VESDE **without training** on the CelebA256 dataset (a human face dataset). From the qualitative perspective, similar to our synthetic experiments (Sec. 7.2), we observe that drifted VESDE can generate more details compared to pure VESDE.

(a) Setting. In this experiment, we adapt well-known VESDE implementation [2] and do experiments on CelebA datasets (size: 256\*256\*3 ). More specifically, we use ve/celebahq_256_ncsnpp_continuous checkpoints provided by [2] and modify the sampling process strictly according to our drifted VESDE. To do a fair comparsion, we fix the random seed and use the reverse PFODE process. Then, we generate 10000 face images to calculate the metrics. We note that when using this checkpoint and pure VESDE pipeline provided by [2], the models would generate almost pure noise with a certain probability. Hence, we use an aesthetic predictor [1] (aesthetic score>=5.5) to filter the generated images to ensure that the images are clear faces.

(b) Discussion. From the qualitative perspective, as shown in the experiment results (**please click our PDF to see the generated images**), the images generated by our drifted VESDE have more detail (such as hair and  beard details). On the contrary, since pure VESDE can not deal with large variance, the images generated by pure VESDE appear blurry and unrealistic in these details. From the quantitative results, our drifted VESDE achieves aesthetic score **5.813**, and IS **4.174**, which is better than the results of baseline pure VESDE (aesthetic score 5.807 and IS: 4.082).

In conclusion, the real-world experiments show the potential of our drifted VESDE, and we will make it clearer in the next version of the paper.

We note that the goal of these experiments is to show that our conservative drifted VESDE is plug-and-play without training instead of achieving a SOTA performance. Hence, we focus on the relative improvement compared to the baseline [2]. There are two interesting empirical future works. For the conservative drifted VESDE,  we will do experiments on the SOTA pure VESDE models [3] and improve their results without training. For the aggressive drifted VESDE, since this process makes a larger modification compared with the conservative one, we need to train a new score function instead of directly using a pre-train one to achieve better results. We will add these discussions to the future work paragraph.

[1]  Christoph Schuhmann. Laion-aesthetics. 2022.

[2]  Song, Y., Sohl-Dickstein, J., Kingma, D. P., Kumar, A., Ermon, S., & Poole, B. (2020). Score-based generative modeling through stochastic differential equations. *arXiv preprint arXiv:2011.13456*.

[3] Karras, T., Aittala, M., Aila, T., & Laine, S. (2022). Elucidating the design space of diffusion-based generative models. *Advances in neural information processing systems*, *35*, 26565-26577.

---

### Author Response · Authors · 2024-08-12

We would like to thank the AC and reviewers again for their time and efforts! We appreciate that the reviewers highlight the contribution of this paper, such as the novel drifted VESDE, the non-trivial tangent process control lemma, and the rigorous analysis.

Also, reviewers provided valuable comments and suggestions to help us polish our work. The two common concerns are (1) the real-world experiments and (2) the extension of our general drifted VESDE. For the real-world experiments, we show that our drifted VESDE is a **plug-and-play** method without training and can generate more realistic images compared with the baseline from the quantitative and qualitative perspective. For the extension of drifted VESDE, we show that drifted VESDE is a general formula (including VP and VESDE) and can provide polynomial results for each setting. Furthermore, the drifted VESDE can go beyond the current VESDE, balances the reverse beginning and discretization error terms, and achieves better results for VE-based models.

We have provided rebuttals for each reviewer's concern (including the above discussion and other concerns such as the general $W\_p$ guarantee and the technique novelty of the reverse PFODE). We are more than happy to discuss our work in detail and answer any further questions in the rebuttal phase.

Thank you very much for your helpful review.

---

### Decision · Program_Chairs · 2024-09-25

**Decision:**

Accept (poster)

**Comment:**

The paper received positive reviews from most reviewers and the several concerns have been clarified during the rebuttal period. After discussing with reviewer bHCt, I have come to the conclusion that the paper merits an acceptance, given all the promised changes/clarifications being implemented. Please make sure you carefully incorporate all the reviews/discussions into account.